# PCB-Bench: Benchmarking LLMs for Printed Circuit Board Placement and Routing

**Jindong Li**[*], **Lianrong Chen**[*], **Bin Yang**[*], **Jiadong Zhu**[*], **Ying Wang**, **Yuzhe Ma**, **Menglin Yang**[†]
The Hong Kong University of Science and Technology (Guangzhou)

## Abstract

Recent advances in Large Language Models (LLMs) have enabled impressive capabilities across diverse reasoning and generation tasks. However, their ability to understand and operate on real-world engineering problems, such as Printed Circuit Board (PCB) placement and routing, remains underexplored due to the lack of standardized benchmarks and high-fidelity datasets. To address this gap, we introduce **PCB-Bench**, the first comprehensive benchmark designed to systematically evaluate LLMs in the context of PCB design. PCB-Bench spans three complementary task settings: **(1) text-based** reasoning with approximately 3,700 expert-annotated instances, consisting of over 1,800 question-answer pairs and their corresponding choice question versions, covering component placement, routing strategies, and design rule compliance; **(2) multimodal image-text** reasoning with approximately 500 problems requiring joint interpretation of PCB visuals and technical specifications, including component identification, function recognition, and visual trace reasoning; **(3) real-world design comprehension** using over 170 complete PCB projects with schematics, placement files, and design documentation. We design structured evaluation protocols to assess both generative and discriminative capabilities, and conduct extensive comparisons across state-of-the-art LLMs. Our results reveal substantial gaps in current models' ability to reason over spatial placements, follow domain-specific constraints, and interpret professional engineering artifacts. PCB-Bench establishes a foundational resource for advancing research toward more capable engineering AI, with implications extending beyond PCB design to broader structured reasoning domains. Data and code are available at `https://github.com/digailab/PCB-Bench`.

## 1 Introduction

Printed Circuit Board (PCB) placement and routing are essential yet challenging steps in the electronic design automation (EDA) workflow (Ling & Isa, 2023; Soon et al., 2024; Li et al., 2023; Vassallo & Bajada, 2024). These tasks require engineers to determine the precise spatial arrangement and interconnection of components, while strictly adhering to physical, electrical, and manufacturing constraints (Petkov & Ivanova, 2024). Historically, layout automation relied on classical EDA algorithms such as analytical/global placement and congestion-driven routing. More recently, advances in reinforcement learning (RL) have demonstrated competitive macro placement and joint place-and-route (P&R) learning at IC scale, while the first PCB-specific RL systems have begun to emerge (Cheng et al., 2018; Lin et al., 2019; Xu et al., 2009; Mirhoseini et al., 2021; Cheng & Yan, 2021; Li et al., 2023; Vassallo & Bajada, 2024). They remain limited to geometric objectives under fixed rules, without tackling multimodal reasoning over authentic PCB design artifacts.

In recent years, large language models (LLMs) (e.g., GPT-4o (Hurst et al., 2024), GPT-5 (OpenAI, 2025), Claude-Opus-4.1 (Anthropic, 2025), Gemini-2.5 (Comanici et al., 2025), DeepSeek-V3 (Liu et al., 2024), DeepSeek-V3.1 (AI, 2025), Qwen-2.5 (Team et al., 2024), LLaMA-2 (Touvron et al., 2023), LLaMA-3 (Dubey et al., 2024), ) have demonstrated remarkable capabilities in open-ended reasoning, code generation, semantic understanding, and even multimodal grounding. This progress has sparked increasing interest in applying these models to engineering domains, where structured

---

[*]Equal contribution.
[†]Corresponding author.

Table 1: The comparison between publicly available benchmarks. (DSLR: digital single-lens reflex)

| Dataset | Visual Data | Text | Schematic Diagram | Source Code | Manufacture Setting | Software Environment | Collection Methods | Purpose (Task) | Year |
|---|---|---|---|---|---|---|---|---|---|
| PCB-DSLR (Pramerdorfer & Kampel, 2015) | RGB Image | ✗ | ✗ | ✗ | ✓ | ✗ | DSLR | Detection | 2015 |
| PCBA-defect (Huang & Wei, 2019) | RGB Image | ✗ | ✗ | ✗ | ✓ | ✗ | Digital Microscope | Detection | 2019 |
| DeepPCB (Tang et al., 2019) | Binary Image | ✗ | ✗ | ✗ | ✓ | ✗ | Linear Scan CCD | Detection | 2019 |
| PCB-Metal (Mahalingam et al., 2019) | RGB Image | ✗ | ✗ | ✗ | ✓ | ✗ | DSLR | Detection | 2019 |
| FICS-PCB (Lu et al., 2020) | RGB Image | ✗ | ✗ | ✗ | ✓ | ✗ | Digital Microscope, DSLR | Detection | 2020 |
| FICS-PCB X-ray (Mehta et al., 2022) | 3D Volumetric Data | ✗ | ✗ | ✗ | ✓ | ✗ | X-Ray Machine | Detection | 2022 |
| FPIC (Jessurun et al., 2023) | RGB Image | ✓ | ✗ | ✗ | ✓ | ✗ | DSLR | Detection | 2023 |
| PCB-Vision (Arbash et al., 2024) | RGB Image, Hyperspectral Cubes | ✗ | ✗ | ✗ | ✓ | ✗ | DSLR, Linescan Spectrometer | Detection | 2024 |
| **PCB-Bench (ours)** | RGB Image | ✓ | ✓ | ✓ | ✗ | ✓ | Human Expert, PCB Design Software, PCB Design File | PCB Design | 2025 |

constraints and real-world semantics intertwine. However, a critical question remains largely unanswered: can these models truly understand and operate on expert-level engineering tasks that require precise interpretation of domain conventions, physical representations, and high-level design logic?

**The Challenge of PCB Evaluation.** The application of foundation models to PCB evaluation faces several fundamental challenges. First, PCB design requires reasoning across multiple modalities such as textual specifications, visual layouts, and structured design files, while maintaining strict adherence to domain-specific rules and constraints. Second, the scarcity of high-quality, large-scale datasets limits systematic evaluation. PCB design is costly, with labor reaching 0.2∼0.5 USD per pin, and designs must undergo expensive validation through hardware fabrication and testing. Due to time, cost, and intellectual property constraints, PCB data is rarely open-sourced, making large-scale evaluation difficult. Even in state-of-the-art academic research (circuit-board-related research field, e.g., (Vassallo & Bajada, 2024), *FanoutNet* (Li et al., 2023)), experiments are typically conducted on fewer than 20 boards, severely limiting reproducibility and benchmarking.

**Current Limitations in LLM Evaluation.** While existing research has explored learning-based approaches for physical design optimization, including deep reinforcement learning for IC macro floorplanning (Mirhoseini et al., 2021) and joint P&R learning (Cheng & Yan, 2021), these approaches primarily optimize geometric objectives under fixed constraints. Similarly, recent work evaluating LLMs in EDA contexts has focused on text-to-HDL design assistants (Chang et al., 2023) and Verilog code generation (Liu et al., 2023). However, none of these directions systematically evaluate cross-modal alignment (text ↔ image ↔ design files) and rule-driven reasoning under PCB constraints, leaving a clear evaluation gap for foundation models on engineering tasks.

**Our Contributions: PCB-Bench.** As shown in Table 1, no existing benchmarks could support the evaluation of LLMs for PCB design. Therefore, we introduce **PCB-Bench**, the first comprehensive benchmark to systematically evaluate LLMs on real-world PCB placement and routing tasks. PCB-Bench spans three complementary task settings and incorporates three key modalities: (1) text-based reasoning (e.g., design requirements and technical questions); (2) multimodal image-text reasoning (e.g., annotated or raw PCB visuals); and (3) real-world design comprehension, such as schematic diagrams, design intents, and routing files.

This multi-modal composition reflects real PCB design, capturing abstract semantics and physical constraints. It aligns with LLMs/MLLMs: text tests reasoning, images assess spatial grounding, and structured files provide domain knowledge. Together, these modalities form a challenging testbed for evaluating foundation models on complex engineering problems.

The first task in PCB-Bench is a large-scale *text-to-text Question and Answer (QA) and Choice Question (CQ) benchmark*, containing about 1,800 free-form QAs with corresponding CQs, for a total of roughly 3,700 questions. This dual-format design enables evaluation of both generative reasoning with **BERTScore** and **SBERT** similarity, and factual accuracy in CQ via **accuracy**. The second task is *image-and-text multimodal reasoning*, comprising about 500 questions, including choice, cloze-style, and free-form QA formats. Each problem pairs a natural language prompt with a PCB placement image, requiring models to integrate visual and textual information for accurate reasoning. The third task centers on *understanding real-world PCB designs*. We collect over 170 complete designs from OSHWHub[1], each including schematic diagrams, design intents, component libraries,

---
[1] https://oshwhub.com/ (operated by JLCPCB), an EDA platform providing accessible PCB designs.

and final placement and routing files. From these, we extract representative editor screenshots as inputs for description tasks, where models are asked to explain board functions, structural features, or application scenarios, thereby evaluating their ability to interpret professional design artifacts.

Our evaluation shows that current LLMs struggle with reasoning under real-world engineering constraints, often producing errors in rule adherence, logical consistency, or component semantics. Especially in multimodal settings, performance varies across models and task types, suggesting that significant gaps remain in adapting foundation models to structured, high-stakes design scenarios. **Our contributions are summarized as follows:**

- *Comprehensive Multimodal Benchmark*. We propose **PCB-Bench**, to our knowledge the first comprehensive multimodal benchmark for assessing LLM/MLLM capabilities in PCB placement and routing, featuring text-based QA, image-text reasoning, and schematic-to-description tasks.
- *High-Quality Dataset Curation.* We curate and release a high-quality dataset of over 170 real-world PCB designs, providing a valuable resource for research on LLM-driven PCB design.
- *Systematic Model Evaluation.* We systematically evaluate state-of-the-art models across multiple tasks and modalities, uncovering limitations in reasoning, compliance, and visual-semantic alignment.
- *Standardized Research Protocols.* We provide unified task formats, evaluation metrics, and prompt design protocols to support reproducible research and standardized comparison in this emerging field.

**PCB-Bench** serves as a foundation for future research into engineering-capable foundation models and offers a standardized platform to evaluate real-world readiness in PCB design automation.

## 2 RELATED WORK

In recent years, several PCB datasets have been introduced, covering tasks such as defect detection, component recognition, and multimodal inspection. These resources provide valuable benchmarks for evaluating computer vision and deep learning methods across different application scenarios. PCB-DSLR (Pramerdorfer & Kampel, 2015) provides 748 images from 165 boards with annotations for segmentation, IC bounding boxes, and chip text, supporting classification and recognition tasks. PCBA-defect (Huang & Wei, 2019) includes 1,386 images with annotated defects across six categories for automated optical inspection. DeepPCB (Tang et al., 2019) offers 1,500 aligned image pairs of defect-free templates and defective boards for defect detection and localization. PCB-METAL (Mahalingam et al., 2019) contains 984 images with bounding boxes for major components, enabling component-level analysis. FICS-PCB (Lu et al., 2020) provides 9,912 images with over 77k annotated components for robust detection and classification, while its X-ray extension (Mehta et al., 2022) introduces tomographic data for inter-layer inspection. FPIC (Jessurun et al., 2023) offers 261 images with fine-grained annotations of over 71k instances including components, silkscreen text, and logos. PCB-Vision (Arbash et al., 2024) integrates 53 RGB images with paired hyperspectral cubes for multimodal component detection and recycling-oriented inspection. Additional benchmarks such as CIRCUIT (Skelic et al., 2025) and EEE-Bench (Li et al., 2025) target general circuit or electronics reasoning rather than PCB-specific layout and routing, and are therefore complementary rather than overlapping with our setting. Concurrently, LLM-based assistants for text-to-HDL and HDL code-generation benchmarks with simulation-verified correctness examine textual competence (Chang et al., 2023; Liu et al., 2023), while position papers discuss opportunities and limitations of LLMs in EDA workflows (He & Yu, 2024). Due to space constraints, more detailed information is provided in Appendix A.

## 3 PCB-BENCH

### 3.1 DATASET DESCRIPTION

PCB-Bench integrates three types of datasets to support multimodal evaluation of (Multimodal) Large Language Models in PCB placement and routing tasks. These datasets are constructed inde-

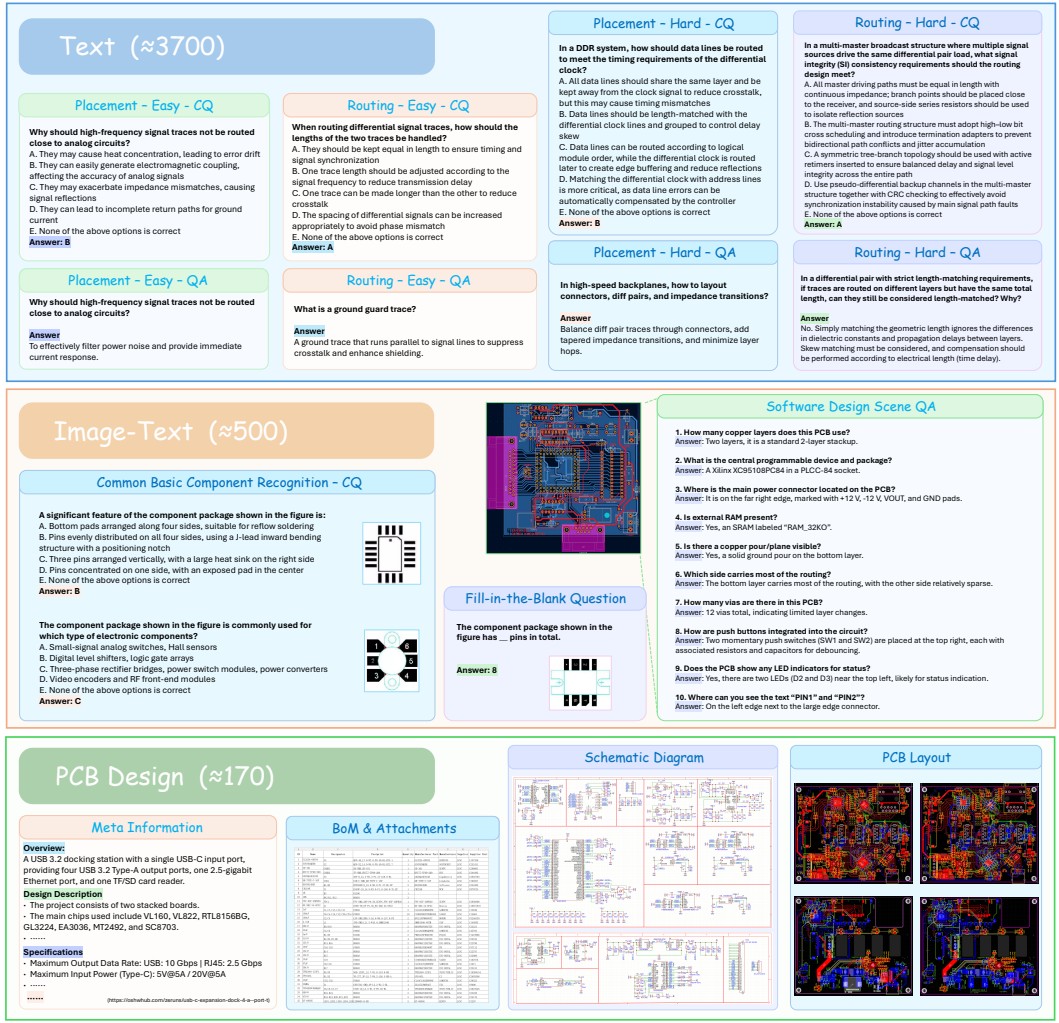

Figure 1: Overview of PCB-Bench with representative cases.

pendently to reflect different reasoning modalities and input formats, including text-only, image-text, and real-world placement screenshots (see Figure 1). More details are provided in Appendix B.

**Text-to-Text QA and CQ Set.** We construct a separate collection of over 1,800 text-based question-answering (QA) instances, independent of the PCB design repository. For each QA instance, we additionally provide a corresponding single-choice question, resulting in a total of about 3,700 questions. These questions are written by domain experts to cover core placement and routing concepts across two levels, namely macro-level and micro-level, where the former emphasizes global design principles (e.g., module placement, power/ground planning) and the latter targets fine-grained implementation details and complex scenarios (e.g., signal integrity, high-speed routing). In total, they span over 25 subtopics (e.g., signal integrity, DFM, EMI/EMC, power delivery, differential pairs). Each QA point contains both a free-form open-ended question and a structured CQ version, enabling both generative and discriminative evaluation. All instances are manually reviewed and annotated with topic labels (see Figure 2 for an overview of the QA topic categories).

**Image-and-Text QA and CQ Set.** We construct a multimodal dataset of about 500 instances that combine PCB layout images with textual prompts. Specifically, it includes 300 single-choice questions, 100 cloze-style fill-in-the-blank questions, and 100 open-ended QA instances. The questions cover diverse visual-semantic subtasks such as component identification, functional block recognition, trace type detection, via presence checking, and differential pair continuity analysis. The layout

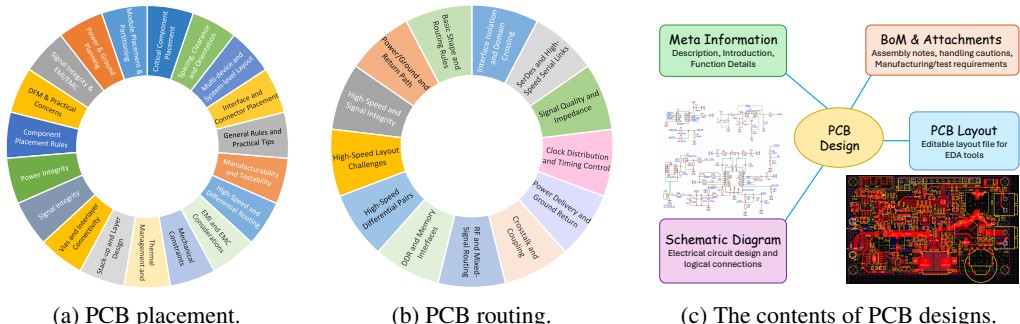

(a) PCB placement.          (b) PCB routing.          (c) The contents of PCB designs.

Figure 2: The category information of PCB-Bench.

images are collected from both real and simulated PCB designs, ensuring visual clarity, semantic richness, and engineering relevance. This dataset supports both objective evaluation through CQ and cloze questions, as well as semantic evaluation through free-form QA.

**Real-World PCB Design Repository.** We collect 174 complete PCB designs (Figure 2) from JL-CPCB[1]. Each design includes schematic diagrams, placement and routing files, design descriptions, component libraries, and representative screenshots from EDA software. Some designs also include video tutorials explaining the design intent and construction process. These designs span a diverse range of applications, such as USB hubs, power modules, and development boards, and are primarily used in PCB-Bench for screenshot-based understanding tasks. This repository serves as a rich resource for future supervised training and pretraining on realistic EDA artifacts.

More detailed descriptions of these datasets are provided in Appendix B. And all datasets are verified for correctness, cleaned for language clarity, and formatted for compatibility with open-source and proprietary foundation models. We release PCB-Bench with open licensing to support reproducibility, evaluation standardization, and future model development in LLM-based EDA tasks.

## 3.2 TASK FORMULATION

PCB-Bench consists of three task settings designed to evaluate the multimodal understanding and reasoning abilities of (Multimodal) Large Language Models in the context of PCB placement and routing. These tasks are formulated based on real-world engineering scenarios and reflect how human designers interpret, analyze, and describe PCB designs through textual, visual, and schematic information. Each task aligns with a specific modality combination and reasoning objective:

**Task 1: Text-to-Text QA and CQ.** This task is constructed from 1,800+ text-based QA instances derived from realistic PCB placement and routing contexts. For each QA instance, we additionally provide a corresponding choice question, resulting in a total of approximately 3,700 questions. The free-form QA setting reflects how models respond under open prompts, where performance may vary depending on prompt design or model-specific generation behavior. In contrast, the choice format offers a more objective and standardized evaluation, though it is inherently easier since models can resort to guessing or elimination strategies when lacking domain knowledge. The questions span two major domains, placement and routing, and are further categorized by difficulty (Easy and Hard) and subtopics such as signal integrity, EMI, power planning, and high-speed routing.

**Task 2: Image-and-Text Multimodal QA and CQ.** This task evaluates a model's ability to answer questions based on both PCB layout images and accompanying textual prompts. It includes approximately 400 questions, presented in either choice question or cloze-style (fill-in-the-blank) formats, and 100 QA-form questions. The questions cover a wide range of visual-semantic subtasks, including component recognition, functional block identification, trace type detection, signal layer classification, and via presence checking. Each instance presents a real PCB placement image along with a natural language query. This task assesses the model's ability to align textual understanding with spatial and symbolic patterns present in PCB images.

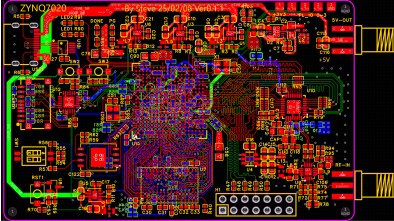 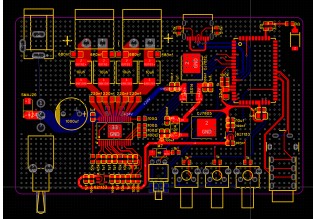 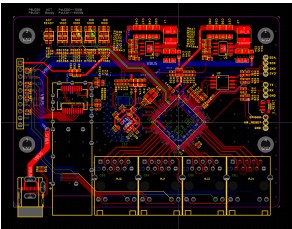

Figure 3: Several samples of PCB designs (Screenshots of PCB design in the development software/environment for presentation).

**Task 3: PCB Design Understanding.** This task focuses on high-level interpretation of professional PCB placement artifacts through visual input alone. We utilize the 174 real-world PCB designs collected from OSHWHub, from which we extract representative placement and routing screenshots (Figure 3) directly from EDA tools (e.g., editor views with silkscreen, copper layers, or via/pad structures). Each screenshot is used as a standalone input, without accompanying textual descriptions or schematic information. The model is prompted to generate a free-form textual description that explains the board's function, structure, or intended application scenario. Unlike Task 2, which provides both image and language input for classification-based answering, this task requires models to produce open-ended, coherent descriptions based solely on visual placement cues. It tests a model's ability in structured visual interpretation such as identifying key component zones, understanding spatial partitioning, and inferring board-level functionality, based on real, noisy, and heterogeneous PCB design screenshots. This setting simulates how human engineers inspect a PCB visually to form an initial technical assessment before delving into schematics or netlists.

Together, these tasks provide a comprehensive and multimodal framework for evaluating the capabilities of foundation models on practical, structured engineering problems.

## 3.3 BENCHMARK CONSTRUCTION

The construction of PCB-Bench follows a rigorous pipeline of knowledge collection and processing, as illustrated in Figure 4. We begin by gathering knowledge from multiple complementary sources, including textbooks, web resources, outputs of commercial multimodal LLMs, domain experts, and syllabi from PCB education institutes. This ensures that the foundation of the dataset is both comprehensive and aligned with real engineering practice.

Based on these sources, human experts first design a structured syllabus and refine the scope of knowledge points that should be covered. Each selected knowledge point is then transformed into candidate questions through a multi-stage generation process. This process involves three key steps: (i) designing questions in both open-ended and structured forms, (ii) standardizing the annotation schema and formatting rules to maintain consistency, and (iii) performing quality assurance to minimize ambiguity or redundancy.

All generated questions undergo iterative human expert review to guarantee correctness, clarity, and practical relevance. The review process not only validates technical accuracy but also adjusts phrasing to better reflect authentic engineering reasoning and instructional suitability.

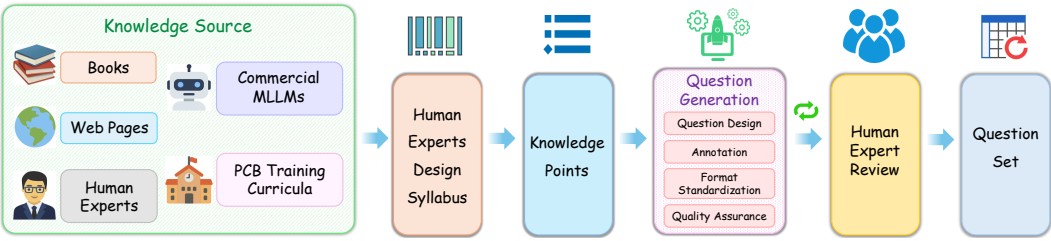

Figure 4: The pipeline of knowledge collection and processing.

The benchmark further benefited from feedback provided by engineers from a PCB design and manufacturing company, who validated the practical relevance of the tasks. While they were not involved in dataset authoring, their input ensured alignment with real industrial workflows.

In addition, the PCB designs used in PCB-Bench are collected from the open-source platform OS-HWHub. During data collection, we strictly comply with open-source licensing terms, and for each design, we provide the corresponding URL link to ensure transparency and the protection of intellectual property. More details are provided in Appendix B.1.

## 4 EXPERIMENTS AND FINDINGS

### 4.1 EXPERIMENTAL SETUP

We evaluate a diverse set of (M)LLMs under a unified **zero-shot** setting across all three tasks in **PCB-Bench**. Each question or input instance is presented to the model independently, without access to prior examples or task-specific demonstrations. This simulates realistic deployment scenarios, where foundation models are expected to generalize directly to engineering problems without fine-tuning or prompt tuning. The detailed task formulations have been introduced in Section 3.

**Task 1.** Text-based QA tasks are used to assess models' understanding of PCB design knowledge. The following models are evaluated: GPT-5 (OpenAI, 2025), GPT-4o (Hurst et al., 2024), Claude-Opus-4.1 (Anthropic, 2025), Gemini-2.5-Pro Comanici et al. (2025), Deepseek-Chat-V3.1-671B (AI, 2025), Qwen2.5-7B-Instruct (Team et al., 2024), LLaMA-4-Maverick-400B (Meta AI, 2025), InternVL3-78B (Zhu et al., 2025), MythoMax-L2-13B (Gryphe, 2024), Rocinante-12B (The-Drummer, 2025), Ministral-3B (AI, 2024). We further evaluate two **domain-specific** variants derived from Qwen2.5-7B-Instruct to examine whether PCB-oriented specialization yields measurable gains. **Task 2.** Multimodal QA tasks combine PCB layout images and textual queries. We evaluate: GPT-5 (OpenAI, 2025), GPT-5-nano (OpenAI, 2025), Gemini-2.5-Pro (Comanici et al., 2025), Gemini-2.5-Flash-Lite (Comanici et al., 2025), LLaMA-4-Maverick-400B (Meta AI, 2025), Qwen-VL-Max (Bai et al., 2023). InternVL3-78B (Zhu et al., 2025), Qwen3-VL-8B-Instruct (Yang et al., 2025), Qwen3-VL-235B-A22B-thinking (Yang et al., 2025). **Task 3.** PCB layout understanding tasks require models to describe full-board screenshots. We evaluate the same set of vision-language models as in Task 2. More details are provided in Appendix C.1.

### 4.2 EVALUATION METRICS

For CQ, we adopt *top-1 accuracy*. For free-form QA, we employ *BERTScore* (Zhang et al., 2020) and *Sentence-BERT similarity (SBERT)* (Reimers & Gurevych, 2019) to assess semantic consistency with reference answers. Compared with exact string matching, these semantic metrics are more suitable for PCB layout and routing QA, where correct answers often involve technical terminology, synonyms, or varying descriptive expressions. For Task 3, we further report *precision*, *recall*, and *F1-score* to capture complementary aspects of prediction quality, ensuring a more comprehensive evaluation. More details are provided in Appendix C.2.

### 4.3 RESULTS AND ANALYSIS

The main results are summarized below, with more detailed results provided in Appendix C.

### 4.3.1 TASK 1: TEXT-BASED QA AND CQ PERFORMANCE

As shown in Table 2 and Figure 5, GPT-4o, Claude-Opus-4.1, and DeepSeek-Chat-V3 form the top tier: Claude leads CQ accuracy, while GPT-4o and DeepSeek show stronger semantic alignment in QA. GPT-5 remains competitive but slightly weaker in QA fidelity. Open-source models such as InternVL3-78B perform well on CQ but show lower QA alignment, and domain-specific variants (QLoRA, RAG) improve semantic similarity without surpassing frontier models. Overall, Task 1 exhibits complementary strengths across models and clear divergence between CQ accuracy and QA alignment metrics, reflecting the different reasoning demands of selection vs. generation tasks. These results highlight that semantic evaluation and accuracy-based evaluation do not always correlate, especially in fine-grained PCB reasoning scenarios.

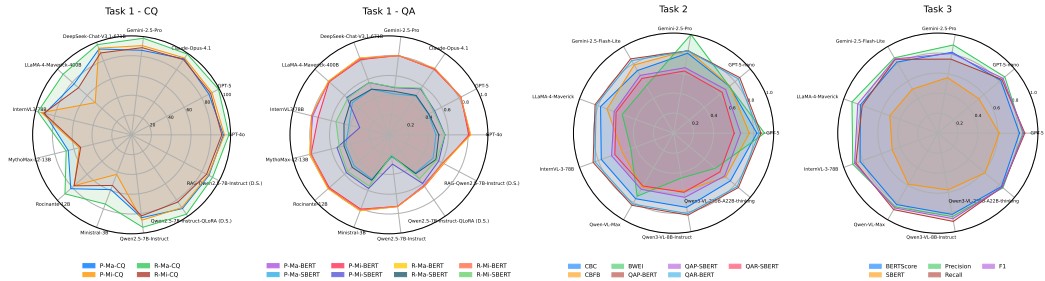

Figure 5: Radar visualization of model performance across different tasks. The charts highlight comparative strengths and weaknesses of each model under diverse evaluation metrics.

Table 2: Performance comparison on task 1. (Acc: Accuracy (%); D.S.: domain-specific. Notation: **best**; second-best; worst.)

| Model | Placement | | | | | | Routing | | | | | |
|---|---|---|---|---|---|---|---|---|---|---|---|---|
| | Macro-Level | | | Micro-Level | | | Macro-Level | | | Micro-Level | | |
| | CQ | QA | | CQ | QA | | CQ | QA | | CQ | QA | |
| | Acc. | BERTScore | SBERT | Acc. | BERTScore | SBERT | Acc. | BERTScore | SBERT | Acc. | BERTScore | SBERT |
| GPT-4o | 92.74 | 0.8244 | 0.4756 | 93.82 | 0.8069 | 0.5664 | 98.32 | 0.8267 | 0.5046 | 91.13 | 0.8169 | 0.5647 |
| GPT-5 | 88.27 | 0.8180 | 0.4502 | 91.79 | **0.8214** | 0.5334 | **99.16** | 0.8193 | 0.4588 | 90.17 | 0.8211 | 0.5137 |
| Claude-Opus-4.1 | **93.30** | 0.8138 | 0.4772 | **94.35** | 0.8054 | 0.5643 | **99.16** | 0.8164 | 0.4837 | **92.32** | 0.8119 | 0.5738 |
| Gemini-2.5-Pro | 86.03 | 0.8073 | 0.4172 | 90.82 | 0.8069 | 0.4804 | 98.31 | 0.8068 | 0.4428 | 88.73 | 0.8068 | 0.4840 |
| DeepSeek-Chat-V3.1-671B | 92.74 | **0.8306** | 0.4912 | 93.64 | 0.8127 | 0.5652 | 97.48 | **0.8310** | 0.4939 | 88.49 | **0.8217** | 0.5657 |
| LLaMA-4-Maverick-400B | 77.66 | 0.8155 | 0.4931 | 49.07 | 0.8074 | 0.5689 | 92.43 | 0.8186 | 0.5149 | 71.70 | 0.8104 | 0.5564 |
| InternVL3-78B | 90.50 | 0.7936 | 0.4162 | 93.91 | 0.7360 | 0.3035 | 97.48 | 0.8071 | 0.4677 | 91.37 | 0.8056 | 0.5511 |
| MythoMax-L2-13B | 65.92 | 0.8257 | 0.4496 | 54.37 | 0.8101 | 0.5406 | 68.07 | 0.8262 | 0.4676 | 53.00 | 0.8113 | 0.5173 |
| Rocinante-12B | 82.12 | 0.8174 | 0.4707 | 77.41 | 0.8054 | 0.5755 | 89.92 | 0.8227 | 0.4953 | 77.70 | 0.8141 | 0.5590 |
| Ministral-3B | 59.21 | 0.8196 | 0.4798 | 42.71 | 0.8027 | 0.5800 | 74.79 | 0.8204 | 0.4918 | 54.68 | 0.8108 | 0.5552 |
| Qwen2.5-7B-Instruct | 84.36 | 0.7302 | 0.2236 | 86.85 | 0.7287 | 0.2978 | 94.12 | 0.7305 | 0.2130 | 82.50 | 0.7270 | 0.2081 |
| Qwen2.5-7B-Instruct-*QLoRA* (*D.S.*) | 90.50 | 0.6203 | 0.4687 | 89.39 | 0.6329 | **0.5971** | 97.48 | 0.6297 | 0.4764 | 82.50 | 0.6267 | 0.5414 |
| *RAG*-Qwen2.5-7B-Instruct (*D.S.*) | 88.26 | 0.6145 | **0.5050** | 87.44 | 0.6259 | 0.5687 | 92.43 | 0.6234 | **0.5365** | 85.61 | 0.6122 | 0.5456 |

### 4.3.2 TASK 2: IMAGE-AND-TEXT MULTIMODAL QA AND CQ PERFORMANCE

Task 2 requires integrating textual prompts with PCB placement images for multimodal QA and CQ. As shown in Table 3 and Figure 5, GPT-5 delivers the most balanced performance, achieving strong CBC and BWEI accuracy while leading in QAP/QAR alignment. GPT-5-nano excels in semantic similarity but remains mid-tier in accuracy. Gemini-2.5-Pro attains top CBFB accuracy and perfect BWEI scores, though its semantic align-

Table 3: Performance comparison on task 2. (CBC: Common Basic CQ, CBFB: Common Basic Fill in Blank, BWEI: Basic Wiring Error Identification, QAP: QA of Placement, QAR: QA of Routing. Notation: **best**; second-best; worst.)

| Model | CBC | CBFB | BWEI | QAP | | QAR | |
|---|---|---|---|---|---|---|---|
| | Acc. (%) | Acc. (%) | Acc. (%) | BERTScore | SBERT | BERTScore | SBERT |
| GPT-5 | **83.26** | 75.67 | 90.90 | **0.8734** | 0.6700 | **0.8561** | 0.6055 |
| GPT-5-nano | 72.86 | 73.33 | 72.72 | 0.8633 | **0.6786** | 0.8443 | **0.6347** |
| Gemini-2.5-Pro | 81.08 | **84.00** | **100.00** | 0.8355 | 0.6637 | 0.8362 | 0.6311 |
| Gemini-2.5-Flash-Lite | 72.20 | 78.66 | 54.54 | 0.8570 | 0.6662 | 0.8361 | 0.6312 |
| LLaMA-4-Maverick | 77.60 | 70.66 | 54.54 | 0.8343 | 0.6310 | 0.8226 | 0.6108 |
| InternVL-3-78B | 76.83 | 54.66 | 45.45 | 0.8552 | 0.6588 | 0.8357 | 0.6242 |
| Qwen-VL-Max | 75.67 | 62.66 | 72.72 | 0.8321 | 0.6476 | 0.8170 | 0.6093 |
| Qwen3-VL-8B-Instruct | 75.28 | 58.66 | 45.45 | 0.8313 | 0.6510 | 0.8161 | 0.5997 |
| Qwen3-VL-235B-A22B-thinking | 74.13 | 68.00 | 54.54 | 0.8455 | 0.6658 | 0.8303 | 0.6037 |

ment trails GPT-5. Among open-source models, LLaMA-4-Maverick achieves moderate CBC accuracy (77.60%) but struggles with BWEI (54.54%). Qwen-VL-Max shows unremarkable results, while Qwen3-VL-235B-A22B-thinking demonstrates mid-tier performance despite chain-of-thought. InternVL-3-78B and Qwen3-VL-8B-Instruct exhibit the weakest BWEI performance (45.45%), with Qwen3-VL-8B-Instruct additionally achieving the lowest semantic similarity scores. Results reveal a clear trade-off between structured accuracy and semantic coherence, with no single model dominating all dimensions. This underscores the complexity of multimodal PCB understanding, where visual grounding and textual reasoning require sophisticated alignment mechanisms.

### 4.3.3 TASK 3: PCB DESIGN UNDERSTANDING.

The results for Task 3 are shown in Table 4 and Figure 5, which evaluate how well models interpret PCB designs from editor screenshots through free-form functional and structural descriptions. Overall, performance differences across models are relatively small: most achieve BERTScore around 0.82 and F1 in the 0.85~0.87 range. Qwen3-VL-235B-A22B-thinking and InternVL-3-78B deliver the strongest semantic alignment, with the highest BERTScore (0.8293 and 0.8267), while Qwen3-VL-8B-Instruct attains the best recall (0.8955) and competi-

Table 4: Performance comparison on task 3. (Notation: **best**; second-best; worst.)

| Model | BertScore | SBERT | Precision | Recall | F1 |
|---|---|---|---|---|---|
| GPT-5 | 0.8156 | 0.6134 | 0.8493 | 0.8691 | 0.8591 |
| GPT-5-nano | 0.8199 | 0.5358 | 0.8739 | 0.8352 | 0.8507 |
| Gemini-2.5-Pro | 0.8209 | 0.5626 | 0.8944 | 0.7507 | 0.8089 |
| Gemini-2.5-Flash-Lite | 0.8194 | 0.5083 | 0.8687 | 0.8518 | 0.8563 |
| LLaMA-4-Maverick | 0.8233 | 0.5101 | **0.9123** | 0.8044 | 0.8511 |
| InternVL-3-78B | 0.8267 | 0.4850 | 0.9044 | 0.8716 | **0.8849** |
| Qwen-VL-Max | 0.8236 | 0.5888 | 0.8501 | 0.8823 | 0.8628 |
| Qwen3-VL-8B-Instruct | 0.8242 | 0.5760 | 0.8463 | **0.8955** | 0.8675 |
| Qwen3-VL-235B-A22B-thinking | **0.8293** | **0.6168** | 0.8493 | 0.8391 | 0.8405 |

tive F1. LLaMA-4-Maverick reaches the highest precision (0.9123), reflecting strong pattern recognition but slightly lower SBERT. GPT-5 maintains balanced performance (F1 = 0.8591, SBERT = 0.6134), whereas GPT-5-nano shows moderate results across all metrics. These findings illustrate that while models can capture high-level PCB layout descriptions, deeper structural understanding remains challenging and varies across semantic metrics.

## 4.4 DISCUSSION AND FINDINGS

Across all tasks, CQ remains easier than free-form QA, while QA consistently shows lower semantic alignment. We also observe that semantic metrics such as BERTScore and SBERT do not always agree, reflecting that existing evaluation tools capture different aspects of answer quality and may miss PCB-specific correctness. Larger models outperform smaller ones, with Gemini-2.5, GPT-4o/5, Claude, and Qwen-VL-Max leading performance, and domain-specific variants (QLoRA fine-tuning, RAG) offering further gains. Despite these improvements, current LLMs still fall short in realistic PCB placement and routing, particularly in fine-grained or open-ended settings. These findings highlight the need for domain-adapted training, EDA-aware integration, and a dedicated evaluation system tailored to PCB design tasks. We can draw several preliminary conclusions:

```
1. Model behaviors differ both across tasks and dimensions (perspectives).
2. Larger-scale models perform better than smaller-scale models.
3. Overall, the strongest performance is achieved by Gemini-2.5, GPT-4o/5, Claude, and Qwen-
   VL-Max.
4. Domain-specific variants (e.g., QLoRA fine-tuning or RAG) can further boost the
   capabilities of general models.
5. Current LLMs show basic knowledge of PCB placement and routing but remain incomplete; in
   realistic open-ended settings they are still insufficient and cannot replace humans.
6. Meaningful progress will likely require domain-adapted training or integration with EDA-
   specific tools to bridge the gap between generic reasoning and PCB design requirements.
7. A dedicated, domain-specific evaluation system for PCB placement and routing is also
   needed to enable more rigorous and realistic capability assessment.
```

## 5 CONCLUSION

In this work, we present PCB-Bench, a multimodal benchmark designed to evaluate the capabilities of large language models on PCB placement and routing tasks. Our study demonstrates the feasibility of integrating text, image, and design artifacts into a unified evaluation framework, providing a first step toward systematic assessment of LLMs in PCB analysis.

## ACKNOWLEDGEMENTS

This work is supported by the Hong Kong University of Science and Technology (Guangzhou) (Grant No. G0101000266). We gratefully acknowledge the support of Qiyunfang Technology Co., LTD. We also thank OSHWHub and JLCPCB for providing access to publicly available PCB design resources.

ETHICS STATEMENT

The main contribution of this work lies in constructing PCB-Bench, a benchmark for systematically evaluating large language models (LLMs) on PCB placement and routing tasks. All PCB designs included in PCB-Bench are collected from publicly available and legally accessible sources, with no proprietary or sensitive industrial data involved. The benchmark and experiments are intended solely for academic research and evaluation, not for direct deployment in high-stakes industrial scenarios. By focusing on transparent dataset construction and responsible assessment of LLM capabilities, we aim to support the community in developing safer and more reliable AI techniques for electronic design automation.

REPRODUCIBILITY STATEMENT

To ensure reproducibility, we describe all experimental settings in detail, including model versions, evaluation metrics, and task formulations. All results are obtained under a unified zero-shot setting without additional fine-tuning, making the evaluation straightforward to replicate. Upon acceptance of this paper, we will release PCB-Bench along with the evaluation scripts and configuration files, enabling the community to reproduce our results and extend the benchmark. When releasing the benchmark, we will also comply with open-source licenses and clearly acknowledge all data sources.

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

APPENDIX CONTENTS

## A  RELATED WORK SUPPLEMENT MATERIALS

In Section 2, we present a concise overview of eight representative PCB datasets. For completeness, this appendix includes the extended versions with detailed statistics, annotation formats, and use cases, which complement the brief summaries in the main text.

PCB-DSLR (Pramerdorfer & Kampel, 2015) is a dataset for computer-vision-based PCB analysis, primarily targeting recycling applications. It contains 748 high-resolution DSLR images from 165 different boards, with annotations including board segmentation masks, 9,313 IC bounding boxes, and text labels for 1,740 chips. The dataset supports multiple tasks such as PCB classification, IC localization, and text recognition, and it provides a benchmark for evaluating vision-based PCB analysis methods under realistic recycling conditions.

PCBA-defect (Huang & Wei, 2019) is designed for PCB defect detection and classification. It consists of 1,386 color images of bare PCBs, each containing 3–5 defects from 6 common categories: missing hole, mouse bite, open circuit, short, spur, and spurious copper. The dataset provides bounding boxes and rotation information for every defect, enabling tasks such as detection and classification. Images are captured with a high-resolution industrial camera under controlled illumination, and the dataset serves as a benchmark for evaluating automated optical inspection methods.

DeepPCB (Tang et al., 2019) provides 1,500 aligned image pairs, each consisting of a defect-free template and a defective tested image. It covers 6 common defect categories, with each tested image containing around 3–12 annotated defects represented by bounding boxes. Images are captured by a linear scan CCD and aligned through template matching, ensuring high consistency for training and evaluation. DeepPCB serves as a benchmark for PCB defect detection, supporting both defect localization and classification tasks.

PCB-METAL (Mahalingam et al., 2019) provides 984 high-resolution images from 123 unique PCBs, each captured under controlled lighting and rotation to ensure consistency. It includes bounding box annotations for 5,844 ICs, 3,175 capacitors, 2,679 resistors, and 542 inductors, supplied in both text and XML (PASCAL VOC) formats. The dataset is designed for component-level PCB analysis, enabling tasks such as object detection, classification, and circuit design extraction.

FICS-PCB (Lu et al., 2020) introduces a multi-modal resource for automated PCB visual inspection, consisting of 9,912 images from 31 boards with 77,347 annotated components. It incorporates two imaging modalities, a digital microscope and a DSLR camera, and includes variations in illumination, scale, and sensor type to reflect realistic application scenarios. The dataset supports component-level detection and classification across 6 categories, and it serves as a benchmark for developing robust PCB assurance methods under diverse visual environments.

FICS-PCB X-ray (Mehta et al., 2022) is the first publicly available tomographic dataset targeting inter-layer PCB inspection. It consists of X-ray projection data, reconstructed 3D volumes converted into Tiff stacks, and annotated slices covering features such as vias, traces, and pads. The dataset contains five PCB samples of varying sizes and layer counts, enabling tasks such as inter-layer defect detection, feature segmentation, and automated netlist extraction. By providing volumetric data beyond surface-level imaging, FICS PCB X-ray facilitates research in hardware assurance, reverse engineering, and multimodal approaches that combine optical and X-ray inspection.

FPIC (Jessurun et al., 2023) is a large-scale semantic dataset for optical PCB assurance, consisting of 261 high-resolution RGB images from 93 boards with over 71,000 annotated instances, including components, silkscreen text, and logos. Unlike previous datasets that only provide binary or RGB images, FPIC offers fine-grained semantic annotations to support tasks such as component detection, text recognition, and hardware assurance. The dataset is collected using a DSLR camera under controlled conditions and is primarily designed for optical inspection and semantic-level PCB analysis.

PCB-Vision (Arbash et al., 2024) introduces the first multiscene benchmark integrating both RGB and hyperspectral imaging for PCB analysis. It consists of 53 high-resolution RGB images paired with corresponding hyperspectral cubes in the VNIR range, annotated for three primary component classes: integrated circuits, capacitors, and connectors. Designed in an industrial conveyor belt setting, PCB-Vision facilitates research on multimodal learning, component detection, and recycling-oriented PCB inspection under realistic conditions.

CIRCUIT (Skelic et al., 2025) focuses on analog circuit interpretation, providing a dataset of 510 question-answer pairs to evaluate LLMs' ability to reason about circuit topologies using both diagrams and netlists. The dataset spans a variety of analog circuit topics, challenging models with tasks that require multi-step reasoning and a deep understanding of circuit configurations.

EEE-Bench (Li et al., 2025) evaluates LLMs in electrical and electronics engineering, offering 2,860 problems across 10 subdomains such as analog circuits and power electronics. It combines visual (circuit diagrams, system images) and textual requirements, challenging models to reason across both modalities in real-world engineering tasks.

While the above resources advance imaging-based PCB inspection, they remain largely vision-only and do not assess whether foundation models can reason over textual requirements, layout images, and authentic design files in tandem. This gap motivates benchmarks that couple visual layouts with textual requirements and authentic design artifacts. For method-centric physical design, classical IC global placement casts optimization over density and wirelength, while global/detailed routing relies on congestion-aware Steiner construction and maze or A*-style search. (Cheng et al., 2018; Lin et al., 2019; Xu et al., 2009). Subsequent learning-driven physical design shows that deep RL can achieve human-competitive IC macro floorplans within hours (Mirhoseini et al., 2021), and follow-up work jointly learns placement and routing to improve end-to-end routability (Cheng & Yan, 2021). On PCBs, RL has been explored for component placement and fan-out automation in detailed routing (Li et al., 2023; Vassallo & Bajada, 2024). These optimize geometric/graph objectives under design rules, but none of these approaches evaluate multimodal understanding or natural-language reasoning over layouts and design files, which is the focus of our benchmark.

In contrast to the dataset-centric, optical-inspection line, the algorithm-centric placement/routing literature, and the text-centric LLM-for-HDL line, our work evaluates multimodal reasoning for PCB placement and routing across text, image, and real design artifacts such as schematics and editor screenshots. Thus, PCB-Bench complements algorithm-centric P&R and text-only HDL benchmarks by supplying a multi-modal, rule-aware evaluation target aligned with real EDA artifacts.

## B  PCB-BENCH SUPPLEMENT MATERIALS

Figure 6 presents the word clouds of PCB-Bench across different tasks and granularity levels, including placement and routing under both macro- and micro-levels. The visualization highlights the frequent occurrence of domain-specific concepts such as power, ground, thermal, signal, and copper, which correspond to key concerns in PCB design like power integrity, signal integrity, electromagnetic compatibility, and thermal management.

These word clouds serve as an intuitive validation of the dataset construction. They demonstrate that PCB-Bench captures a wide range of terminology and technical aspects at different abstraction levels, thereby ensuring both breadth and depth of coverage for evaluating LLMs and MLLMs on PCB-related reasoning tasks.

### B.1  PCB-BENCH CONSTRUCTION

To further strengthen reliability, the entire curation and annotation workflow was supervised by two domain experts in circuit and PCB design—one senior professor and one PhD-level researcher. Every question was either authored or reviewed by these experts, ensuring that all knowledge points, reasoning steps, and answer choices are technically precise. When disagreements emerged, such as ambiguous wording or borderline-acceptable distractors, experts discussed the case and refined the item until reaching full consensus on both the target reasoning path and the correct answer. Although we did not compute formal inter-annotator agreement, each item in PCB-Bench has undergone expert oversight and revision, guaranteeing high-quality and instructionally sound content.

In addition, the construction of PCB-Bench benefited from external validation provided by engineers from an industrial PCB design and manufacturing company. Their feedback helped verify that the tasks align with practical engineering workflows and reflect real-world design considerations. While these engineers did not participate directly in dataset creation or authorship, their input ensured that the benchmark remains grounded in authentic industrial practice.

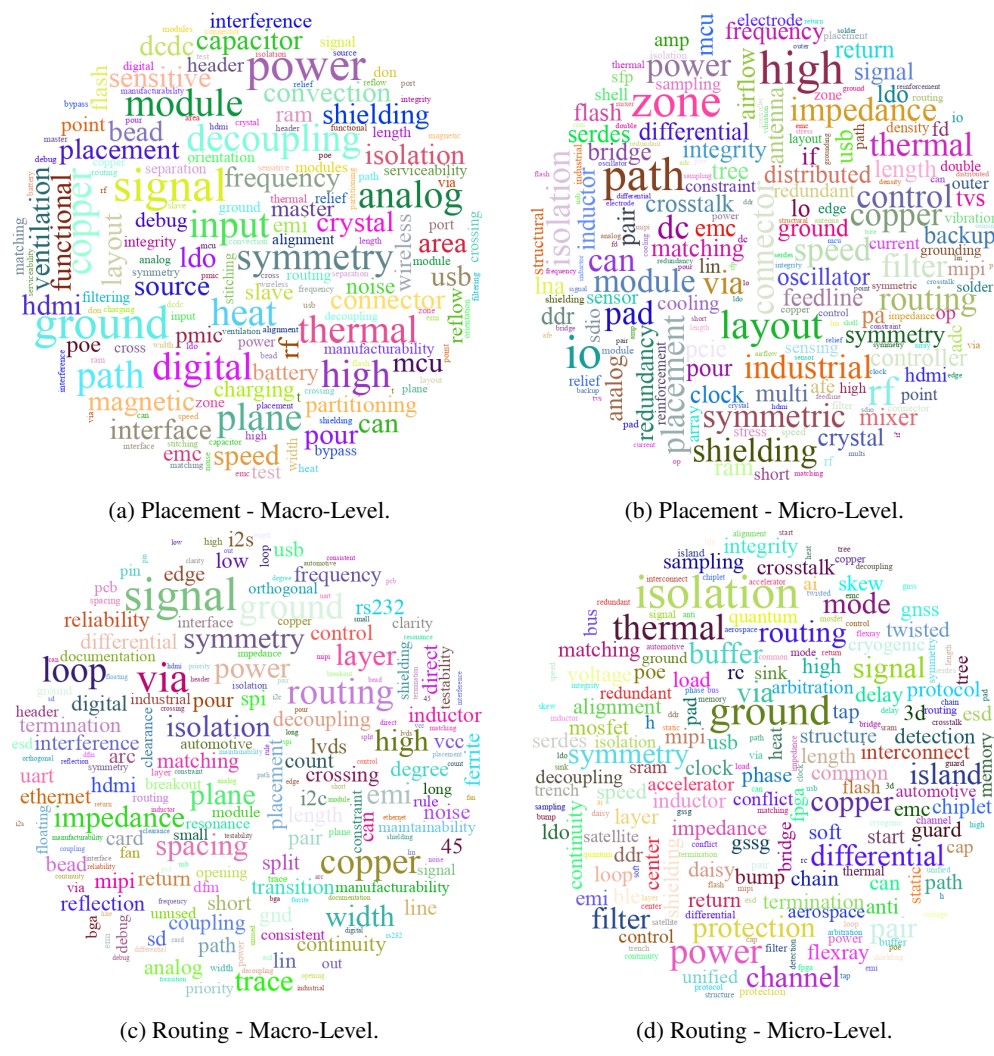

(a) Placement - Macro-Level.

(b) Placement - Micro-Level.

(c) Routing - Macro-Level.

(d) Routing - Micro-Level.

Figure 6: The word cloud of PCB-Bench.

# C EXPERIMENTS SUPPLEMENT MATERIALS

## C.1 EXPERIMENTAL SETUP SUPPLEMENT MATERIALS

**Domain-Specific Model 1: QLoRA-Tuned Qwen2.5-7B.** We fine-tuned Qwen2.5-7B-Instruct using a curated PCB corpus consisting of approximately 3,000 samples extracted from authoritative engineering documents (e.g., RK3588 Design Guide) via rule-based templates. The dataset integrates reasoning-focused multiple-choice questions and open-ended technical QA. The model is trained with 4-bit QLoRA (rank = 16, $\alpha$ = 32) to achieve efficient adaptation under limited computational cost. This specialized model exhibits substantial improvements in semantic alignment (SBERT) compared with its base version, confirming that lightweight domain tuning can enhance PCB knowledge comprehension.

**Domain-Specific Model 2: RAG-Enhanced Qwen2.5-7B.** We additionally construct a Retrieval-Augmented Generation (RAG) system based on Qwen2.5-7B-Instruct. The retrieval corpus contains 178 PCB-related documents ($\approx$ 4.5M tokens), indexed using FAISS with all-MiniLM-L6-v2 embeddings (chunk size 512/50, Top-K = 8). The architecture retrieves relevant PCB knowledge before generation, enabling more grounded reasoning. As shown in Table 2, the RAG-enhanced model

significantly surpasses the standalone Qwen baseline in accuracy and hallucination reduction, particularly on complex engineering queries requiring factual grounding.

## C.2 Evaluation Metrics Supplement Materials

We include BERTScore and SBERT as they are widely adopted semantic evaluation metrics in the community. Since our open-ended tasks emphasize semantic correctness and factual grounding, exact-match metrics alone are insufficient. Embedding-based metrics provide a more reliable assessment for technical QA and domain-specific language generation, as demonstrated in prior work (Gehrmann et al., 2021; Chang et al., 2024; Manakul et al., 2023; Zhou et al., 2024; Gao et al., 2021; Saakyan et al., 2021; Liu et al., 2025).

BERTScore (Zhang et al., 2020) compares candidate and reference sentences in a semantic space using contextual embeddings from BERT (Devlin et al., 2019), which computes token-level cosine similarities and aggregates them into precision, recall, and F1 scores. Unlike n-gram overlap metrics such as BLEU (Papineni et al., 2002) and ROUGE (Chin-Yew, 2004), BERTScore prioritizes semantic similarity over surface-form matching, making it more robust to paraphrasing. We adopt the F1 aggregation (BERTScore-F1) and refer to it as BERTScore throughout.

SBERT (Reimers & Gurevych, 2019) adapts BERT (Devlin et al., 2019) using Siamese or Triplet architectures (Bromley et al., 1993; Schroff et al., 2015) to produce fixed-size sentence embeddings. Compared with vanilla cross-encoder BERT, SBERT allows efficient computation of semantic similarity by encoding sentences into embeddings, between which cosine similarity is measured to quantify sentence-level proximity.

We also report token-level Precision, Recall, and F1 based on lexical overlap between the prediction and the reference. Let $\mathcal{P}$ be the set of tokens in the prediction and $\mathcal{R}$ the set of tokens in the reference; we define:

$$\text{Precision} = \frac{|\mathcal{P} \cap \mathcal{R}|}{|\mathcal{P}|}, \qquad \text{Recall} = \frac{|\mathcal{P} \cap \mathcal{R}|}{|\mathcal{R}|}, \qquad F1 = \frac{2 \cdot \text{Precision} \cdot \text{Recall}}{\text{Precision} + \text{Recall}}. \qquad (1)$$

Precision measures exactness, Recall measures coverage, and F1 summarizes both via the harmonic mean.

To further examine how existing semantic metrics behave in PCB reasoning tasks, Figure 7 visualizes the relationship between BERTScore and SBERT across all three tasks. The results reveal a consistent pattern: although both metrics aim to quantify semantic similarity, they capture different aspects of model outputs. BERTScore, being token-alignment–oriented, tends to reward surface-form overlap, while SBERT, as a sentence-embedding metric, is more sensitive to global structural coherence. As a consequence, models do not lie along a single diagonal trend; their positions vary across tasks and modalities.

This divergence is particularly visible in Task 1, where free-form QA often leads to lexical variations that affect BERTScore more than SBERT, with Qwen2.5 and its domain-specific variants showing especially large gaps between the two metrics. Task 2 exhibits tighter clustering, reflecting more structured answers driven by visual grounding. Task 3 shows the smallest spread in BERTScore but broader variation in SBERT, indicating that layout-understanding descriptions differ more in discourse structure than in token choice. Together, these observations highlight that general-purpose semantic metrics do not provide a unified or stable signal for evaluating PCB reasoning quality, especially when answers differ in style, granularity, or engineering specificity.

## C.3 Prompt Template

To ensure fair and meaningful evaluation, we also provide the prompt templates used in our experiments. These templates are derived from extensive practical testing under the current accessibility settings of mainstream LLMs, aiming to preserve formatting consistency while allowing models to fully express their reasoning capabilities. When using this benchmark to evaluate other models, please first verify that the chosen template enables the model to perform at its best. Our templates serve as a reference rather than the only or the optimal choice.

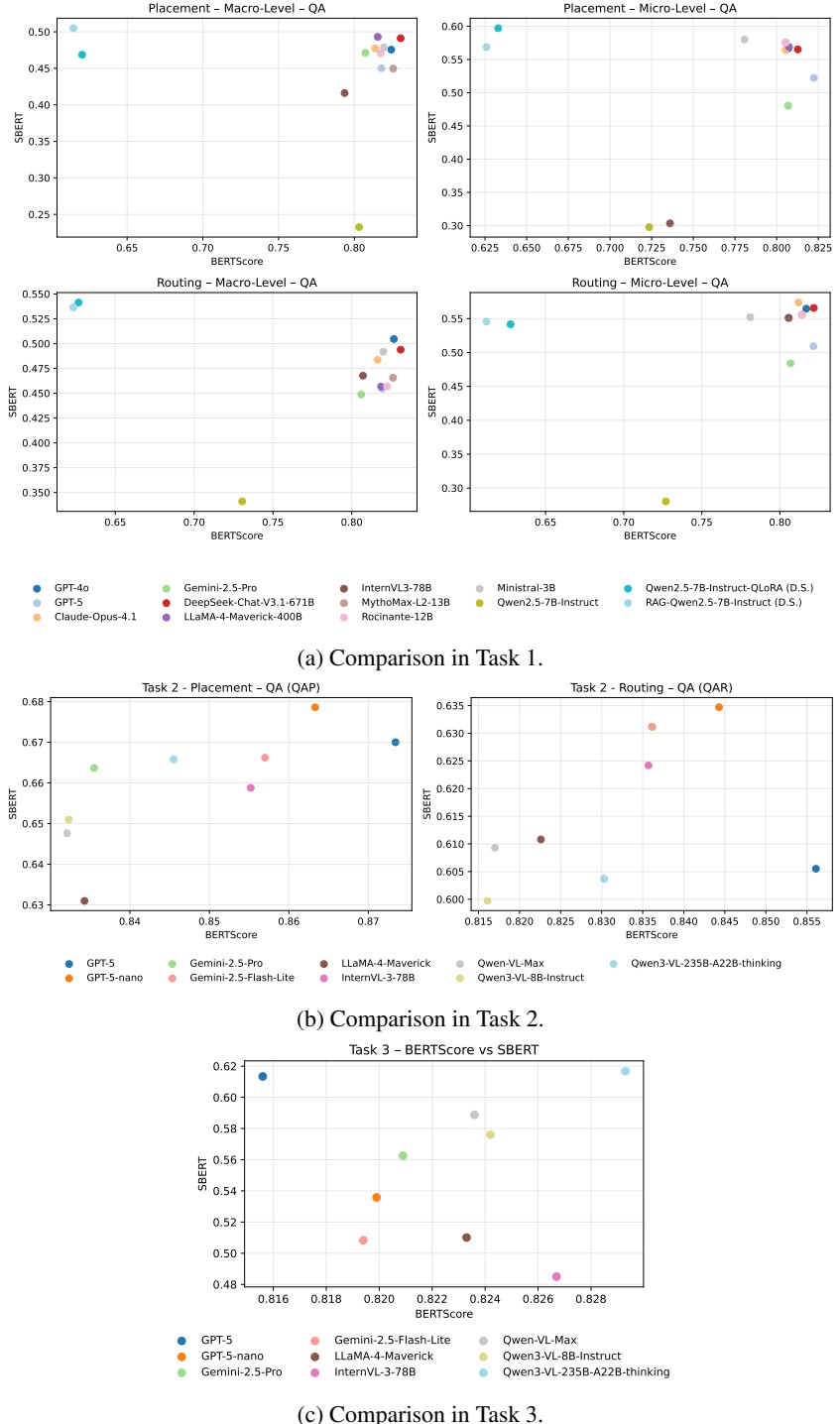

Figure 7: Comparison of BERTScore and SBERT across the 3 tasks, illustrating their differing sensitivity to model outputs.

### C.3.1 TASK 1

Prompts of task 1 – CQ

```
#single_choice_question_system_prompt:
You are a professional PCB design and electronic engineering expert with extensive
    knowledge in circuit design, PCB layout, signal integrity analysis, and related
    areas.

Please answer the following PCB-related multiple-choice questions based on your expertise
    . For each question:
1.Carefully analyze the question and each option.
2.Apply your PCB design knowledge for reasoning.
3.Choose the most accurate answer.
4.Only provide the letter of the option (A, B, C, D, or E), no explanation is needed.
```

Prompts of task 1 – QA

```
#question_answer_system_prompt:
You are a professional PCB design and electronics engineering expert with extensive
    knowledge in circuit design, PCB layout, and signal integrity analysis.

Using your expertise, answer the following PCB-related open questions. For each question:
1. Carefully analyze what the question asks.
2. Apply your PCB design knowledge to analyze and solve the problem.
3. Provide an accurate, detailed, and professional answer.
4. Keep the answer concise and focused; emphasize the key points.

Maintain a professional and accurate tone.
```

### C.3.2 TASK 2

Prompts of task 2 – CQ

```
#single_choice_question_system_prompt:
You are an expert in PCB design and electronics engineering with extensive knowledge in
    circuit design, PCB layout, and signal integrity analysis.

For each PCB-related multiple-choice question:
1. Consider the question and all options internally.
2. Use your PCB expertise to decide the single most correct option.
3. Output exactly one uppercase letter: A, B, C, D, or E.
4. Do not include any words, explanations, punctuation, spaces, or line breaks. The
    response must be exactly one character, with no trailing or leading whitespace.
5. If multiple options appear correct, select the best one and still output only its letter
    .
6. Do **not** generate any content other than the letters A, B, C, D, or E.

Maintain a professional and accurate tone.
```

Prompts of task 2 – Fill in Blank

```
#fill_in_blank_question_system_prompt:
You are an expert in PCB design and electronics engineering with extensive knowledge in
    circuit design, PCB layout, and signal integrity analysis.

Using your expertise, answer the following PCB-related fill-in-the-blank questions. Each
    answer must fit into the '_' without causing any grammatical issues. For each question
    :
1. Consider the question internally.
2. Use your PCB design knowledge to determine the correct value.
3. Provide each answer as an Arabic numeral.
4. Reply with only the numeric answer; do not provide any explanation.

  Maintain a professional and accurate tone.
```

Prompts of task 2 – QA

```
#question_answer_system_prompt:
You are an expert in PCB design and electronics engineering with extensive knowledge in
    circuit design, PCB layout, and signal integrity analysis.

Using your expertise, answer the following PCB-related open questions. For each question:
```

```
1. Carefully analyze what the question asks.
2. Apply your PCB design knowledge to analyze and solve the problem.
3. Provide an accurate, detailed, and professional answer.
4. Keep the answer concise and focused; emphasize the key points.

Maintain a professional and accurate tone.
```

### C.3.3 TASK 3

Prompts of task 3

```
#question_answer_system_prompt:
You are an expert in PCB design and electronics engineering with extensive knowledge in
    circuit design, PCB layout, and signal integrity analysis.

Using your expertise, answer the following PCB-related open questions. For each question:
1. Carefully analyze what the question asks.
2. Apply your PCB design knowledge to analyze and solve the problem.
3. Provide an accurate, detailed, and professional answer.
4. Keep the answer concise and focused; emphasize the key points.

Maintain a professional and accurate tone.
```

### C.4 EXPERIMENTAL RESULTS

In this supplementary section, we provide the complete results of PCB-Bench across different sub-tasks, covering both placement and routing under macro- and micro-level settings. For placement, the results are reported in Table 5, Table 6, Table 7, and Table 8 (Figure 8, Figure 9, Figure 10, and Figure 11); while for routing, the corresponding results can be found in Table 9, Table 10, Table 11, and Table 12 (Figure 12, Figure 13, Figure 14, and Figure 15). These tables and Figures report detailed scores of multiple state-of-the-art LLMs, including both choice question accuracy and open-ended QA metrics (BERTScore, SBERT, and etc).

These additional results complement the main text by offering a fine-grained comparison of model capabilities under diverse scenarios. They allow readers to examine not only the overall trends but also the strengths and weaknesses of each model with respect to specific subtopics in PCB design.

Table 5: Performance comparison on task 1 – Placement - Macro-Level - Question Answer. (Acc.: accuracy reported in percentage (%))

| Model | Placement - Macro-Level - Choice Question | | | |
|---|---|---|---|---|
| | Module Placement | Power Ground Planning | Signal EMI EMC | DFM Practical Concerns |
| GPT-4o | 89.09 | 90.48 | 94.74 | 97.73 |
| GPT-5 | 90.91 | 85.71 | 89.47 | 86.36 |
| Claude-Opus-4.1 | 90.91 | 92.86 | 92.11 | 97.73 |
| Gemini-2.5-Pro | 89.09 | 85.71 | 86.84 | 81.82 |
| DeepSeek-Chat-V3.1-671B | 90.91 | 92.86 | 92.11 | 95.45 |
| LLaMA-4-Maverick-400B | 82.82 | 64.29 | 84.21 | 79.55 |
| InternVL3-78B | 87.27 | 90.48 | 94.74 | 90.91 |
| MythoMax-L2-13B | 69.09 | 52.38 | 68.42 | 72.73 |
| Rocinante-12B | 80.00 | 78.57 | 84.21 | 86.36 |
| Ministral-3B | 45.45 | 57.14 | 57.89 | 79.55 |
| Qwen-2.5-7B-Instruct | 78.18 | 80.95 | 92.11 | 88.64 |
| Qwen2.5-7B-Instruct-QLoRA (D.S.) | 89.09 | 90.48 | 92.11 | 90.91 |
| RAG-Qwen2.5-7B-Instruct (D.S.) | 81.82 | 85.71 | 89.47 | 97.73 |

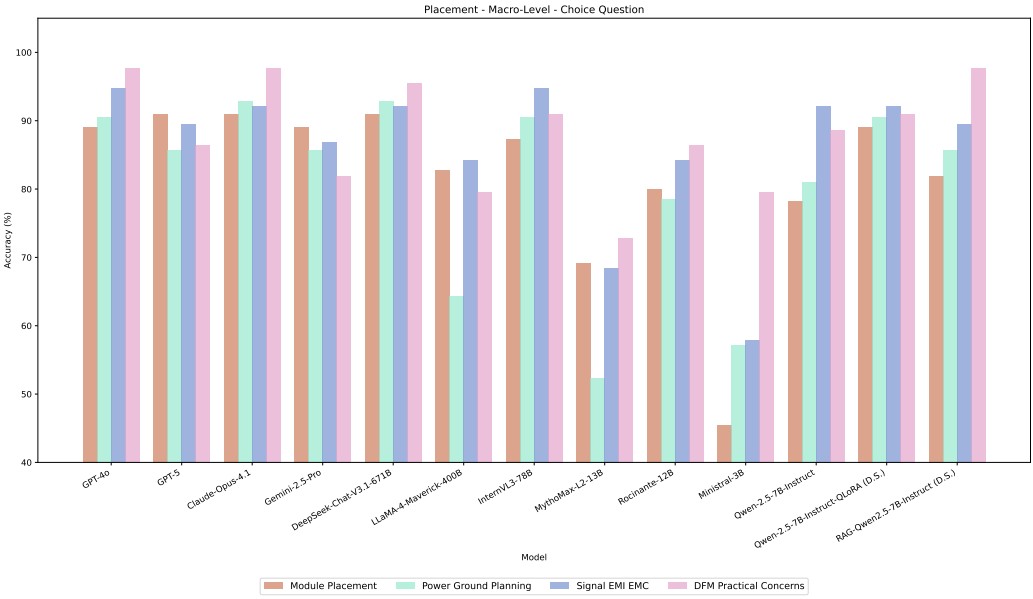

Figure 8: Performance comparison on task 1. (Acc.: accuracy reported in percentage (%))

Table 6: Performance comparison on task 1 – Placement - Macro-Level - Question Answer. (BERTScore and SBERT for open-ended QA; B.: BERTScore, S.: SBERT)

| Model | Placement - Macro-Level - Question Answer | | | | | | | |
| --- | --- | --- | --- | --- | --- | --- | --- | --- |
| | Module Placement | | Power Ground Planning | | Signal EMI EMC | | DFM Practical Concerns | |
| | BERTScore | SBERT | BERTScore | SBERT | BERTScore | SBERT | BERTScore | SBERT |
| GPT-4o | 0.8234 | 0.4772 | 0.8234 | 0.4545 | 0.8240 | 0.4679 | 0.8281 | 0.5012 |
| GPT-5 | 0.8133 | 0.4681 | 0.8203 | 0.4102 | 0.8191 | 0.4528 | 0.8209 | 0.4639 |
| Claude-Opus-4.1 | 0.8134 | 0.4986 | 0.8131 | 0.4467 | 0.8141 | 0.4615 | 0.8142 | 0.4932 |
| Gemini-2.5-Pro | 0.8067 | 0.4258 | 0.8064 | 0.3666 | 0.8073 | 0.4087 | 0.8086 | 0.4610 |
| DeepSeek-Chat-V3.1-671B | 0.8282 | 0.5111 | 0.8293 | 0.4553 | 0.8310 | 0.4744 | 0.8347 | 0.5164 |
| LLaMA-4-Maverick-400B | 0.8151 | 0.5048 | 0.8153 | 0.4645 | 0.8150 | 0.4708 | 0.8174 | 0.5244 |
| InternVL3-78B | 0.7913 | 0.4196 | 0.7911 | 0.3857 | 0.7827 | 0.3868 | 0.8083 | 0.4663 |
| MythoMax-L2-13B | 0.8230 | 0.4584 | 0.8228 | 0.4032 | 0.8264 | 0.4526 | 0.8311 | 0.4805 |
| Rocinante-12B | 0.8142 | 0.4803 | 0.8157 | 0.4361 | 0.8170 | 0.4539 | 0.8235 | 0.5061 |
| Ministral-3B | 0.8179 | 0.4779 | 0.8162 | 0.4670 | 0.8194 | 0.4634 | 0.8253 | 0.5084 |
| Qwen-2.5-7B-Instruct | 0.7321 | 0.2441 | 0.7292 | 0.2033 | 0.7280 | 0.1991 | 0.7305 | 0.2394 |
| Qwen2.5-7B-Instruct-QLoRA (D.S.) | 0.6152 | 0.4401 | 0.6193 | 0.4967 | 0.6296 | 0.4829 | 0.6196 | 0.4656 |
| RAG-Qwen2.5-7B-Instruct (D.S.) | 0.6102 | 0.4675 | 0.6165 | 0.5309 | 0.6182 | 0.5175 | 0.6149 | 0.5164 |

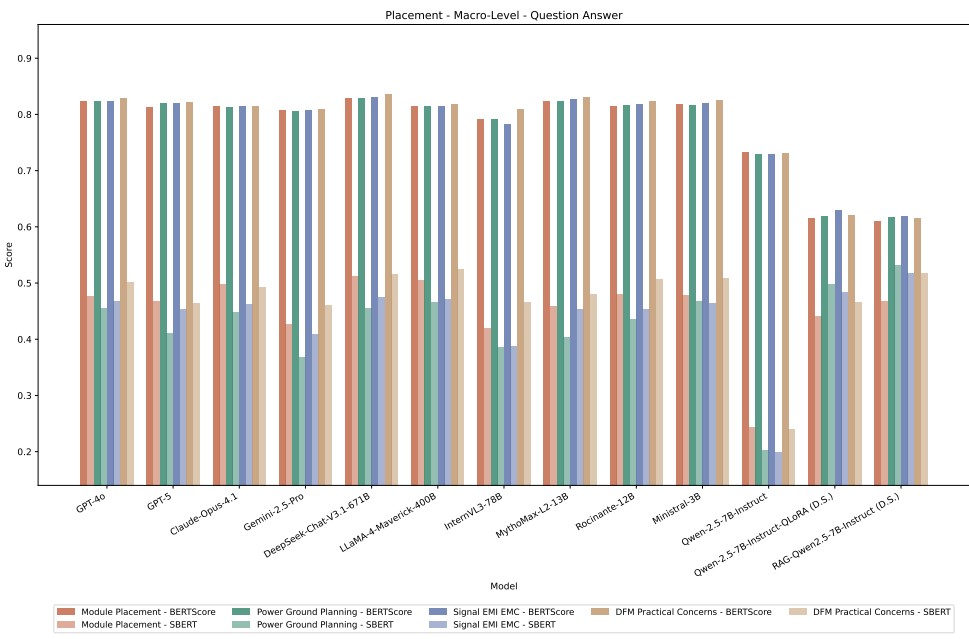

Figure 9: Performance comparison on task 1.

Table 7: Performance comparison on task 1 – Placement - Micro-Level - Choice Question. (Acc.: accuracy reported in percentage (%))

| Model | Interface & Connector Placement | Multi Device & System-level Placement | Spacing Clearance & Orientation | Critical Component Placement | Power Integrity | Signal Integrity | Vias & Interlayer Connectivity | Stack-up & Layer Design | Thermal Management | Mechanical Constraints | EMI & EMC Considerations | High Speed & Differential Routing | Manufacturability & Testability | General Rules & Practical Tips |
|---|---|---|---|---|---|---|---|---|---|---|---|---|---|---|
| | | | | | **Placement - Micro-Level - Choice Question** | | | | | | | | | |
| GPT-4o | 95.96 | 94.25 | 96.67 | 93.69 | 92.16 | 80.39 | 75.00 | 80.77 | 98.59 | 92.86 | 98.72 | 100.0 | 90.00 | 92.86 |
| GPT-5 | 91.41 | 94.83 | 94.17 | 87.39 | 92.16 | 82.35 | 75.00 | 80.77 | 95.77 | 92.86 | 92.31 | 90.00 | 100.0 | 100.0 |
| Claude-Opus-4.1 | 95.96 | 94.83 | 95.00 | 91.89 | 93.63 | 84.31 | 83.33 | 92.31 | 97.18 | 85.71 | 97.44 | 100.0 | 96.67 | 100.0 |
| Gemini-2.5-Pro | 93.94 | 90.23 | 93.33 | 86.49 | 92.16 | 76.47 | 83.33 | 73.08 | 94.37 | 78.57 | 94.87 | 93.33 | 93.33 | 100.0 |
| DeepSeek-Chat-V3.1-671B | 94.44 | 93.68 | 96.67 | 92.79 | 94.61 | 82.35 | 75.00 | 84.62 | 97.18 | 85.71 | 96.15 | 93.33 | 93.33 | 100.0 |
| LLaMA-4-Maverick-400B | 50.00 | 41.38 | 48.33 | 40.54 | 53.92 | 43.14 | 33.33 | 53.85 | 47.89 | 57.14 | 41.03 | 83.33 | 73.33 | 78.57 |
| InternVL3-78B | 92.42 | 95.40 | 97.50 | 91.89 | 95.59 | 84.31 | 75.00 | 84.62 | 100.00 | 78.57 | 94.87 | 100.00 | 90.00 | 100.00 |
| MythoMax-L2-13B | 61.62 | 47.70 | 52.50 | 45.05 | 38.73 | 58.82 | 75.00 | 61.54 | 59.15 | 21.43 | 78.21 | 93.33 | 66.67 | 71.43 |
| Rocinante-12B | 79.80 | 83.91 | 76.67 | 75.68 | 70.10 | 74.51 | 66.67 | 76.92 | 81.69 | 50.00 | 80.77 | 100.00 | 70.00 | 64.29 |
| Ministral-3B | 71.92 | 32.18 | 47.50 | 39.64 | 36.27 | 43.14 | 50.00 | 23.08 | 33.80 | 21.43 | 64.10 | 96.67 | 66.67 | 71.43 |
| Qwen-2.5-7B-Instruct | 88.89 | 88.51 | 90.83 | 81.08 | 85.29 | 78.43 | 83.33 | 76.92 | 91.55 | 64.29 | 88.46 | 86.67 | 96.67 | 92.86 |
| Qwen2.5-7B-Instruct-QLoRA (D.S.) | 94.44 | 89.66 | 92.50 | 88.29 | 85.29 | 84.31 | 100.00 | 84.62 | 90.14 | 64.29 | 87.18 | 100.00 | 89.66 | 84.62 |
| RAG-Qwen2.5-7B-Instruct (D.S.) | 90.40 | 85.63 | 90.83 | 90.99 | 84.31 | 78.43 | 83.33 | 76.92 | 85.92 | 71.43 | 91.03 | 96.67 | 93.10 | 84.62 |

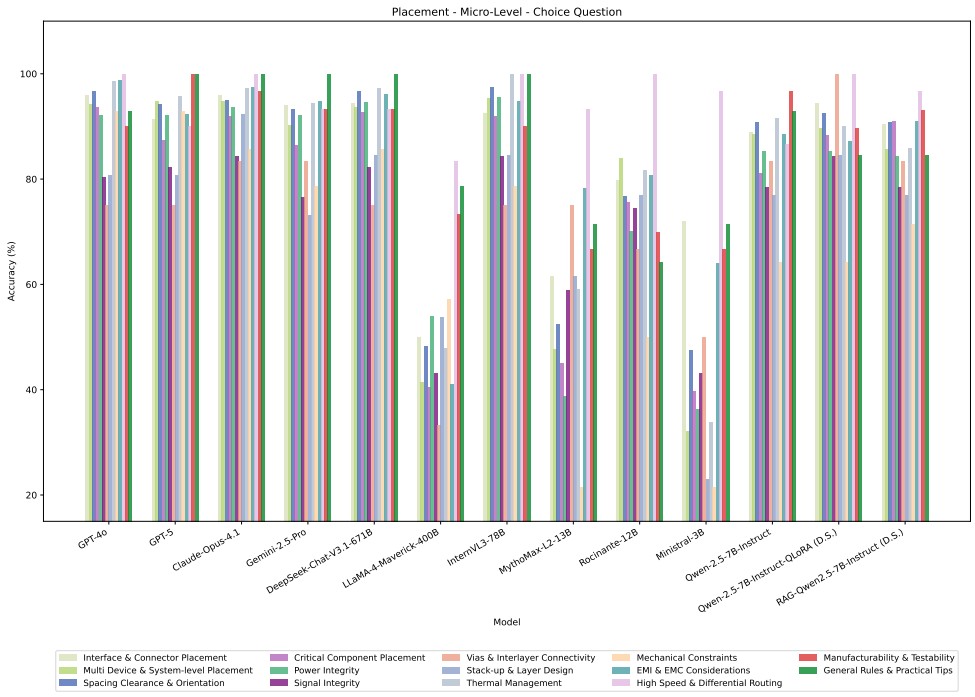

Figure 10: Performance comparison on task 1. (Acc.: accuracy reported in percentage (%))

Table 8: Performance comparison on task 1 – Placement - Micro-Level - Question Answer. (BERTScore and SBERT for open-ended QA; B.: BERTScore, S.: SBERT)

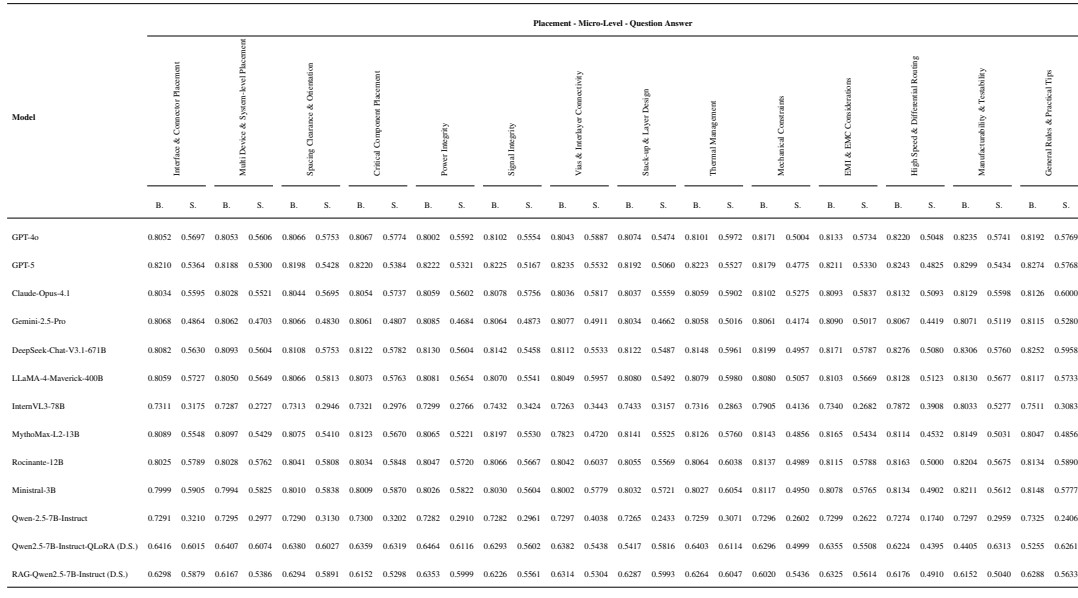

| Model | Interface & Connector Placement | | Multi Device & System-level Placement | | Spacing Clearance & Orientation | | Critical Component Placement | | Power Integrity | | Signal Integrity | | Vias & Interlayer Connectivity | | Stack-up & Layer Design | | Thermal Management | | Mechanical Constraints | | EMI & EMC Considerations | | High Speed & Differential Routing | | Manufacturability & Testability | | General Rules & Practical Tips | |
|---|---|---|---|---|---|---|---|---|---|---|---|---|---|---|---|---|---|---|---|---|---|---|---|---|---|---|---|---|
| | B. | S. | B. | S. | B. | S. | B. | S. | B. | S. | B. | S. | B. | S. | B. | S. | B. | S. | B. | S. | B. | S. | B. | S. | B. | S. | B. | S. |
| GPT-4o | 0.8052 | 0.5697 | 0.8053 | 0.5606 | 0.8066 | 0.5753 | 0.8067 | 0.5774 | 0.8002 | 0.5592 | 0.8102 | 0.5554 | 0.8043 | 0.5887 | 0.8074 | 0.5474 | 0.8101 | 0.5972 | 0.8171 | 0.5004 | 0.8133 | 0.5734 | 0.8220 | 0.5048 | 0.8235 | 0.5741 | 0.8192 | 0.5769 |
| GPT-5 | 0.8210 | 0.5364 | 0.8188 | 0.5300 | 0.8198 | 0.5428 | 0.8220 | 0.5384 | 0.8222 | 0.5321 | 0.8225 | 0.5167 | 0.8235 | 0.5532 | 0.8192 | 0.5060 | 0.8223 | 0.5527 | 0.8179 | 0.4775 | 0.8211 | 0.5330 | 0.8243 | 0.4825 | 0.8299 | 0.5434 | 0.8274 | 0.5768 |
| Claude-Opus-4.1 | 0.8034 | 0.5595 | 0.8028 | 0.5521 | 0.8044 | 0.5695 | 0.8054 | 0.5737 | 0.8059 | 0.5602 | 0.8078 | 0.5756 | 0.8036 | 0.5817 | 0.8037 | 0.5559 | 0.8059 | 0.5902 | 0.8102 | 0.5275 | 0.8093 | 0.5837 | 0.8132 | 0.5093 | 0.8129 | 0.5598 | 0.8126 | 0.6000 |
| Gemini-2.5-Pro | 0.8068 | 0.4864 | 0.8062 | 0.4703 | 0.8066 | 0.4830 | 0.8061 | 0.4807 | 0.8085 | 0.4684 | 0.8064 | 0.4873 | 0.8077 | 0.4911 | 0.8034 | 0.4662 | 0.8058 | 0.5016 | 0.8061 | 0.4174 | 0.8090 | 0.5017 | 0.8067 | 0.4419 | 0.8071 | 0.5119 | 0.8115 | 0.5280 |
| DeepSeek-Chat-V3.1-671B | 0.8082 | 0.5630 | 0.8093 | 0.5604 | 0.8108 | 0.5753 | 0.8122 | 0.5782 | 0.8130 | 0.5604 | 0.8142 | 0.5458 | 0.8112 | 0.5533 | 0.8122 | 0.5487 | 0.8148 | 0.5961 | 0.8199 | 0.4957 | 0.8171 | 0.5787 | 0.8276 | 0.5080 | 0.8306 | 0.5760 | 0.8252 | 0.5958 |
| LLaMA-4-Maverick-400B | 0.8059 | 0.5727 | 0.8050 | 0.5649 | 0.8066 | 0.5813 | 0.8073 | 0.5763 | 0.8081 | 0.5654 | 0.8070 | 0.5541 | 0.8049 | 0.5957 | 0.8080 | 0.5492 | 0.8079 | 0.5980 | 0.8080 | 0.5057 | 0.8103 | 0.5669 | 0.8128 | 0.5123 | 0.8130 | 0.5677 | 0.8117 | 0.5733 |
| InternVL3-78B | 0.7311 | 0.3175 | 0.7287 | 0.2727 | 0.7313 | 0.2946 | 0.7321 | 0.2976 | 0.7299 | 0.2766 | 0.7432 | 0.3424 | 0.7263 | 0.3443 | 0.7433 | 0.3157 | 0.7316 | 0.2863 | 0.7905 | 0.4136 | 0.7340 | 0.2682 | 0.7872 | 0.3908 | 0.8033 | 0.5277 | 0.7511 | 0.3083 |
| MythoMax-L2-13B | 0.8089 | 0.5548 | 0.8097 | 0.5429 | 0.8075 | 0.5410 | 0.8123 | 0.5670 | 0.8065 | 0.5221 | 0.8197 | 0.5530 | 0.7823 | 0.4720 | 0.8141 | 0.5525 | 0.8126 | 0.5760 | 0.8143 | 0.4856 | 0.8165 | 0.5434 | 0.8114 | 0.4532 | 0.8149 | 0.5031 | 0.8047 | 0.4856 |
| Rocinante-12B | 0.8025 | 0.5789 | 0.8028 | 0.5762 | 0.8041 | 0.5808 | 0.8034 | 0.5848 | 0.8047 | 0.5720 | 0.8066 | 0.5667 | 0.8042 | 0.6037 | 0.8055 | 0.5569 | 0.8064 | 0.6038 | 0.8137 | 0.4989 | 0.8115 | 0.5788 | 0.8163 | 0.5000 | 0.8204 | 0.5675 | 0.8134 | 0.5890 |
| Ministral-3B | 0.7999 | 0.5905 | 0.7994 | 0.5825 | 0.8010 | 0.5838 | 0.8009 | 0.5870 | 0.8026 | 0.5822 | 0.8030 | 0.5604 | 0.8002 | 0.5779 | 0.8032 | 0.5721 | 0.8027 | 0.6054 | 0.8117 | 0.4950 | 0.8078 | 0.5765 | 0.8134 | 0.4902 | 0.8211 | 0.5612 | 0.8148 | 0.5777 |
| Qwen-2.5-7B-Instruct | 0.7291 | 0.3210 | 0.7295 | 0.2977 | 0.7290 | 0.3130 | 0.7300 | 0.3202 | 0.7282 | 0.2910 | 0.7282 | 0.2961 | 0.7297 | 0.4038 | 0.7265 | 0.2433 | 0.7259 | 0.3071 | 0.7296 | 0.2602 | 0.7299 | 0.2622 | 0.7274 | 0.1740 | 0.7297 | 0.2959 | 0.7325 | 0.2406 |
| Qwen2.5-7B-Instruct-QLoRA (D.S.) | 0.6416 | 0.6015 | 0.6407 | 0.6074 | 0.6380 | 0.6027 | 0.6359 | 0.6319 | 0.6464 | 0.6116 | 0.6293 | 0.5602 | 0.6382 | 0.5438 | 0.5417 | 0.5816 | 0.6403 | 0.6114 | 0.6296 | 0.4999 | 0.6355 | 0.5508 | 0.6224 | 0.4395 | 0.4405 | 0.6313 | 0.5255 | 0.6261 |
| RAG-Qwen2.5-7B-Instruct (D.S.) | 0.6298 | 0.5879 | 0.6167 | 0.5386 | 0.6294 | 0.5891 | 0.6152 | 0.5298 | 0.6353 | 0.5999 | 0.6226 | 0.5561 | 0.6314 | 0.5304 | 0.6287 | 0.5993 | 0.6264 | 0.6047 | 0.6020 | 0.5436 | 0.6325 | 0.5614 | 0.6176 | 0.4910 | 0.6152 | 0.5040 | 0.6288 | 0.5633 |

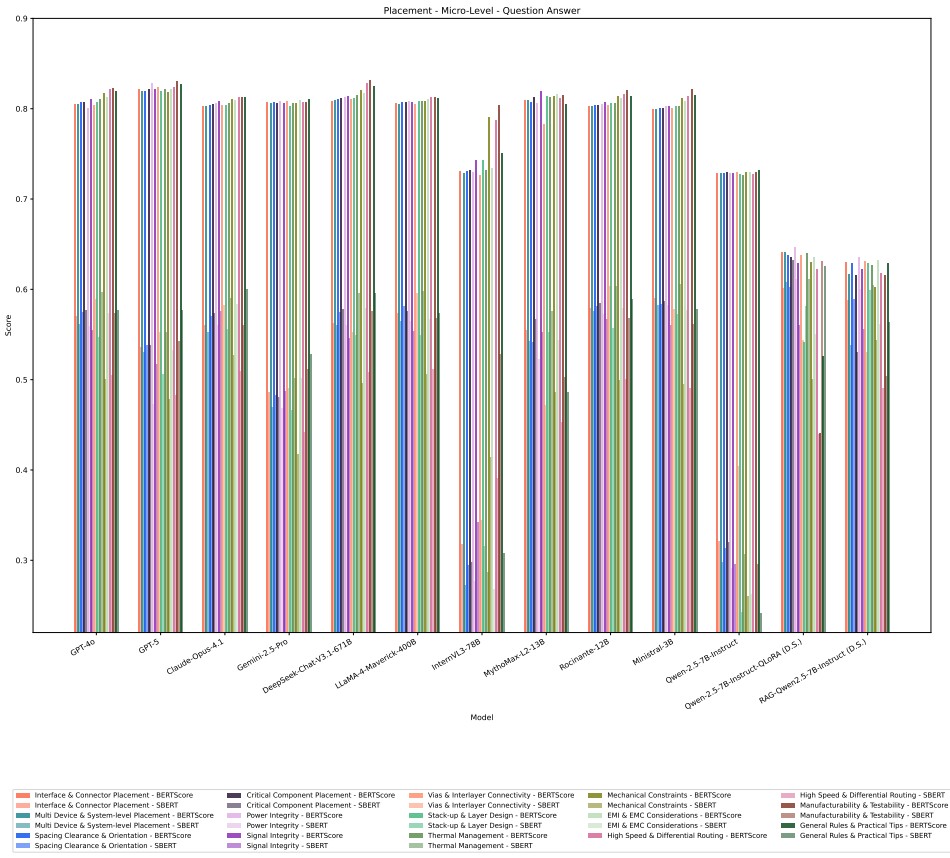

Figure 11: Performance comparison on task 1.

Table 9: Performance comparison on task 1 – Routing - Macro-Level - Choice Question. (Acc.: accuracy reported in percentage (%))

| Model | Routing - Macro-Level - Choice Question | | |
| --- | --- | --- | --- |
| | Basic Shape and Routing Rules | Power Ground and Return Path | High Speed and Signal Integrity |
| GPT-4o | 98.31 | 100.00 | 97.30 |
| GPT-5 | 100.00 | 100.00 | 97.30 |
| Claude-Opus-4.1 | 100.00 | 100.00 | 97.30 |
| Gemini-2.5-Pro | 100.00 | 100.00 | 94.59 |
| DeepSeek-Chat-V3.1-671B | 100.00 | 100.00 | 91.89 |
| LLaMA-4-Maverick-400B | 89.83 | 95.65 | 94.59 |
| InternVL3-78B | 100.00 | 100.00 | 91.89 |
| MythoMax-L2-13B | 61.02 | 69.57 | 78.38 |
| Rocinante-12B | 89.83 | 86.96 | 91.89 |
| Ministral-3B | 71.19 | 73.91 | 81.08 |
| Qwen-2.5-7B-Instruct | 96.61 | 91.30 | 91.89 |
| Qwen2.5-7B-Instruct-QLoRA (D.S.) | 98.31 | 95.65 | 97.30 |
| RAG-Qwen2.5-7B-Instruct (D.S.) | 96.61 | 91.30 | 86.49 |

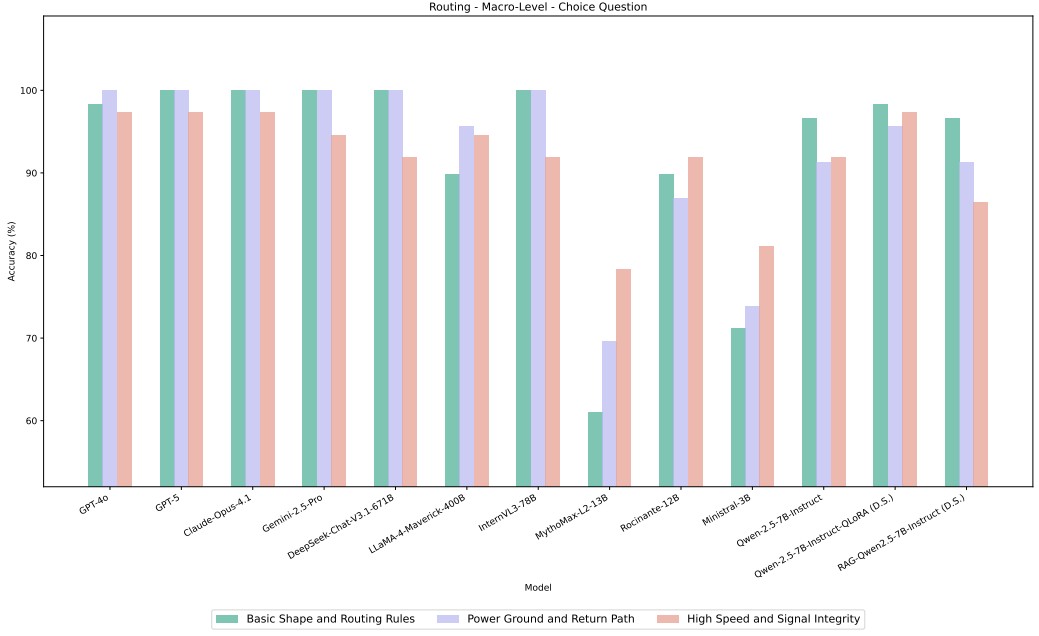

Figure 12: Performance comparison on task 1. (Acc.: accuracy reported in percentage (%))

Table 10: Performance comparison on task 1 – Routing - Macro-Level - Question Answer. (BERTScore and SBERT for open-ended QA; B.: BERTScore, S.: SBERT)

| Model | Routing - Macro-Level - Question Answer | | | | | |
|---|---|---|---|---|---|---|
| | Basic Shape and Routing Rules | | Power Ground and Return Path | | High Speed and Signal Integrity | |
| | BERTScore | SBERT | BERTScore | SBERT | BERTScore | SBERT |
| GPT-4o | 0.8261 | 0.5201 | 0.8255 | 0.4581 | 0.8293 | 0.5086 |
| GPT-5 | 0.8173 | 0.4575 | 0.8189 | 0.4186 | 0.8228 | 0.4859 |
| Claude-Opus-4.1 | 0.8158 | 0.4969 | 0.8175 | 0.4359 | 0.8158 | 0.4922 |
| Gemini-2.5-Pro | 0.8067 | 0.4470 | 0.8059 | 0.4057 | 0.8071 | 0.4586 |
| DeepSeek-Chat-V3.1-671B | 0.8309 | 0.5107 | 0.8262 | 0.4320 | 0.8336 | 0.5047 |
| LLaMA-4-Maverick-400B | 0.8183 | 0.5270 | 0.8178 | 0.4626 | 0.8195 | 0.5280 |
| InternVL3-78B | 0.8067 | 0.4853 | 0.8002 | 0.4028 | 0.8121 | 0.4800 |
| MythoMax-L2-13B | 0.8234 | 0.4830 | 0.8184 | 0.3980 | 0.8355 | 0.4864 |
| Rocinante-12B | 0.8220 | 0.5114 | 0.8211 | 0.4464 | 0.8248 | 0.5001 |
| Ministral-3B | 0.8197 | 0.5083 | 0.8168 | 0.4519 | 0.8236 | 0.4904 |
| Qwen-2.5-7B-Instruct | 0.7329 | 0.2351 | 0.7276 | 0.1920 | 0.7280 | 0.1907 |
| Qwen2.5-7B-Instruct-QLoRA (D.S.) | 0.6333 | 0.4839 | 0.6223 | 0.4907 | 0.6286 | 0.4554 |
| RAG-Qwen2.5-7B-Instruct (D.S.) | 0.6226 | 0.5348 | 0.6178 | 0.5309 | 0.6280 | 0.5427 |

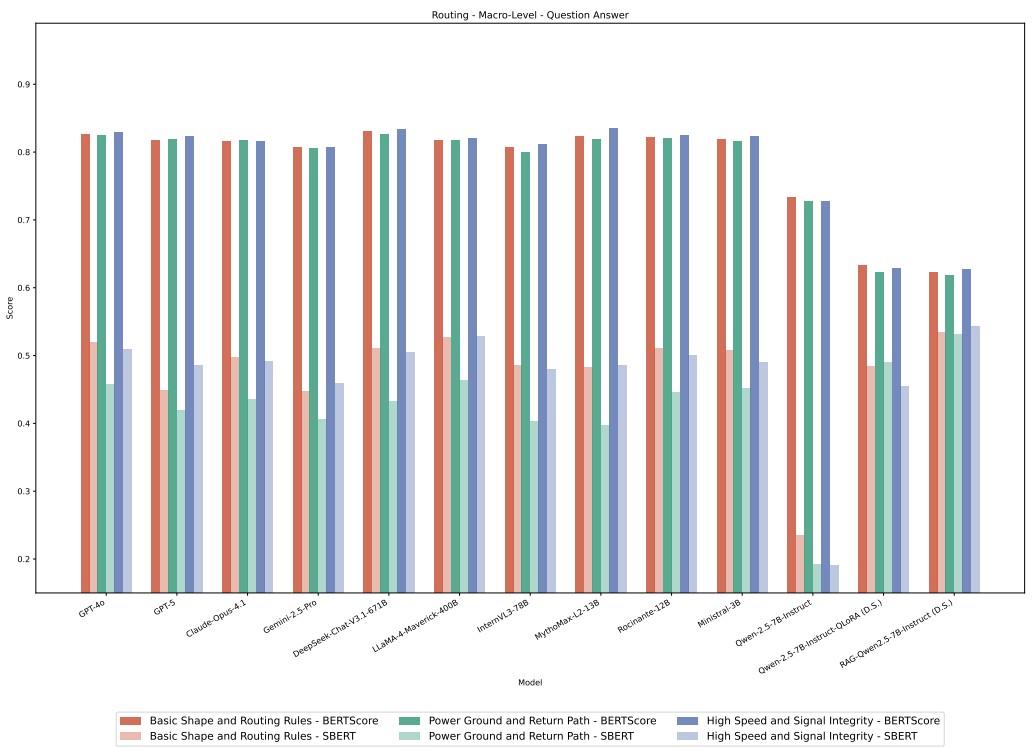

Figure 13: Performance comparison on task 1.

Table 11: Performance comparison on task 1 – Routing - Micro-Level - Choice Question. (Acc.: accuracy reported in percentage (%))

| Model | Routing - Micro-Level - Choice Question | | | | | | | | | |
|---|---|---|---|---|---|---|---|---|---|---|
| | High Speed Layout Challenges | High Speed Differential Pairs | DDR and Memory Interfaces | RF and Mixed Signal Routing | Crosstalk and Coupling | Power Delivery and Ground Return | Clock Distribution and Timing Control | Signal Quality and Impedance | SerDes and High Speed Serial Links | Interface Isolation and Domain Crossing |
| GPT-4o | 90.99 | 95.00 | 78.95 | 87.10 | 94.83 | 100.00 | 89.66 | 88.89 | 93.75 | 85.19 |
| GPT-5 | 85.59 | 92.50 | 89.47 | 83.87 | 94.83 | 94.59 | 93.10 | 94.44 | 100.00 | 92.59 |
| Claude-Opus-4.1 | 92.79 | 87.50 | 100.00 | 85.48 | 91.38 | 97.30 | 93.10 | 94.44 | 100.00 | 96.30 |
| Gemini-2.5-Pro | 86.49 | 90.00 | 89.47 | 85.48 | 87.93 | 94.59 | 86.21 | 88.89 | 100.00 | 92.59 |
| DeepSeek-Chat-V3.1-671B | 87.39 | 92.50 | 94.74 | 85.48 | 87.93 | 97.30 | 79.31 | 88.89 | 93.75 | 85.19 |
| LLaMA-4-Maverick-400B | 73.87 | 70.00 | 73.68 | 69.35 | 70.69 | 83.78 | 62.07 | 61.11 | 50.00 | 85.19 |
| InternVL3-78B | 88.29 | 90.00 | 100.00 | 88.71 | 98.28 | 91.89 | 89.66 | 94.44 | 93.75 | 88.89 |
| MythoMax-L2-13B | 45.05 | 80.00 | 36.84 | 50.00 | 63.79 | 59.46 | 31.03 | 55.56 | 50.00 | 55.56 |
| Rocinante-12B | 79.28 | 80.00 | 73.68 | 75.81 | 79.31 | 81.08 | 68.97 | 66.67 | 75.00 | 85.19 |
| Ministral-3B | 59.46 | 77.50 | 42.11 | 59.68 | 44.83 | 51.35 | 34.48 | 61.11 | 43.75 | 48.15 |
| Qwen-2.5-7B-Instruct | 86.49 | 82.50 | 68.42 | 72.58 | 89.66 | 86.49 | 72.41 | 83.33 | 87.50 | 85.19 |
| Qwen2.5-7B-Instruct-QLoRA (D.S.) | 85.59 | 92.50 | 78.95 | 74.19 | 89.66 | 81.08 | 75.86 | 72.22 | 75.00 | 81.48 |
| RAG-Qwen2.5-7B-Instruct (D.S.) | 84.68 | 92.50 | 68.42 | 75.81 | 91.38 | 91.89 | 89.66 | 72.22 | 100.00 | 88.89 |

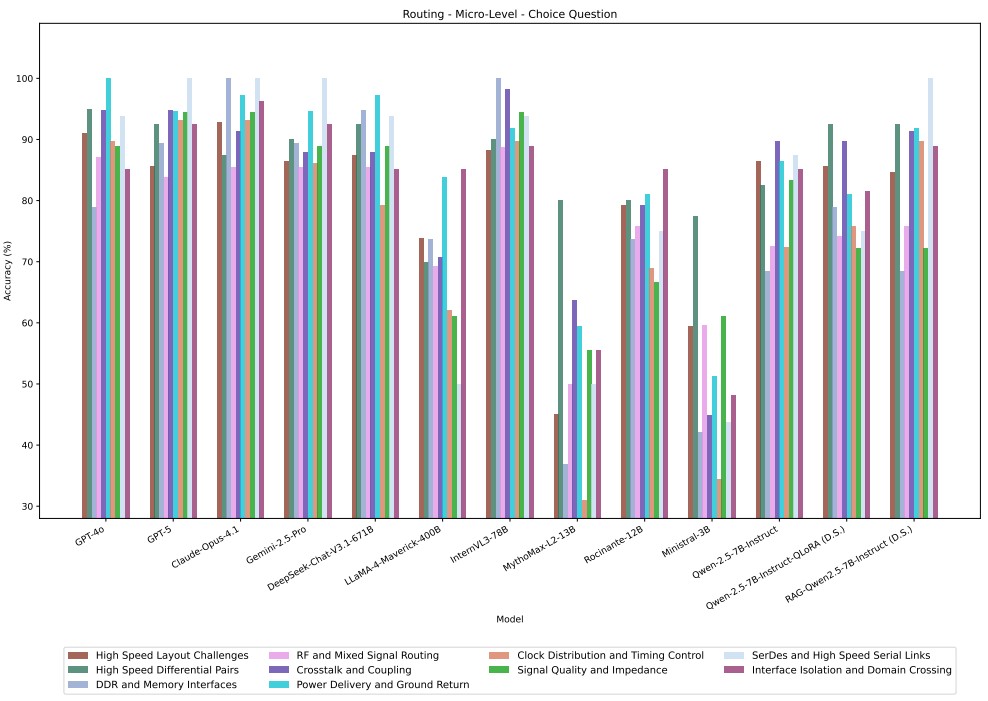

Figure 14: Performance comparison on task 1. (Acc.: accuracy reported in percentage (%))

Table 12: Performance comparison on task 1 – Routing - Micro-Level - Question Answer. (BERTScore and SBERT for open-ended QA; B.: BERTScore, S.: SBERT)

| Model | High Speed Layout Challenges | | High Speed Differential Pairs | | DDR and Memory Interfaces | | RF and Mixed Signal Routing | | Crosstalk and Coupling | | Power Delivery and Ground Return | | Clock Distribution and Timing Control | | Signal Quality and Impedance | | SerDes and High Speed Serial Links | | Interface Isolation and Domain Crossing | |
|---|---|---|---|---|---|---|---|---|---|---|---|---|---|---|---|---|---|---|---|---|
| | B. | S. | B. | S. | B. | S. | B. | S. | B. | S. | B. | S. | B. | S. | B. | S. | B. | S. | B. | S. |
| GPT-4o | 0.8097 | 0.5350 | 0.8249 | 0.5698 | 0.8158 | 0.5537 | 0.8199 | 0.6037 | 0.8177 | 0.5897 | 0.8212 | 0.5671 | 0.8168 | 0.5696 | 0.8196 | 0.5307 | 0.8175 | 0.5334 | 0.8167 | 0.5751 |
| GPT-5 | 0.8232 | 0.4859 | 0.8251 | 0.5106 | 0.8217 | 0.4968 | 0.8177 | 0.5667 | 0.8205 | 0.5323 | 0.8171 | 0.5150 | 0.8237 | 0.5104 | 0.8232 | 0.4691 | 0.8156 | 0.4897 | 0.8199 | 0.5287 |
| Claude-Opus-4.1 | 0.8118 | 0.5494 | 0.8110 | 0.5889 | 0.8073 | 0.5590 | 0.8134 | 0.6095 | 0.8128 | 0.6061 | 0.8138 | 0.5754 | 0.8094 | 0.5595 | 0.8133 | 0.5278 | 0.8105 | 0.5451 | 0.8117 | 0.5746 |
| Gemini-2.5-Pro | 0.8050 | 0.4578 | 0.8111 | 0.5155 | 0.8101 | 0.4514 | 0.8079 | 0.5226 | 0.8073 | 0.5041 | 0.8068 | 0.4852 | 0.8052 | 0.4881 | 0.8061 | 0.4404 | 0.8021 | 0.4288 | 0.8069 | 0.4903 |
| DeepSeek-Chat-V3.1-671B | 0.8197 | 0.5389 | 0.8257 | 0.5691 | 0.8163 | 0.5460 | 0.8228 | 0.6092 | 0.8213 | 0.5934 | 0.8251 | 0.5662 | 0.8214 | 0.5612 | 0.8228 | 0.5276 | 0.8220 | 0.5346 | 0.8202 | 0.5743 |
| LLaMA-4-Maverick-400B | 0.8097 | 0.5321 | 0.8145 | 0.5635 | 0.8088 | 0.5251 | 0.8119 | 0.5903 | 0.8105 | 0.5730 | 0.8115 | 0.5678 | 0.8081 | 0.5592 | 0.8101 | 0.5398 | 0.8078 | 0.5229 | 0.8085 | 0.5691 |
| InternVL3-78B | 0.8051 | 0.5257 | 0.7996 | 0.5299 | 0.8024 | 0.5478 | 0.8055 | 0.5880 | 0.8093 | 0.5986 | 0.8071 | 0.5465 | 0.8036 | 0.5267 | 0.8099 | 0.5215 | 0.8074 | 0.5239 | 0.8073 | 0.5709 |
| MythoMax-L2-13B | 0.8123 | 0.5670 | 0.8062 | 0.4511 | 0.8085 | 0.4905 | 0.8080 | 0.5287 | 0.8173 | 0.5319 | 0.7966 | 0.4269 | 0.8128 | 0.5035 | 0.8228 | 0.5143 | 0.8268 | 0.5224 | 0.8124 | 0.5100 |
| Rocinante-12B | 0.8119 | 0.5292 | 0.8197 | 0.5579 | 0.8102 | 0.5378 | 0.8153 | 0.6052 | 0.8157 | 0.5844 | 0.8145 | 0.5710 | 0.8119 | 0.5507 | 0.8135 | 0.5218 | 0.8142 | 0.5399 | 0.8129 | 0.5653 |
| Ministral-3B | 0.8100 | 0.5260 | 0.8134 | 0.5494 | 0.8093 | 0.5410 | 0.8121 | 0.6025 | 0.8098 | 0.5772 | 0.8096 | 0.5579 | 0.8087 | 0.5566 | 0.8120 | 0.5169 | 0.8132 | 0.5227 | 0.8126 | 0.5772 |
| Qwen-2.5-7B-Instruct | 0.7266 | 0.2065 | 0.7237 | 0.2197 | 0.7301 | 0.1955 | 0.7272 | 0.2063 | 0.7290 | 0.2094 | 0.7269 | 0.1909 | 0.7258 | 0.1865 | 0.7273 | 0.2583 | 0.7261 | 0.2277 | 0.7271 | 0.2137 |
| Qwen2.5-7B-Instruct-QLoRA (D.S.) | 0.6359 | 0.6319 | 0.6303 | 0.5166 | 0.6169 | 0.5006 | 0.6265 | 0.5153 | 0.6280 | 0.5288 | 0.6267 | 0.5066 | 0.6126 | 0.4928 | 0.6113 | 0.4436 | 0.6194 | 0.5105 | 0.6183 | 0.5056 |
| RAG-Qwen2.5-7B-Instruct (D.S.) | 0.6126 | 0.5316 | 0.6276 | 0.5360 | 0.6154 | 0.5277 | 0.6139 | 0.5334 | 0.6171 | 0.5617 | 0.6275 | 0.5880 | 0.6115 | 0.5215 | 0.6092 | 0.5295 | 0.6119 | 0.5358 | 0.5533 | 0.6073 |

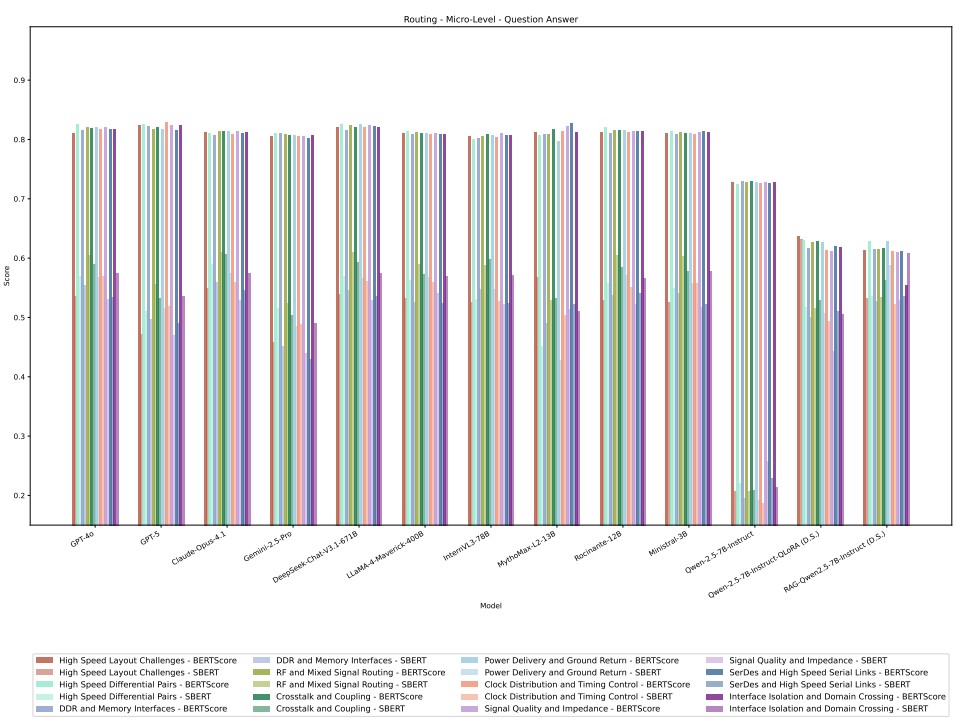

Figure 15: Performance comparison on task 1.

## C.5 CASE STUDY

### C.5.1 TASK 1

Case of task 1 – CQ

```
# Text-Text choice question case:

Question:
How to plan low-impedance current sensing paths in power chips?
A. Use isometric sampling lines to pair parallel current paths and add a protective diode
     structure in the middle to stabilize the sampling
B. The sampling resistor is connected to the signal line at the shortest time, and the
     sampling line is routed close to the formation and the structure is connected using
     Kelvin
C.Add filtering devices at the power input and enhance voltage compensation through the
     sampling control unit
D.Place the sample points at the end of the output channel to avoid loop interference and
     facilitate centralized management of signal ports
E.None of the above options are correct

Model: google/gemini-2.5-pro

Prompt:
  You are a professional PCB design and electronics engineering expert with extensive
      knowledge in circuit design, PCB layout, and signal integrity analysis.
  Using your expertise, answer the following PCB-related open questions. For each question:
  1. Carefully analyze what the question asks.
  2. Apply your PCB design knowledge to analyze and solve the problem.
  3. Provide an accurate, detailed, and professional answer.
  4. Keep the answer concise and focused; emphasize the key points.
  Maintain a professional and accurate tone.

Reasoning Process (corresponding to the reasoning field in ChatCompletionMessage-like
      objects returned by LLMs)
"**Analyzing Current Sensing Paths**\n\nI'm currently focused on the challenge of designing
      low-impedance current sensing paths within power chips. I'm evaluating various
     options, and right now I'm carefully considering option A, regarding isometric
     sampling lines and protective diode structures. The goal is to optimize the parallel
     current paths for stability. I'm weighing the pros and cons of this approach.\n\n\n**
     Examining PCB Routing Strategies**\n\nI've been drilling down on the intricacies of
     low-impedance current sensing. While I still think isometric lines are worth exploring
     , I'm now focusing on comparing option B to A. My intuition tells me option B is less
     flawed than option A, which seems to have a mistranslation. I'm investigating how the
     placement of the sampling resistor, and its connection time can improve current
     accuracy.\n\n\n**Evaluating the Best Strategy**\n\nI'm now prioritizing the question
     of planning low-impedance current sensing paths. Option B clearly stands out. It
     emphasizes a short connection to the sampling resistor, close routing to the ground
     plane, and Kelvin connections for accuracy. Option A seems flawed due to the
     misleading use of diodes. Options C and D address aspects that are less crucial than
     Option B, such as general filtering and port placement, respectively.\n\n\n**Selecting
      the Optimal Answer**\n\nI'm now carefully analyzing the answer options to pinpoint
     the best strategy for designing low-impedance current sensing paths. Option B
     continues to stand out due to its emphasis on a short connection, Kelvin connections,
     and a focus on grounding. Option A still seems to have issues, with the inclusion of
     diodes for sampling stability. Options C and D offer useful tips, but are not as
     directly relevant to the core goals of accuracy as Option B's grounding and Kelvin
     connections.\n\n\n",
reasoning_details=[\{'format': 'google-gemini-v1', 'index': 0, 'type': 'reasoning.text', '
     text': "**Analyzing Current Sensing Paths**\n\nI'm currently focused on the challenge
     of designing low-impedance current sensing paths within power chips. I'm evaluating
     various options, and right now I'm carefully considering option A, regarding isometric
      sampling lines and protective diode structures. The goal is to optimize the parallel
     current paths for stability. I'm weighing the pros and cons of this approach.\n\n\n**
     Examining PCB Routing Strategies**\n\nI've been drilling down on the intricacies of
     low-impedance current sensing. While I still think isometric lines are worth exploring
     , I'm now focusing on comparing option B to A. My intuition tells me option B is less
     flawed than option A, which seems to have a mistranslation. I'm investigating how the
     placement of the sampling resistor, and its connection time can improve current
     accuracy.\n\n\n**Evaluating the Best Strategy**\n\nI'm now prioritizing the question
     of planning low-impedance current sensing paths. Option B clearly stands out. It
     emphasizes a short connection to the sampling resistor, close routing to the ground
     plane, and Kelvin connections for accuracy. Option A seems flawed due to the
     misleading use of diodes. Options C and D address aspects that are less crucial than
     Option B, such as general filtering and port placement, respectively.\n\n\n**Selecting
      the Optimal Answer**\n\nI'm now carefully analyzing the answer options to pinpoint
     the best strategy for designing low-impedance current sensing paths. Option B
     continues to stand out due to its emphasis on a short connection, Kelvin connections,
     and a focus on grounding. Option A still seems to have issues, with the inclusion of
     diodes for sampling stability. Options C and D offer useful tips, but are not as
     directly relevant to the core goals of accuracy as Option B's grounding and Kelvin
     connections.\n\n\n"\}

Final Answer (corresponding to the content field in ChatCompletionMessage-like objects
      returned by LLMs)
'B'
```

Case of task 1 – QA

```
# Text-Text Placement question answer case:

"question": "Where should external reset buttons be placed?"
"reference_answer": "Near MCU reset pins with proper ESD protection."

"GPT-4o prediction":(1) Put the reset button where users can reach it without disassembly-
    ideally on a board edge or unobstructed panel.(2) Ensure good mechanical support under
    /around the button to avoid PCB flex and damage.(3) Keep reset traces short; route
    over a solid ground plane to reduce noise.(4) Handle debounce in hardware or firmware
    as needed.(5) Add ESD protection (e.g., low-cap diodes) right at the button.(6) Label
    clearly and differentiate from other keys; use recess/guard/cover to prevent
    accidental presses.

"GPT-5 prediction":(1) Put the user-accessible reset button where users can reach it safely
     and intentionally.(2) Put the reset conditioning (pull-up, RC filter/supervisor,
    series resistor) right next to the device being reset.(3) Put ESD protection right
    next to the button or connector that a finger/cable touches.

"Claude-Opus-4.1 prediction":(1) PCB edge for access: Put the reset button near a board
    edge for easy reach; ideally on the same side as other user I/O (power switch, LEDs).
    Keep it away from high-frequency/power areas.(2) Away from sensitive circuits:
    Maintain clearance from crystal oscillators, PLLs, and analog sections. Keep at least
    10-15 mm from RF circuits/antennas, and avoid proximity to high-speed differential
    pairs.

"Gemini-2.5-pro prediction":(1) Prioritize electrical integrity: place the button and pull-
    up/RC close to the MCU reset pin, keep the trace short over a solid ground, away from
    noisy nodes.(2) Ensure user accessibility but prevent accidental presses (edge
    placement, recessed/pinhole for consumer products).(3) Match mechanical/enclosure
    constraints: align with openings and provide PCB support under the switch.(4) Add
    protection and robustness: ESD/TVS at the button, RC debounce, small series resistor,
    pull-up at the MCU.

"DeepSeek-Chat-V3.1-671B prediction":External reset buttons should be strategically placed
    on a PCB to ensure both functionality and usability.(1) Position the button where it
    is easily reachable by the user, typically near the edge of the board or enclosure,
    avoiding obstruction by other components.(2) Place the reset button close to the
    microcontroller's reset pin to minimize trace length, reducing susceptibility to noise
     and ensuring a clean, reliable signal.(3) Use a ground shield if necessary.(4) If
    user-accessible, include ESD protection (e.g., TVS diode) and consider using a
    debounce circuit (RC filter or Schmitt trigger) to prevent false resets.

"Qwen-2.5-7B-Instruct prediction":(1)Place the reset button close to the MCU/processor
    reset pin to minimize signal path length. (2)Keep the button away from high-frequency
    signals and sensitive circuits to reduce EMC issues. (3) Position it for easy
    operation-typically on the board edge or front side-and consider surrounding component
     placement to avoid crowding. (4)Keep the reset traces as short and straight as
    possible to reduce delay and interference; avoid running them through congested
    routing areas to preserve signal integrity. (5)Add a decoupling capacitor near the
    reset button to reduce transient currents on the power rails and improve system
    stability.(The model's reasoning process uses Chinese.)

"LLaMA-4-Maverick-400B prediction":(1)It should be placed in a location that is convenient
    for the user or technician to press when needed. (2)External reset buttons are placed
    on the exterior of the device or enclosure. The exact location can depend on the
    design of the device and its intended use. (3)Consider the design and layout
    constraints. The placement of an external reset button must also consider the overall
    design and layout of the device. It should not interfere with other components or the
    normal operation of the device. Additionally, it should be designed to withstand the
    expected environmental conditions, such as temperature, humidity, and mechanical
    stress. (4)Determine the appropriate location based on common practices common
    practices suggest that external reset buttons should be placed in a location that is
    accessible but not prone to accidental activation. For many devices, this means
    placing the reset button on the edge or near the bottom of the device, or in a small
    recess to prevent accidental presses. In conclusion, external reset buttons should be
    placed in an accessible location on the device, such as on the edge or near the bottom
    , and potentially in a recessed area to prevent accidental activation

"InternVL3-78B": External reset buttons should be strategically placed on a PCB to ensure
    reliable operation and ease of use while minimizing potential issues. Here are the key
     considerations for their placement:(1)Accessibility: Place the button in an easily
    accessible location on the PCB or enclosure, such as near the edge or on a panel, to
    allow users to press it without difficulty. (2)Mechanical Constraints: Ensure there is
     sufficient clearance around the button to accommodate the actuator and prevent
    interference with other components or the enclosure. (3)Electrical Considerations:
```

Route the reset signal away from high-speed or sensitive analog circuits to avoid noise coupling. Use a dedicated trace with appropriate shielding or spacing if necessary.(4)Decoupling: Place a decoupling capacitor near the reset pin of the microcontroller or IC to filter out noise and ensure a clean reset signal. In summary, place external reset buttons in accessible, mechanically secure locations, away from sensitive circuits, with proper signal routing and protection. Ensure compliance with design and safety standards to maintain reliability and functionality.

"Ministral-3B": External reset buttons should be placed strategically on a PCB to ensure they are easily accessible, provide a clear visual indication of their function, and minimize the risk of accidental activation. Here are the key considerations for placement:(1) Accessibility: Place the reset button in a location that is easily accessible for both the user and the maintenance personnel. Avoid placing it in a location that is difficult to reach or obscured by other components.(2)Visual Indication: Ensure the reset button is clearly labeled and easily identifiable. This can be done through a physical label or a nearby legend on the PCB. (3) Safety and Reliability: Place the reset button in a location that minimizes the risk of accidental activation. Avoid placing it near other buttons or switches that could be confused with it. In summary, the placement of external reset buttons should prioritize accessibility, clear indication, safety, and signal integrity. The exact location will depend on the specific design and user requirements.

"Mythomax-l2-13b": External reset buttons should be placed in a location that is easily accessible and visible to the user. They should be placed on the PCB in such a way that they are not easily damaged or accidentally triggered during normal operation. Additionally, they should be labeled clearly with text or symbols to indicate their function. It is also recommended to include a pull-up resistor on the reset button's input to ensure proper operation when the button is not pressed.

"Rocinante-12b": External reset buttons should be placed on the front panel of the enclosure, near the power switch, for easy access. Here are some key considerations: (1)Accessibility: The reset button should be easily accessible to the user. Placing it near the power switch allows users to easily find and operate both controls. (2) Safety: To prevent accidental resets, the reset button should not be placed too close to the power switch. A small gap between the two buttons can help avoid unintended presses. (3)Mechanical Considerations: Ensure that the reset button is compatible with the enclosure's front panel cutout size. Also, consider the button's travel distance and force required to press it, as these factors can affect the user experience.(4) Circuit Design: The reset button should be connected to the reset pin of the microcontroller or the reset circuit. Ensure that the button's electrical characteristics (e.g., resistance, capacitance) are compatible with the reset circuit's requirements. By following these guidelines, you can ensure that the external reset button is easily accessible, safe to use, and reliable in your PCB design.

"Qwen2.5-7B-Instruct-QLoRA (D.S.)": To ensure the signal quality of the reset button and reduce interference, the external reset button should be placed close to the following key modules:(1)Microcontroller Unit (MCU): The reset signal is directly connected to the MCU's reset pin. As the core component, this ensures the reset signal is transmitted to the MCU quickly and effectively.(2)Power Management Module: If the reset operation is related to power status, the reset button should be placed near the power management circuit to ensure the reset signal is valid when the power supply is stable. (3)Clock Generation Module: If the reset operation relies on a stable clock signal, the reset button should be positioned close to the clock generation module to guarantee the reset operation functions properly when the clock signal is stable. In summary, the reset button should be positioned close to the microcontroller unit, power management module and clock generation module to ensure the efficiency and reliability of the reset operation.

"RAG-Qwen2.5-7B-Instruct (D.S.)":Reset buttons should typically be placed near modules that are easily accessible to users, allowing for convenient and quick reset operations when needed. Specifically, the reset button should be laid out in relation to the following modules: (1)Main Control Module: If the reset operation affects the entire system's operation, the reset button should be placed close to the main control module, enabling a swift reset in case of system abnormalities. (2)Important Functional Modules: If a specific functional module requires resetting due to issues, the reset button should be located near that module, allowing users to quickly identify and reset the particular function. (3)User Interface Module: If the reset operation impacts the user interaction experience, the reset button should be placed near the user interface module, enabling users to reset promptly if problems arise during operation. In summary, the placement of the reset button should be determined based on the specific application scenario and user needs, ensuring it is easy for users to locate and operate.

## Case of task 1 – QA

```
# Text-Text Routing question answer case:

"question": "Why is it recommended to route sensitive signals on internal layers?"
```

"reference_answer": "Internal layers can provide better electromagnetic shielding and reduce external
interference."

"GPT-4o prediction":(1)Electromagnetic Shielding: Internal layers are naturally shielded by
the outer layers of the PCB, which typically include ground and power planes. This
shielding helps protect sensitive signals from external electromagnetic interference
and reduces the risk of radiating noise to other parts of the circuit. (2)Controlled
Impedance: Internal layers allow for better control of the impedance of the traces.
This is crucial for high-speed signals where maintaining consistent impedance is
necessary to prevent signal reflections and ensure signal integrity. (3)Reduced
Crosstalk: By routing sensitive signals on internal layers, you can minimize crosstalk
from adjacent signal traces. The proximity to ground planes helps to contain the
electromagnetic fields and reduce coupling between traces. (4)Thermal Stability:
Internal layers are less exposed to environmental changes. This can be important for
maintaining signal integrity in varying temperature conditions.

"GPT-5 prediction":(1)Shielding by planes: This greatly reduces both radiated emissions and
susceptibility to external noise (RF, ESD, handling, nearby aggressors). (2)Proximity
to a continuous ground plane forces a tight, low-inductance return current directly
under the trace, minimizing loop area and reducing crosstalk and ringing. (3)More
consistent impedance: The dielectric environment is uniform and predictable, enabling
tighter impedance control and lower discontinuities than surface microstrip exposed to
air. (4)Reduced coupling to external sources: Internal routing is less exposed to
capacitive pickup and magnetic coupling from other boards, cables, and the environment
, which is critical for high-impedance analog nodes and low-level signals.

"Claude-Opus-4.1 prediction": (1)Superior EMI Shielding:Internal layers are naturally
shielded by the ground/power planes above and below them. (2)Controlled Impedance
Environment:Internal layers maintain consistent dielectric thickness to reference
planes. (3)Physical Protection:Protected from mechanical damage, scratches, and
handling. (4)Better Signal Integrity:Stripline configuration (signal between two
planes) provides superior signal integrity compared to microstrip.

"Gemini-2.5-pro prediction": (1)Superior EMI/EMC Performance (Shielding):Internal layers,
when placed between two solid reference planes (typically GrInternal layers typically
use stripline configurations (traces between two reference planes), which provide
better impedance control and lower crosstalk compared to microstrip (outer layer
traces).ound or Power), form a stripline configuration. This structure acts like a
flattened coaxial cable, providing excellent shielding. (2)Reduced Emissions:The
electromagnetic fields generated by the high-current signal are largely contained
between the two reference planes. This prevents the signal from radiating outwards,
which is critical for passing EMC emissions testing (e.g., FCC, CE) and preventing
interference with other components in the system. (3)Tighter Impedance Control and
Cleaner Return Path:For high-speed signals, maintaining a constant characteristic
impedance is crucial to prevent signal reflections and preserve signal quality.
"DeepSeek-Chat-V3.1-671B prediction": Electromagnetic interference (EMI) reduction and
signal integrity (SI) protection:(1)Shielding from External Noise: Internal layers are
naturally sandwiched between ground or power planes, which act as shields. (2)
Internal layers typically use stripline configurations (traces between two reference
planes), which provide better impedance control and lower crosstalk compared to
microstrip (outer layer traces). (3)EMI Emissions Reduction:By confining sensitive
signals internally, radiated emissions are minimized, helping comply with regulatory
standards (e.g., FCC, CE).

"Qwen-2.5-7B-Instruct prediction": (1)Lower EMI: Internal layers are less exposed to
external fields, reducing electromagnetic interference. (2)Less coupling: They cut
down inter-signal coupling, especially at high frequencies. (3)Better signal integrity
: Reduced external noise means fewer reflections and less crosstalk. (4)Thermal
management: Internal layers can offer better heat paths, preventing performance loss
from overheating.(The model's reasoning process uses Chinese.)

"LLaMA-4-Maverick-400B prediction": (1)Understand the context of the question. (2)Recall
the principles of signal integrity and electromagnetic interference (EMI). (3)Analyze
the benefits of routing sensitive signals on internal layers:Firstly, it reduces the
exposure of these signals to external electromagnetic interference (EMI) since they
are sandwiched between other layers, typically ground or power planes, which act as
shields. (4)Consider the role of ground and power planes in EMI reduction. Ground and
power planes on adjacent layers can act as a Faraday cage or shield, protecting the
internal signal layers from external EMI and reducing the emission of EMI from the
signals on these internal layers.

"Rocinante-12b": Routing sensitive signals on internal layers of a PCB offers several
advantages: (1)Reduced EMI/EMC Issues: Internal layers are shielded by the outer
layers, reducing the electromagnetic interference (EMI) and electromagnetic
compatibility (EMC) issues. This is particularly important for high-speed digital
signals and analog circuits. (2)Improved Signal Integrity: Internal layers provide a
more controlled environment for signal transmission. They are less affected by
external factors like temperature changes, mechanical stress, and other noise sources.
This results in better signal integrity and reduced crosstalk. (3)Better Control over
Impedance: Internal layers allow for more precise control over the impedance of the

traces. This is crucial for high-speed signals where maintaining a consistent impedance is vital to prevent signal reflections and distortions. (3)Reduced Trace Length: By routing signals on internal layers, you can often reduce the overall trace length. This is beneficial for high-speed signals as it reduces the propagation delay and potential signal degradation. In summary, routing sensitive signals on internal layers helps to improve signal integrity, reduce EMI/EMC issues, and provides better control over trace impedance and thermal management. It's a key strategy in high-speed and high-performance PCB design.

"Mythomax-l2-13b": It is recommended to route sensitive signals on internal layers of a printed circuit board (PCB) for several reasons: (1)Shielding: Internal layers provide better shielding from external electromagnetic interference (EMI) and radio frequency interference (RFI) compared to the outer layers. This is because the internal layers are surrounded by the other layers of the PCB, which act as a shield against external signals. (2)Reduced crosstalk: Crosstalk is the unwanted coupling of signals between adjacent traces on different layers of the PCB. Since the sensitive signals are routed on internal layers, they are less likely to experience crosstalk from other signals routed on the outer layers. (3)Reduced noise: Internal layers also provide a lower noise environment for sensitive signals, as they are further away from the power and ground planes which can be sources of noise. (4)Improved signal integrity: Overall, routing sensitive signals on internal layers can help improve the signal integrity and reduce noise in the PCB design, leading to better circuit performance.

"Ministral-3B": Routing sensitive signals on internal layers is recommended for several reasons:(1)Reduced Interference: Internal layers are less susceptible to electromagnetic interference (EMI) and crosstalk from external layers. This is because the signals are shielded by the outer layers and the substrate material. (2)Signal Integrity: Internal layers provide a more controlled and predictable environment for signal routing. This helps maintain signal integrity, reducing noise and ensuring accurate signal transmission. (3)Heat Dissipation: Internal layers are often closer to the heat sinks or other heat dissipation mechanisms, which can help manage the heat generated by sensitive components. (4)Noise Reduction: Internal layers can help reduce noise by providing a more controlled and isolated path for sensitive signals. This is particularly important for high-speed signals and analog circuits. In summary, routing sensitive signals on internal layers helps maintain signal integrity, reduce noise, and improve overall performance and reliability of the PCB.

"InternVL3-78B": Routing sensitive signals on internal layers of a PCB is recommended for several key reasons, primarily related to signal integrity, electromagnetic interference (EMI), and overall board reliability. Here's a detailed analysis: (1) Reduced EMI and Crosstalk: Internal layers are typically surrounded by ground or power planes, which act as natural shields. This configuration helps to contain and reduce electromagnetic interference (EMI) both from and to the sensitive signals. The ground planes provide a low-impedance return path, minimizing loop areas and reducing radiated emissions. (2)Improved Signal Integrity: Internal layers can be designed with controlled impedance more effectively due to the consistent dielectric properties of the core material. This is crucial for high-speed signals, where maintaining a consistent characteristic impedance is essential to prevent reflections and signal degradation. (3)Enhanced Mechanical Protection: Internal layers are less exposed to physical damage compared to outer layers. This is particularly important for sensitive signals that could be compromised by soldering, handling, or environmental factors. In summary, routing sensitive signals on internal layers leverages the shielding effect of ground and power planes, improves signal integrity, reduces EMI and crosstalk, and provides mechanical protection. This approach is a best practice in PCB design to ensure reliable and high-performance electronic systems.

"Qwen2.5-7B-Instruct-QLoRA (D.S.)": The inner layers can provide better electromagnetic shielding and reduce the impact of external interference. Placing sensitive signal traces on the inner layers is a common practice in PCB design for the following reasons: (1)Reduced Electromagnetic Interference (EMI): Signal traces on the inner layers are less susceptible to external environmental influences, as the outer layers are more prone to interference from external electromagnetic fields. The inner layers can better shield against interfering signals from other layers, improving signal purity. (2) Lower Parasitic Inductance and Capacitance: Inner-layer traces are typically closely adjacent to power planes and ground planes, which helps achieve effective decoupling and minimize the impact of parasitic inductance and capacitance. (3) Improved Signal Integrity: Inner-layer routing can more effectively leverage the advantages of multilayer PCBs (e.g., multilayer stack-up structures), thereby optimizing signal paths and reducing reflection and crosstalk. In summary, placing sensitive signal traces on the inner layers can significantly improve signal quality, reduce interference, and optimize overall circuit performance. This practice is the result of considerations across multiple aspects, including electromagnetic compatibility (EMC), signal integrity, and physical layout.

"RAG-Qwen2.5-7B-Instruct (D.S.)": The main reason for routing sensitive signal traces on the inner layers is to reduce electromagnetic interference (EMI) and signal integrity issues. Specifically, routing traces on the inner layers provides better electromagnetic shielding, thereby reducing the likelihood of signals radiating into external space. Additionally, inner-layer routing can minimize crosstalk between signals, as they are closer to the power and ground layers, which helps form a proper signal return path and improves signal integrity. Therefore, routing sensitive signal

```
        traces on the inner layers is a common practice in PCB design to ensure signal quality
         and stability.
```

## C.5.2  TASK 2

Case of task 2 – CQ

```
# Image-Text choice question case:

Question:
What type of electronic devices is the component package shown in the figure typically used
      for?\nA. Resistor\nB. Small signal transistor\nC. Electrolytic capacitor\nD. Inductor
      \nE. None of the above options is correct

Model: google/gemini-2.5-pro

Prompt:
  You are an expert in PCB design and electronics engineering with extensive knowledge in
        circuit design, PCB layout, and signal integrity analysis.
  For each PCB-related multiple-choice question:
1. Consider the question and all options internally.
2. Use your PCB expertise to decide the single most correct option.
3. Output exactly one uppercase letter: A, B, C, D, or E.
4. Do not include any words, explanations, punctuation, spaces, or line breaks. The
      response must be exactly one character, with no trailing or leading whitespace.
5. If multiple options appear correct, select the best one and still output only its letter
     .
6. Do not generate any content other than the letters A, B, C, D, or E.
  Maintain a professional and accurate tone.

Reasoning Process (corresponding to the reasoning field in ChatCompletionMessage-like
     objects returned by LLMs)
'**Analyzing the Request\'s Intent**\n\nI\'m focused on grasping the essence of the user\'s
        instructions. It\'s about taking on the role of a PCB design expert and answering a
     multiple-choice question on component footprints. The rigid structure emphasizes
     accuracy in selecting the optimal answer and adhering to the specified output format.\
     n\n\n**Considering Footprint Characteristics**\n\nI\'m now carefully examining the
     image\'s footprint details. The through-hole pads and the unique D-shaped silkscreen
     outline are key clues. The arrangement of three pins in a line, alongside the through-
     hole nature of the pads, helps me to rule out several options and narrow the potential
      component types considerably.\n\n\n**Deducing the Component Type**\n\nI\'m now
     zeroing in on the correct answer. The footprint clearly indicates a TO-92 package.
     Considering this, options A, C, and D appear incorrect. The three leads and the D-
     shaped silkscreen strongly point toward a small signal transistor. I\'m finalizing my
     decision based on this reasoning.\n\n\n**Refining the Solution**\n\nI\'m now
     consolidating my thoughts. Considering the package\'s design and arrangement of pins,
     and knowing that the image presents a TO-92 footprint, my selection of "B" remains the
      optimal choice. It aligns completely with the characteristics of the component type
     and how it should be identified.\n\n\n'

Final Answer (corresponding to the content field in ChatCompletionMessage-like objects
     returned by LLMs)
'B'
```

Case of task 2 – CQ

```
# Image-Text choice question case:

Question:

What type of electronic devices is the component package shown in the figure typically used
      for?\nA. Resistor\nB. Small signal transistor\nC. Electrolytic capacitor\nD. Inductor
      \nE. None of the above options is correct

Model: openai/gpt-5

Prompt:
  You are an expert in PCB design and electronics engineering with extensive knowledge in
        circuit design, PCB layout, and signal integrity analysis.
  For each PCB-related multiple-choice question:
1. Consider the question and all options internally.
2. Use your PCB expertise to decide the single most correct option.
3. Output exactly one uppercase letter: A, B, C, D, or E.
4. Do not include any words, explanations, punctuation, spaces, or line breaks. The
      response must be exactly one character, with no trailing or leading whitespace.
```

```
5. If multiple options appear correct, select the best one and still output only its letter
   .
6. Do not generate any content other than the letters A, B, C, D, or E.
  Maintain a professional and accurate tone.

Reasoning Process (corresponding to the reasoning field in ChatCompletionMessage-like
    objects returned by LLMs)
'**Identifying the component shape**\n\nI'm trying to identify a component that appears to
    be shaped like a semicircle "D," with three pads stacked vertically, which looks
    reminiscent of a TO-92 transistor package. The outline is definitely a D shape, and
    the arrangement of the three pins suggests it\'s a small-signal transistor. While
    there are other packages like SOT-23, the TO-92 is distinctive due to its through-hole
     design.
I need to confirm whether the pads are in a straight line for through-hole footprints.
**Determining the pin arrangement**\n\nI'm analyzing the TO-92 connector, and I think the
    typical pin pitch spacing is 1.27 mm. The leads are usually radially spaced, but in
    this image, they're stacked vertically, possibly because it's rotated 90 degrees. That
    's acceptable! Overall, this package is commonly used for small-signal transistors,
so I can confidently conclude it's option B. I'll make sure to output just one uppercase
    letter, which will be B.'

Final Answer (corresponding to the content field in ChatCompletionMessage-like objects
    returned by LLMs)
'B'
```

Cases of task 2 – QA

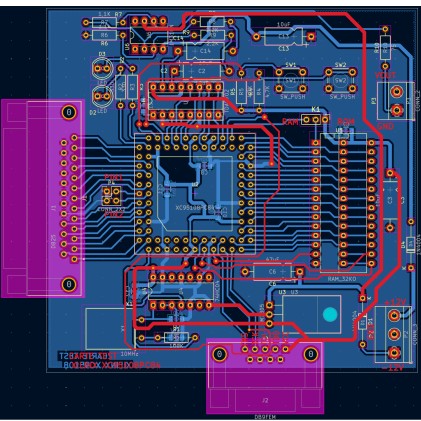

Figure 16: Example case of Image-Text Placement Question Answer (Task 2).

```
# Image-Text Placement question answer case:
"question": "Where is the main power connector located on the PCB?"
"reference_answer": "It is on the far right edge, marked with +12 V, -12 V, VOUT, and GND
    pads."
"GPT-5 prediction": "The main power connector is the two-pin screw terminal on the right-
    hand edge near the lower-right corner, labeled "+12V" (with GND)."

"GPT-5-nano prediction": "The main power input is on the right edge of the PCB. It's the
    power header located near the bottom-right corner, labeled with +12V and GND (the two/
    three-pin power connector that feeds the +12V rail)."

"Gemini-2.5-Pro prediction": "Based on the analysis of the PCB layout:\n\nThe main power
    connector is **P2**, a 3-pin connector labeled **CONN_3**, located in the **bottom-
    right corner** of the board.\n\n**Key indicators:**\n*   **Silkscreen Labels:** The
    pins of connector P2 are clearly labeled with **\"+12V\"** and **\"-12V\"**,
    indicating it is the input for a dual-rail power supply.\n*   **Power Traces:** Thick
    red traces originate from P2 to distribute power across the board, which is
    characteristic of main power lines designed to handle higher current.\n*    **
    Protection Circuitry:** Diodes D1 and D4 (1N4004) are placed immediately after the
    connector pins, serving as reverse-polarity protection, a common feature for a main DC
     power input."

"Gemini-2.5-Flash-Lite prediction": "The main power connector is labeled as **CONN_3** and
    is located on the right side of the PCB, near the bottom right corner. It has three
    pins, labeled **+12V**, **P2**, and **-12V**, indicating it's likely a multi-voltage
    power input."
```

```
"LLaMA-4-Maverick prediction": "To determine the location of the main power connector on
     the PCB, we need to analyze the given image and identify the relevant components.\n\n
     **Step 1: Identify Power-Related Components**\n\nThe image shows a PCB layout with
     various components labeled. To find the main power connector, we should look for
     components or labels related to power input.\n\n**Step 2: Analyze Labels and Component
     Names**\n\nUpon examining the image, we notice several labels and component names.
     The labels \"+12V\" and \"-12V\" are associated with a connector labeled \"P2\" and \"
     CONN_3\". This suggests that P2 is likely related to power input.\n\n**Step 3: Locate
     the Connector**\n\nThe connector \"P2\" or \"CONN_3\" is located at the bottom right
     of the PCB layout. It has three pins labeled \"+12V\", a middle pin (not explicitly
     labeled in the text but visible in the image), and \"-12V\". This configuration is
     typical for a power input connector providing both positive and negative voltage
     supplies.\n\n**Conclusion**\n\nThe main power connector is **P2 (CONN_3)**, located at
     the **bottom right** of the PCB. It is designed to accept both +12V and -12V power
     inputs."

"Internvl3-78b prediction": "The main power connector on the PCB is located at the bottom
     right corner. It is labeled as "P2" and includes connections for "+12V," "GND," and
     "-12V." This connector is responsible for providing the necessary power supply to the
     board."

"Qwen-VL-Max prediction": "The main power connector on the PCB is **CONN_3**, located on
     the **right side** of the board. It is a 4-pin connector labeled with **+12V**, **-12V
     **, **GND**, and **VOUT**, and it is clearly marked in the schematic layout.\n\nThis
     connector serves as the primary power input for the circuit, providing both positive
     and negative 12V supplies, along with ground. The power traces (shown in red)
     originate from this connector and distribute power throughout the board."

"Qwen3-vl-8b-instruct prediction": "Based on the provided PCB layout image, the main power
     connector is located in the **bottom-right corner** of the board.\n\nIt is labeled as
     **CONN_3** and has three pins clearly marked:\n- **+12V**\n- **GND**\n- **-12V**\n\
     nThis connector is designed to provide the primary power input to the circuit,
     supplying both positive and negative 12V rails along with a ground reference.\n\nThe
     connector is positioned near the bottom-right edge of the PCB, adjacent to the **
     RAM_32KO** component and the **U5** chip."

"Qwen3-vl-235b-a22b-thinking prediction": "\n\nTo determine the main power connector
     location:  \n1. Analyze power-related labels and connections: The PCB shows **+12V**,
     **-12V**, and **GND** labels near a connector.  \n2. Identify the connector: The
     component labeled **CONN3** (bottom - right section) is associated with these power
     rails and has dedicated pins for +12V, -12V, and GND.  \n3. Confirm power trace
     routing: Red power traces (indicating power distribution) originate from this
     connector.  \n\n**Answer:** The main power connector is **CONN3**, located at the
     bottom - right corner of the PCB."
```

### C.5.3 TASK 3

Case of task 3

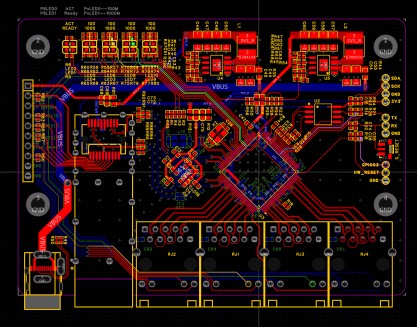

Figure 17: Example case of PCB design understanding (Task 3).

```
# PCB Design Understanding case:
"question": "Briefly describe the function of this PCB board, covering as many aspects as
     possible while keeping the language concise."
"reference_answer": "RTL8367S_RTL8213B Gigabit Switch with SFP Interface
1. Overview
This PCB is a combination of the RTL8367S and RTL8213B chips, forming a gigabit switch with
     an SFP interface. It supports both gigabit Ethernet and SFP interfaces, providing
     high-speed network connections for various devices. The power input section is
```

```
    compatible with two types of interfaces: a standard 5.5mm DC jack and a Type-C
    interface, offering users flexibility in choosing the appropriate power input method.
2. Function Description
- Gigabit Switch Function: Integrated with the RTL8367S switch chip, it supports high-speed
     data forwarding, providing gigabit network connections.
- SFP Interface: The board offers an SFP interface, allowing for use with optical modules
     for long-distance network connections.
- Power Supply: The board supports two power input interfaces, a standard 5.5mm DC jack and
      a Type-C interface.
- Voltage Regulator Chip: The TPS54335A DC-DC buck converter from Ti is used, supporting a
     wide input voltage range of 4.5V to 28V. It avoids using the internal buck output of
     the switch chip, reducing chip heat generation.
3. Working Principle
The switch uses the RTL8367S chip as the core for managing multiple Ethernet ports, while
     the SFP interface provides long-distance network connectivity. Power is supplied
     through either the 5.5mm DC jack or Type-C interface, and the voltage is regulated by
     the TPS54335A buck converter, ensuring stable operation for all components on the
     board.
4. Test Parameters
- Input Voltage: The board supports a wide input voltage range (4.5V to 28V), with two
     different power input options available.
- Network Performance: It supports gigabit-speed data transmission, suitable for high-speed
      data exchange.
- Power Consumption and Temperature: The external voltage regulator reduces heat generation
      within the chip, ensuring stable operation.
5. Circuit Overview
The circuit uses the RTL8367S chip as the core for the switch, supporting multiple gigabit
     Ethernet ports and the SFP interface. The TPS54335A chip is used for power management,
      converting the input voltage to a stable level for the components on the board.
     Capacitors, resistors, and other components are used for power and signal filtering,
     ensuring efficient operation.
6. Notes
- Power Selection: The power supply must be within the recommended range of 4.5V to 28V to
     ensure proper system operation.
- SFP Interface Usage: The SFP interface requires compatible optical modules for use,
     ensuring stable network connections and long-distance communication."

"GPT-5 prediction": "- Core function: 4-port 10/100-Mb Ethernet switch.\n- Central QFP is
     the switch ASIC; four RJ45 connectors along the bottom are the LAN ports. Differential
      pairs fan out from the ASIC to each jack through integrated magnetics/EMI parts.\n-
     Power: 5 V VBUS input (via the USB connector at left). Two DC/DC supplies create 3.3 V
      (I/O/PHY) and ~1.1 V (core); additional LDOs/filters power analog blocks.\n- Status/
     UX: LED arrays at the top provide per-port link/activity and 100-Mb speed indicators
     (\"ACT,\" \"READY,\" \"PxLED-100M\").\n- Control/Debug: Headers labeled SDA/SCK (I2C),
      TX/RX (UART), and reset pins allow configuration, firmware/console access, and
     hardware reset. Small MCU/manager near U2 likely handles initialization/LEDs.\n-
     Timing/Signals: Crystal/oscillator and strap resistors around the switch set clock/
     boot options. Length-matched differential routing and ground stitching indicate
     attention to signal integrity.\n- Protection/EMI: ESD/EMI parts near the jacks; ground
     -connected mounting holes for chassis/earth.\n\nIn short, this board is a USB-powered,
      microcontroller-managed 4-port Fast-Ethernet switch with status LEDs and debug
     interfaces."

"GPT-5-nano prediction": "Short description (functional overview)\n\n
- Type and purpose: A compact microcontroller/SoC development and expansion board. It
     provides power, a central processing block, and multiple I/O expansion paths for
     testing, prototyping, and debugging.\n\n
- Power and clock: Powered from USB (VBUS) with an on-board regulator generating 3.3 V (3
     V3_R). There are status indicators and reset control for the main controller.\n\n
- Central logic: A central MCU/SoC (U? in the middle) with support chips around it (U2, U4,
      etc.). The layout and surrounding parts suggest USB interfacing, clock distribution,
     and signal conditioning for high-speed or mixed-signal operation.\n\n
- Interfaces and I/O:\n
  - USB-related signals and power sense (VBUS, 3V3, regulator, likely USB transceiver
       support).\n
  - Debug/programming headers: TX/RX (UART), SDA/SCK (I2C/SPI family), GND, 3V3, HW_RESET,
       etc., for easy access to the core.\n
  - Four expansion/connector banks at the bottom labeled RJ1-RJ4, likely for external
       modules, sensors, or peripheral buses.\n
  - Front-panel indicators: ACT/READY and multiple PxLED indicators plus a bank of 7-
       segment/LED displays for status or simple readouts.\n
  - Left-side I/O: additional header holes/pads for more GPIO, power, or signals.\n\n
- Layout and purpose: The dense routing and high-speed nets imply a mixed-signal board
     intended for rapid prototyping of peripherals, bus interfaces (I2C/SPI/UART), and
     external module integration; the four RJ ports and front-panel indicators support
     quick connection and visibility of activity, status, and measurements.\n\n
In short: a USB-powered, 3.3 V logic development board with a central MCU/SoC, multiple
     buses (I2C/SPI/UART), several expansion RJ ports, and visible status indicators for
     debugging, testing, and fast prototyping."
```

"Gemini-2.5-Pro prediction": "Based on an expert analysis of the PCB layout, here is a
    concise description of its function:\n\nThis PCB is a high-performance, multi-port **
    network switch or a complex network interface card**.\n\n### Key Functional Aspects:\n
    \n*   **Core Function:** The board's primary role is to manage and route network
    traffic. The central BGA chip (U3) is likely a dedicated **network switch IC or an
    FPGA/SoC** that acts as the system's core, handling data packet switching between the
    various ports.\n\n*   **Network Interfaces:**\n    *   **Four RJ45 Ethernet Ports (RJ1
    -RJ4):** These provide standard 10/100/1000 Mbps copper Ethernet connectivity. The
    associated LEDs (labeled 100/1000, ACT/Ready) indicate link speed and activity status
    for each port.\n    *   **One SFP Port:** The footprint on the left is for a Small
    Form-factor Pluggable (SFP) cage. This allows for flexible, high-speed connectivity,
    typically for 1G or higher-speed fiber optic or copper transceivers.\n\n*   **System
    Control and Management:**\n    *   **Management MCU (U2):** A smaller microcontroller
    (U2) likely manages board-level functions, such as initializing the main switch IC,
    controlling LEDs, and providing external management interfaces.\n    *   **
    Configuration/Debug Ports:** The board includes a **USB port** (for power,
    configuration, or firmware updates), a **UART serial console** (TX/RX pins), and an **
    I2C header** (SDA/SCK), all common for device management and debugging.\n\n*   **Power
     Management:** The circuit includes an onboard power delivery system featuring DC-DC
    switching regulators (U4, U5) and inductors (L1, L2)."

"Gemini-2.5-Flash-Lite prediction": "This PCB board appears to be a control or interface
    board, likely for a system involving multiple communication channels and status
    indicators.\n\nKey functions and components observed:\n\n*   **Power Distribution:**
    Multiple \"VBUS\" and \"GND\" connectors/pads suggest power input and distribution.
    There are also voltage regulators or power conditioning components (e.g., L1, L2, U4,
    U5) likely providing different voltage rails (e.g., 3V3_R, 1V1_R).\n*   **LED
    Indicators:** Numerous LEDs (LED1-LED10) are present, along with associated current-
    limiting resistors (R5-R10, R36-R43, R57-R80). These likely indicate system status,
    activity, or error conditions. The labels \"ACT\", \"READY\", \"PxLED0\", and \"PxLED1
    \" further support this.\n*   **Communication Interfaces:**\n    *   \"SDA\" and \"SCK
    \" signals indicate an I2C or similar serial communication interface.\n    *   \"TX\"
    and \"RX\" signals suggest a UART or other serial data transmission interface.\n    *
     The presence of multiple RJ connectors (RJ1, RJ2, RJ3, RJ4) with labeled pins (P1-
    P10) implies modular or external connectivity, possibly for sensors, actuators, or
    other peripherals.\n*   **Control and Logic:** The central IC (U2, likely a
    microcontroller or FPGA) is the core processing unit, managing the various functions
    and interfaces. Components like R12, R13, R14, R16, R19, R23-R35, R44-R56, C7-C15, C36
    , C42, C44, C45, C47-C51, C54, C57-C60, C62, C68, C72, C73, C75, and B1 (likely a
    crystal or oscillator) support the operation of this central IC.\n*   **Reset and
    Configuration:** \"HW_RESET\" and \"S_RESET\" signals indicate hardware and software
    reset capabilities. \"GPIO03\" suggests general-purpose input/output pins for
    additional control or monitoring.\n\nIn summary, this board acts as a central hub for
    receiving power, processing data, communicating with external devices via serial
    interfaces and modular connectors, and providing visual feedback through LEDs."

"LLaMA-4-Maverick prediction": "The PCB board depicted in the image is a complex electronic
     circuit board with multiple components and functionalities. The board appears to be a
     network switch or a similar device, given the presence of four RJ45 connectors (RJ1-
    RJ4) at the bottom, which are commonly used for Ethernet connections.\n\n**Key
    Components and Features:**\n\n1. **RJ45 Connectors (RJ1-RJ4):** These connectors are
    used for Ethernet connections, indicating that the board is designed for networking
    applications.\n2. **LED Indicators:** A series of LEDs (LED1-LED10) are present on the
     board, which are likely used to indicate the status of various functions such as link
     activity, power, or other operational states.\n3. **Voltage Regulators:** The
    presence of voltage regulators (e.g., U4 and U5) suggests that the board is designed
    to operate with different voltage levels, possibly converting an input voltage to a
    stable output voltage for the components on the board.\n4. **Microcontroller or
    Processor:** Although not explicitly labeled, the central component (U3) is likely a
    microcontroller or processor, given its size and the complexity of the surrounding
    circuitry. This component is crucial for controlling the board's operations.\n5. **
    Power Supply:** The board has a power input section, indicated by the \"VBUS\" labels,
     suggesting that it is designed to be powered via a USB connection or another external
     power source.\n6. **Reset and Configuration Options:** The presence of reset buttons
    or switches (e.g., S_RESET and HW_RESET) allows for resetting the device to its
    default state or configuring it in different modes.\n\n**Functionality:**\n\nThe PCB
    board is designed to manage and facilitate Ethernet connections through the four RJ45
    connectors. It likely includes features such as:\n\n* Network switching or routing\n*
    LED indicators for status monitoring\n* Voltage regulation for stable operation\n*
    Control and processing via a microcontroller or processor\n* Reset and configuration
    options for device management\n\nOverall, the board is a sophisticated electronic
    device designed for networking applications, with a range of features and
    functionalities that enable it to manage and facilitate Ethernet connections."

"Internvl3-78b prediction": "This PCB appears to be a complex interface board, likely for a
     networking or communication application. Here's a concise overview of its functions:\
    n\n1. **Power Management**: \n  - Includes voltage regulators (3V3, 1V1) for power
    supply regulation.\n  - VBUS lines indicate connections for power distribution.\n\n2.
    **Communication Interfaces**:\n  - RJ1, RJ2, RJ3, RJ4 connectors suggest Ethernet or
    serial communication ports.\n  - SDA, SCK, TX, RX pins indicate I2C and UART

interfaces for data communication.\n\n3. **LED Indicators**:\n  – Multiple LED arrays
(P6LED0, PxLED0) for status indication (e.g., activity, link speed).\n\n4. **Reset
and Control**:\n  – HW_RESET and GPIO03 pins for hardware reset and general-purpose
input/output control.\n\n5. **Connectivity**:\n  – Multiple ground (GND) connections
ensure stable power and signal integrity.\n  – Connectors for external devices or
modules (e.g., USB, GPIO).\n\n6. **Signal Integrity**:\n  – Careful routing and
decoupling capacitors (C48, C49, etc.) to minimize noise and ensure reliable operation
.\n\nOverall, the board facilitates communication, power management, and status
monitoring in a networked or embedded system."

"Qwen-VL-Max prediction": "This PCB is a **microcontroller-based development or interface
board** designed for embedded applications, featuring power management, communication
interfaces, and user feedback elements. Key functions include:\n\n- **Power Supply**:
Incorporates voltage regulators (U1: 3.3V, U5: 1.1V) to provide stable power to
digital components from a VBUS input, likely USB-powered.\n- **Microcontroller
Interface**: Features a central microcontroller (U2) with GPIO, UART (TX/RX), I2C (SCL
/SDA), and reset (HW_RESET) signals, enabling communication and control.\n- **LED
Indicators**: Multiple LEDs (LED1-LED10) with current-limiting resistors provide
visual status feedback (e.g., ACT, READY, PLED).\n- **External Connectivity**:
Includes multiple headers (RJ1-RJ4) for peripheral expansion, likely for sensors,
actuators, or other modules.\n- **Clocking**: Uses an external crystal (X1) with load
capacitors (C7, C8) for precise timing.\n- **USB Interface**: The VBUS and USB-related
traces suggest USB connectivity, possibly for programming or power.\n- **Reset and
Debug**: HW_RESET and GPIO3 suggest support for hardware reset and debugging (e.g.,
JTAG/SWD via GPIOs).\n- **Decoupling and Filtering**: Numerous capacitors (C1-C90)
ensure stable power delivery and minimize noise.\n- **Layout**: Dual-layer design with
clear signal routing, ground planes, and power distribution (VBUS, 3V3, 1V1),
indicating attention to signal integrity and EMI control.\n\nOverall, this board
serves as a **compact, USB-powered embedded platform** for prototyping or system
integration, supporting digital I/O, serial communication, and real-time feedback."

"Qwen3-vl-8b-instruct prediction": "This PCB is a **multi-port USB 3.0 hub controller board
**, designed to expand a single USB host port into multiple downstream USB ports while
managing power and data.\n\n**Key Functions:**\n\n1.  **Port Expansion:** Provides
multiple USB ports (likely 4x USB 3.0, indicated by RJ1-RJ4 connectors) for connecting
peripherals.\n2.  **Power Management:** Features dedicated power regulators (U4, U5)
to convert and distribute 5V/3.3V/1.8V rails for the controller and ports. Includes
power monitoring (P6LED0, P6LED1) and current limiting (R60-R77).\n3.  **Data Routing
:** Uses a USB 3.0 hub controller (U3) to route data between the upstream VBUS port
and the downstream ports.\n4.  **Communication & Control:** Includes an I2C interface
(SDA/SCK) for configuration or status monitoring, and a hardware reset (HW_RESET) and
software reset (S_RESET) for system control.\n5.  **Status Indication:** LED
indicators (LED1-LED10) provide visual feedback on port activity, power status, and
potentially error conditions.\n6.  **Signal Integrity:** Uses differential pairs (e.g
., TX/RX, SDA/SCK) for high-speed USB 3.0 data transmission, with proper routing and
termination.\n7.  **Power Distribution:** VBUS traces are routed to each port,
ensuring adequate power delivery to connected devices.\n\nIn essence, it acts as a
bridge, allowing a single USB host to connect to multiple USB devices while managing
power, data, and status signals."

"Qwen3-vl-235b-a22b-thinking prediction": "\n\nThis PCB functions as a **multi – port
Ethernet switch/hub with Power over Ethernet (PoE) capability**. Key aspects:  \n– **
Connectivity**: Four RJ45 ports (RJ1-RJ4) support 100M/1000M Ethernet, enabling
network data transmission. \n– **Power Management**: Voltage regulators (U4/U5 for 3.3
V/1.1V) and VBUS distribution provide regulated power; PoE is implied via RJ45 (common
in Ethernet – powered devices). \n– **Status Indication**: LEDs (ACT, READY, P6LED0
/1) monitor port activity, link status, and speed (100M/1000M). \n– **Control &
Communication**: I2C (SDA/SCK), UART (TX/RX), and reset signals (S_RESET, HW_RESET)
enable microcontroller – based configuration and diagnostics.  \n– **Signal Integrity
**: Ground planes (GND) and controlled trace routing ensure stable high – speed data
transmission.  \n\nIt integrates network switching, power delivery, and system
monitoring in a compact layout."

# D  LLM USAGE STATEMENT

We use large language models (LLMs) only as a general-purpose assistant tool for minor tasks,
including grammar checking, correcting formatting errors, and polishing individual sentences.

As our paper is a benchmark-oriented study, we also use LLMs as tools during dataset construction
(e.g., format conversion). These usages are explicitly described in the corresponding sections of the
main text.

In addition, we evaluate several widely used LLMs as part of our experimental study. This consti-
tutes standard benchmarking practice and is reported in the experimental section.

