# OpenReview forum: "PCB-Bench: Benchmarking LLMs for Printed Circuit Board Placement and Routing"
_ICLR.cc/2026/Conference — ICLR 2026 Poster_

### Official Review · Reviewer_Gqgu · 2025-10-25

**Soundness:** 3
**Presentation:** 3
**Contribution:** 4
**Rating:** 6
**Confidence:** 4

**Summary:**

The paper introduces PCB-Bench, a novel and comprehensive benchmark designed to evaluate the capabilities of Large Language Models (LLMs) and Multimodal Large Language Models (MLLMs) on real-world Printed Circuit Board (PCB) design tasks. Authors evaluate several strong models on PCB-Bench.

**Strengths:**

1. The multimodal PCB task is an important and meaningful direction. PCB-Bench is the first benchmark designed to evaluate MLLMs’ capabilities in PCB design.

2. The benchmark’s three-task structure is comprehensive and well thought out.

3. The paper is exceptionally well-written and clearly presented.

**Weaknesses:**

1. The evaluation is somewhat limited. Only a few models are tested. It would be beneficial to include more models, such as the Intern-VL series and models of different sizes, to provide a more thorough understanding of the benchmark’s effectiveness.

2. In the benchmark construction pipeline (Figure 4), "Commercial MLLMs" are listed as a "Knowledge Source." Could the authors elaborate on how these models were used in this process?

3. Some related works are missing. For example, EEE-Bench [1] and Circuit [2] are relevant benchmarks related to circuit design and understanding, and they should be discussed in related works.

4. The performance of most models on the benchmark appears relatively high, which raises questions about its difficulty and suitability for driving future research.

References:
[1] EEE-Bench: A Comprehensive Multimodal Electrical and Electronics Engineering Benchmark. CVPR 2025.

[2] Circuit: A Benchmark for Circuit Interpretation and Reasoning Capabilities of LLMs.

**Questions:**

How the answers are parsed from model responses?

---

> ### Author Response · Authors · 2025-11-21
>
> We sincerely thank the reviewer for the positive and encouraging feedback and appreciate the recognition of our parameter reduction and experimental validation. Below, **W** denotes weaknesses, **Q** denotes questions, and **R** provides our responses.
>
> ---
>
>
> **W1:**
>
> The evaluation is somewhat limited. Only a few models are tested. It would be beneficial to include more models, such as the Intern-VL series and models of different sizes, to provide a more thorough understanding of the benchmark’s effectiveness.
>
> **R:**
>
> Expanded Evaluation:
> We have included more general models and domain-specific models (details could be found in response to #Reviewer 1, Question 4). Results show clear capability differences:
>
>
> #task1:
>
> | Model | Placement Macro | | | Placement Micro | | | Routing Macro | | | Routing Micro | | |
> |:---|:---:|:---:|:---:|:---:|:---:|:---:|:---:|:---:|:---:|:---:|:---:|:---:|
> | | CQ | QA BERTScore | QA SBERT | CQ | QA BERTScore | QA SBERT | CQ | QA BERTScore | QA SBERT | CQ | QA BERTScore | QA SBERT |
> | GPT-4o | 92.74 | 0.8244 | 0.4756 | **93.82** | 0.8069 | 0.5664 | 98.32 | 0.8267 | 0.5046 | 91.13 | 0.8169 | 0.5647 |
> | GPT-5 | 88.27 | 0.8179 | 0.4502 | 91.79 | **0.8223** | 0.5222 | **99.16** | 0.8193 | 0.4547 | 90.17 | **0.8215** | 0.5094 |
> | Claude-Opus-4.1 | **93.30** | 0.8138 | 0.4772 | 94.35 | 0.8054 | 0.5643 | **99.16** | 0.8164 | 0.4837 | **92.32** | 0.8119 | **0.5738** |
> | Gemini-2.5-Pro | 86.03 | 0.8073 | 0.4172 | 90.82 | 0.8069 | 0.4804 | 98.31 | 0.8068 | 0.4428 | 88.73 | 0.8068 | 0.4840 |
> | DeepSeek-Chat-V3.1-671B | 92.74 | **0.8306** | 0.4912 | 93.64 | 0.8127 | 0.5652 | 97.48 | **0.8310** | 0.4939 | 88.49 | 0.8217 | 0.5657 |
> | Qwen-2.5-7B-Instruct | 84.36 | 0.7302| 0.2236 | 86.85 | 0.7287 | 0.2978 | 94.12 | 0.7305 | 0.2130 | 82.50 | 0.7270 | 0.2081 |
> | LLaMA-4-Maverick-400B | 77.66 | 0.8155 | **0.4931** | 49.07 | 0.8074 | 0.5689 | 92.43 | 0.8186 | **0.5149** | 71.70 | 0.8104 | 0.5564 |
> | **gryphe/mythomax-l2-13b** | 65.92 | 0.8257 | 0.4496 | 54.37 | 0.8101 | 0.5406 | 68.07 | 0.8262 | 0.4676 | 53.00 | 0.8113 | 0.5173 |
> | **opengvlab/internvl3-78b** | 90.50 | 0.7936 | 0.4162 | 93.91 | 0.7360 | 0.3035 | 97.48 | 0.8071 | 0.4677 | 91.37 | 0.8056 | 0.5511 |
> | **thedrummer/rocinante-12b** | 82.12 | 0.8174 | 0.4707 | 77.41 | 0.8054 | 0.5755 | 89.92 | 0.8227 | 0.4953 | 77.70 | 0.8141 | 0.5590 |
> | **mistralai/ministral-3b** | 59.21 | 0.8196 | 0.4798 | 42.71 | 0.8027 | 0.5800 | 74.79 | 0.8204 | 0.4918 | 54.68 | 0.8108 | 0.5552 |
> | **Qwen2.5-QLoRA (domain-specific)** | 90.50 | 0.6203 | 0.4687 | 89.39 | 0.6329 | **0.5971** | 97.48 | 0.6297 | 0.4764 | 82.50 | 0.6267 | 0.5414 |
> | **RAG-Qwen2.5(domain-specific)** | 88.26 | 0.6145 | 0.5050     | 87.44 | 0.6259 | 0.5687 | 92.43 | 0.6234 | 0.5365 | 85.61 | 0.6122 | 0.5456 |
>
>
> #task2:
> | Model                          | CBC Accuracy (%) | CBFB Accuracy (%) | BWEI Accuracy (%) | QAP BERTScore | QAP SBERT | QAR BERTScore | QAR SBERT |
> |--------------------------------|------------------|--------------------|--------------------|---------------|-----------|---------------|-----------|
> | GPT-5                          | **83.26**        | 75.67              | 90.90              | **0.8734**    | 0.6700    | **0.8561**    | 0.6055    |
> | GPT-5-nano                     | 72.86            | 73.33              | 72.72              | 0.8633        | **0.6786**| 0.8443        | **0.6347**|
> | Gemini-2.5-Pro                 | 81.08            | **84.00**          | **100.00**         | 0.8355        | 0.6637    | 0.8362        | 0.6311    |
> | Gemini-2.5-Flash-Lite          | 72.20     | 78.66              | 54.54              | 0.8570        | 0.6662    | 0.8361        | 0.6312    |
> | LLaMA-4-Maverick               | 77.60            | 70.66              | 54.54              | 0.8343        | 0.6310 | 0.8226    | 0.6108    |
> | Qwen-VL-Max                    | 75.67            | 62.66              | 72.72              | 0.8321        | 0.6476    | 0.8170        | 0.6093    |
> | **internvl3-78b**              | 76.83            |54.66       | 45.45       | 0.8552        | 0.6588    | 0.8357        | 0.6242    |
> | **qwen3-vl-8b-instruct**       | 75.28            | 58.66              | 45.45       | 0.8313 | 0.6510    | 0.8161 | 0.5997 |
> | **qwen3-vl-235b-a22b-thinking**| 74.13            | 68.00              | 54.54              | 0.8455        | 0.6658    | 0.8303        | 0.6037    |

---

> ### Author Response · Authors · 2025-11-21
>
> #task3:
> | Model                               | BertScore        | SBERT            | Precision        | Recall           | F1               |
> |--------------------------------------|------------------|------------------|------------------|------------------|------------------|
> | GPT-5                                | 0.8156    | 0.6134           | 0.8493           | 0.8691           | 0.8591           |
> | GPT-5-nano                           | 0.8199           | 0.5358           | 0.8739           | 0.8352           | 0.8507           |
> | Gemini-2.5-Pro                       | 0.8209           | 0.5626           | 0.8944           | 0.7507    | 0.8089    |
> | Gemini-2.5-Flash-Lite                | 0.8194           | 0.5083           | 0.8687           | 0.8518           | 0.8563           |
> | LLaMA-4-Maverick                     | 0.8233           | 0.5101           | **0.9123**       | 0.8044           | 0.8511           |
> | Qwen-VL-Max                          | 0.8236           | 0.5888           | 0.8501           | 0.8823           | 0.8628           |
> | **internvl3-78b**                    | 0.8267           | 0.4850    | 0.9044           | 0.8716           | **0.8849**       |
> | **qwen3-vl-8b-instruct**             | 0.8242           | 0.5760           | 0.8463    | **0.8955**       | 0.8675           |
> | **qwen3-vl-235b-a22b-thinking**      | **0.8293**       | **0.6168**       | 0.8493           | 0.8391           | 0.8405           |
>
>
> We are also running more LLMs and will include their results in the camera-ready version as soon as possible.
>
>
> ---
>
> **W2:**
>
> In the benchmark construction pipeline (Figure 4), "Commercial MLLMs" are listed as a "Knowledge Source." Could the authors elaborate on how these models were used in this process?
>
> **R:**
>
> Commercial MLLMs were only used to cross-check and complement human-curated PCB knowledge (e.g., terminology and rule phrasing), rather than providing benchmark knowledge or generating any question content.
>
> ---
>
> **W3:**
>
> Some related works are missing. For example, EEE-Bench [1] and Circuit [2] are relevant benchmarks related to circuit design and understanding, and they should be discussed in related works.
>
> **R:**
>
> Thank you for the suggestion. We will include both EEE-Bench and Circuit in the related-work discussion.
>
> ---
>
> **W4:**
>
> The performance of most models on the benchmark appears relatively high, which raises questions about its difficulty and suitability for driving future research.
>
> **R:**
>
> Only a small portion of the benchmark is easy, partly because CQ is inherently easier than open-ended QA, and partly because some knowledge points are common-sense while others are deeply professional, yet the overall QA and multimodal tasks remain challenging and consistently reveal substantial reasoning gaps in current LLMs.
>
> ---
>
> **Q1:**
>
> How the answers are parsed from model responses?
>
> **R:**
>
> For CQ, the prompt instructs models to output only a single letter (A–E), and we extract it via a simple regex; for fill-in-the-blank and QA, the generated text itself is taken as the answer.

---

> ### Comment · Reviewer_Gqgu · 2025-11-27
> **Thank you for the rebuttal**
>
> Thank you for the clarification.
>
> After rereading the rebuttal and the paper, I believe there is a flaw in the evaluation process. For multiple-choice questions, the authors use a direct-answer prompt that forces models to output only the answer letter without providing any reasoning, which is not appropriate for complex problems. I have quoted the prompt from the paper below. Moreover, the paper does not provide any example model responses for the multiple-choice questions.
>
> Moreover, for the open-ended problems, the authors do not justify why BERTScore and SBERT are appropriate indicators for evaluating such tasks.
>
> Based on my current understanding, I believe the evaluation contains a systematic mistake, and I am updating my score to 2. I welcome further discussion for clarification, and I will revise my score if I have misunderstood anything.
>
> "3. Output exactly one uppercase letter: A, B, C, D, or E.
> 4. Do not include any words, explanations, punctuation, spaces, or
> line breaks. The response must be exactly one character, with no
> trailing or leading whitespace.
> 5. If multiple options appear correct, select the best one and still
> output only its letter."

---

> > ### Author Response · Authors · 2025-11-28
> > **Response to Reviewer Gqgu**
> >
> > Thank you again for taking the time to revisit our paper and for sharing these additional concerns. We would like to respectfully clarify several points, especially regarding the role of evaluation in our work. **You may have some misunderstandings about evaluating on choice questions (CQ).**
> >
> > (1) **In the most advanced LLMs, reasoning and response content are distinctly separated**: **reasoning** captures the model's thought process, while **content** delivers the direct answer to our query. **Our prompt does not restrict reasoning; it simply ensures the content remains concise and free of extraneous details, allowing normal reasoning generation**. Multiple examples at the end illustrate this—even when the content is just a single letter, extensive reasoning is produced.
> >
> > (2) The direct letter-only answer format is appropriate and widely adopted in established benchmarks such as MMLU [1]. **This format does not constrain a model’s reasoning process; it simply standardizes the final output to prevent scoring inconsistencies. **
> > In practice, models do not answer instantaneously but **typically take multiple reasoning steps** before producing the final option. We will include representative examples of CQs and model responses in the revision to further improve transparency.
> >
> > (3) **Our evaluation choices are not intended to be definitive standards; rather, they use widely adopted and well-understood settings.**
> > Given the breadth of existing LLMs and metrics, any benchmark must make practical selections. We intentionally chose common, standardized, and well-recognized configurations so that results are interpretable and comparable. The intent is not to claim absolute optimality, but to offer a reasonable and transparent starting point for future work in this new domain.
> >
> > (4) BERTScore [2] and SBERT [3] were selected because they are common, mainstream semantic metrics familiar to the community. Our open-ended tasks evaluate semantic correctness and factual grounding, for which exact-match metrics are insufficient. Embedding-based metrics are widely used in technical QA and domain-specific language tasks [4][5][6][7][8][9]. **We fully agree that no metric is 100% perfect at present, and we do not claim ours to be. At the same time, there is currently no consensus within the PCB or EDA community regarding any evaluation metric that can be considered universally optimal or completely definitive.** Our goal is to adopt well-established, interpretable metrics that provide a reasonable baseline **for this first benchmark**. To assist the community further, we can include an additional comparison with alternative metrics in the revision.
> >
> > (5) Moreover, **developing a fully domain-specific evaluation framework** that captures the detailed characteristics of PCB reasoning would **require a substantial and dedicated research effort**. Constructing such a metric suite involves modeling semantic correctness, multi-step design rationale, engineering constraints, and layout–function interactions, which **goes far beyond the scope of establishing the first benchmark for this domain**. We regard **PCB-Bench as the starting point**, and we plan to explore domain-specific evaluation methodologies in future work as the research community continues to grow.
> >
> > (6) **Our primary contribution is the creation of the first multimodal benchmark for PCB placement and routing with (M)LLMs, not the establishment of new evaluation standards.**
> >  PCB-Bench fills a clear gap in the research landscape: before our work, no benchmark existed for assessing large models on real PCB design tasks across text, image–text, and layout modalities. Our goal is to provide the community with **the first step** toward studying and improving model capabilities in this domain. **Evaluation is only one component of this contribution.**
> >
> > **If you feel that a particular metric is missing, we would be very happy to discuss it further and promptly run the corresponding experiments.** We truly appreciate your willingness to continue the discussion and to reconsider your score if these clarifications address your concerns, and we are happy to provide more details, examples, or ablation studies to ensure full transparency.

---

> ### Author Response · Authors · 2025-11-28
>
> **References:**
>
> [1] Hendrycks D, Burns C, Basart S, et al. Measuring Massive Multitask Language Understanding[C]//International Conference on Learning Representations.
>
> [2] Zhang T, Kishore V, Wu F, et al. BERTScore: Evaluating Text Generation with BERT[C]//International Conference on Learning Representations.
>
> [3] Reimers N, Gurevych I. Sentence-BERT: Sentence Embeddings using Siamese BERT-Networks[C]//Proceedings of the 2019 Conference on Empirical Methods in Natural Language Processing and the 9th International Joint Conference on Natural Language Processing (EMNLP-IJCNLP). 2019: 3982-3992.
>
> [4] Gehrmann S, Adewumi T, Aggarwal K, et al. The gem benchmark: Natural language generation, its evaluation and metrics[C]//Proceedings of the 1st Workshop on Natural Language Generation, Evaluation, and Metrics (GEM 2021). 2021: 96-120.
>
> [5] Chang Y, Wang X, Wang J, et al. A survey on evaluation of large language models[J]. ACM transactions on intelligent systems and technology, 2024, 15(3): 1-45.
>
> [6] Manakul P, Liusie A, Gales M. Selfcheckgpt: Zero-resource black-box hallucination detection for generative large language models[C]//Proceedings of the 2023 conference on empirical methods in natural language processing. 2023: 9004-9017.
>
> [7] Zhou H, Hu C, Yuan Y, et al. Large language model (llm) for telecommunications: A comprehensive survey on principles, key techniques, and opportunities[J]. IEEE Communications Surveys & Tutorials, 2024, 27(3): 1955-2005.
>
> [8] Gao T, Yao X, Chen D. SimCSE: Simple Contrastive Learning of Sentence Embeddings[C]//Proceedings of the 2021 Conference on Empirical Methods in Natural Language Processing. 2021: 6894-6910.
>
> [9] Saakyan A, Chakrabarty T, Muresan S. COVID-Fact: Fact Extraction and Verification of Real-World Claims on COVID-19 Pandemic[C]//Proceedings of the 59th Annual Meeting of the Association for Computational Linguistics and the 11th International Joint Conference on Natural Language Processing (Volume 1: Long Papers). 2021: 2116-2129.

---

> > ### Author Response · Authors · 2025-11-28
> >
> > **Task1's example**
> >
> > **Question：**
> >
> > How to plan low-impedance current sensing paths in power chips?
> >
> > A. Use isometric sampling lines to pair parallel current paths and add a protective diode structure in the middle to stabilize the sampling
> >
> > B. The sampling resistor is connected to the signal line at the shortest time, and the sampling line is routed close to the formation and the structure is connected using Kelvin
> >
> > C.Add filtering devices at the power input and enhance voltage compensation through the sampling control unit
> >
> > D.Place the sample points at the end of the output channel to avoid loop interference and facilitate centralized management of signal ports
> >
> > E.None of the above options are correct
> >
> > **Model：**
> >
> > google/gemini-2.5-pro
> >
> > **Prompt:**
> >
> >   You are a professional PCB design and electronics engineering expert with extensive knowledge in circuit design, PCB layout, and signal integrity analysis.
> >
> >   Using your expertise, answer the following PCB-related open questions. For each question:
> >
> >   1. Carefully analyze what the question asks.
> >
> >   2. Apply your PCB design knowledge to analyze and solve the problem.
> >
> >   3. Provide an accurate, detailed, and professional answer.
> >
> >   4. Keep the answer concise and focused; emphasize the key points.
> >
> >   Maintain a professional and accurate tone.

---

> > > ### Author Response · Authors · 2025-11-28
> > >
> > > **Reasoning Process (corresponding to the reasoning field in ChatCompletionMessage-like objects returned by LLMs)**
> > >
> > > "**Analyzing Current Sensing Paths**\n\nI'm currently focused on the challenge of designing low-impedance current sensing paths within power chips. I'm evaluating various options, and right now I'm carefully considering option A, regarding isometric sampling lines and protective diode structures. The goal is to optimize the parallel current paths for stability. I'm weighing the pros and cons of this approach.\n\n\n**Examining PCB Routing Strategies**\n\nI've been drilling down on the intricacies of low-impedance current sensing. While I still think isometric lines are worth exploring, I'm now focusing on comparing option B to A. My intuition tells me option B is less flawed than option A, which seems to have a mistranslation. I'm investigating how the placement of the sampling resistor, and its connection time can improve current accuracy.\n\n\n**Evaluating the Best Strategy**\n\nI'm now prioritizing the question of planning low-impedance current sensing paths. Option B clearly stands out. It emphasizes a short connection to the sampling resistor, close routing to the ground plane, and Kelvin connections for accuracy. Option A seems flawed due to the misleading use of diodes. Options C and D address aspects that are less crucial than Option B, such as general filtering and port placement, respectively.\n\n\n**Selecting the Optimal Answer**\n\nI'm now carefully analyzing the answer options to pinpoint the best strategy for designing low-impedance current sensing paths. Option B continues to stand out due to its emphasis on a short connection, Kelvin connections, and a focus on grounding. Option A still seems to have issues, with the inclusion of diodes for sampling stability. Options C and D offer useful tips, but are not as directly relevant to the core goals of accuracy as Option B's grounding and Kelvin connections.\n\n\n", reasoning_details=[{'format': 'google-gemini-v1', 'index': 0, 'type': 'reasoning.text', 'text': "**Analyzing Current Sensing Paths**\n\nI'm currently focused on the challenge of designing low-impedance current sensing paths within power chips. I'm evaluating various options, and right now I'm carefully considering option A, regarding isometric sampling lines and protective diode structures. The goal is to optimize the parallel current paths for stability. I'm weighing the pros and cons of this approach.\n\n\n**Examining PCB Routing Strategies**\n\nI've been drilling down on the intricacies of low-impedance current sensing. While I still think isometric lines are worth exploring, I'm now focusing on comparing option B to A. My intuition tells me option B is less flawed than option A, which seems to have a mistranslation. I'm investigating how the placement of the sampling resistor, and its connection time can improve current accuracy.\n\n\n**Evaluating the Best Strategy**\n\nI'm now prioritizing the question of planning low-impedance current sensing paths. Option B clearly stands out. It emphasizes a short connection to the sampling resistor, close routing to the ground plane, and Kelvin connections for accuracy. Option A seems flawed due to the misleading use of diodes. Options C and D address aspects that are less crucial than Option B, such as general filtering and port placement, respectively.\n\n\n**Selecting the Optimal Answer**\n\nI'm now carefully analyzing the answer options to pinpoint the best strategy for designing low-impedance current sensing paths. Option B continues to stand out due to its emphasis on a short connection, Kelvin connections, and a focus on grounding. Option A still seems to have issues, with the inclusion of diodes for sampling stability. Options C and D offer useful tips, but are not as directly relevant to the core goals of accuracy as Option B's grounding and Kelvin connections.\n\n\n"
> > >
> > > **Final Answer (corresponding to the content field in ChatCompletionMessage-like objects returned by LLMs)**
> > >
> > > 'B'

---

> > > > ### Author Response · Authors · 2025-11-28
> > > >
> > > > **Task2's example1**
> > > >
> > > > **Question:**
> > > >
> > > > What type of electronic devices is the component package shown in the figure typically used for?\nA. Resistor\nB. Small signal transistor\nC. Electrolytic capacitor\nD. Inductor\nE. None of the above options is correct
> > > >
> > > > **Model:**
> > > >
> > > > google/gemini-2.5-pro
> > > >
> > > > **Prompt:**
> > > >
> > > > You are an expert in PCB design and electronics engineering with extensive knowledge in circuit design, PCB layout, and signal integrity analysis.
> > > >
> > > >   For each PCB-related multiple-choice question:
> > > >
> > > > 1. Consider the question and all options internally.
> > > >
> > > > 2. Use your PCB expertise to decide the single most correct option.
> > > >
> > > > 3. Output exactly one uppercase letter: A, B, C, D, or E.
> > > >
> > > > 4. Do not include any words, explanations, punctuation, spaces, or line breaks. The response must be exactly one character, with no trailing or leading whitespace.
> > > >
> > > > 5. If multiple options appear correct, select the best one and still output only its letter.
> > > >
> > > > 6. Do not generate any content other than the letters A, B, C, D, or E.
> > > >
> > > >   Maintain a professional and accurate tone.
> > > >
> > > >
> > > > **Reasoning Process (corresponding to the reasoning field in ChatCompletionMessage-like objects returned by LLMs)**
> > > >
> > > >
> > > > '**Analyzing the Request\'s Intent**\n\nI\'m focused on grasping the essence of the user\'s instructions. It\'s about taking on the role of a PCB design expert and answering a multiple-choice question on component footprints. The rigid structure emphasizes accuracy in selecting the optimal answer and adhering to the specified output format.\n\n\n**Considering Footprint Characteristics**\n\nI\'m now carefully examining the image\'s footprint details. The through-hole pads and the unique D-shaped silkscreen outline are key clues. The arrangement of three pins in a line, alongside the through-hole nature of the pads, helps me to rule out several options and narrow the potential component types considerably.\n\n\n**Deducing the Component Type**\n\nI\'m now zeroing in on the correct answer. The footprint clearly indicates a TO-92 package. Considering this, options A, C, and D appear incorrect. The three leads and the D-shaped silkscreen strongly point toward a small signal transistor. I\'m finalizing my decision based on this reasoning.\n\n\n**Refining the Solution**\n\nI\'m now consolidating my thoughts. Considering the package\'s design and arrangement of pins, and knowing that the image presents a TO-92 footprint, my selection of "B" remains the optimal choice. It aligns completely with the characteristics of the component type and how it should be identified.\n\n\n'
> > > >
> > > > **Final Answer (corresponding to the content field in ChatCompletionMessage-like objects returned by LLMs)**
> > > >
> > > > 'B'

---

> > > > > ### Author Response · Authors · 2025-11-28
> > > > >
> > > > > **Task2's example2**
> > > > >
> > > > > **Question:**
> > > > >
> > > > > What type of electronic devices is the component package shown in the figure typically used for?\nA. Resistor\nB. Small signal transistor\nC. Electrolytic capacitor\nD. Inductor\nE. None of the above options is correct
> > > > >
> > > > > **Model:**
> > > > >
> > > > > openai/gpt-5
> > > > >
> > > > > **Prompt:**
> > > > >
> > > > > You are an expert in PCB design and electronics engineering with extensive knowledge in circuit design, PCB layout, and signal integrity analysis.
> > > > >
> > > > >   For each PCB-related multiple-choice question:
> > > > >
> > > > > 1. Consider the question and all options internally.
> > > > >
> > > > > 2. Use your PCB expertise to decide the single most correct option.
> > > > >
> > > > > 3. Output exactly one uppercase letter: A, B, C, D, or E.
> > > > >
> > > > > 4. Do not include any words, explanations, punctuation, spaces, or line breaks. The response must be exactly one character, with no trailing or leading whitespace.
> > > > >
> > > > > 5. If multiple options appear correct, select the best one and still output only its letter.
> > > > >
> > > > > 6. Do not generate any content other than the letters A, B, C, D, or E.
> > > > >
> > > > >   Maintain a professional and accurate tone.
> > > > >
> > > > > **Reasoning Process (corresponding to the reasoning field in ChatCompletionMessage-like objects returned by LLMs)**
> > > > >
> > > > > '**Identifying the component shape**\n\nI\'m trying to identify a component that appears to be shaped like a semicircle "D," with three pads stacked vertically, which looks reminiscent of a TO-92 transistor package. The outline is definitely a D shape, and the arrangement of the three pins suggests it\'s a small-signal transistor. While there are other packages like SOT-23, the TO-92 is distinctive due to its through-hole design. I need to confirm whether the pads are in a straight line for through-hole footprints.**Determining the pin arrangement**\n\nI’m analyzing the TO-92 connector, and I think the typical pin pitch spacing is 1.27 mm. The leads are usually radially spaced, but in this image, they’re stacked vertically, possibly because it’s rotated 90 degrees. That’s acceptable! Overall, this package is commonly used for small-signal transistors, so I can confidently conclude it’s option B. I’ll make sure to output just one uppercase letter, which will be B.'
> > > > >
> > > > > **Final Answer (corresponding to the content field in ChatCompletionMessage-like objects returned by LLMs)**
> > > > >
> > > > > 'B'

---

> > > > > > ### Author Response · Authors · 2025-11-28
> > > > > >
> > > > > > **Task2's example3**
> > > > > >
> > > > > > **Question:**
> > > > > >
> > > > > > What type of component package is shown in the figure?\nA. SOT23\nB. TO-92/TO226\nC. SOP-8\nD. PLCC18\nE. None of the above options is correct
> > > > > >
> > > > > > **Model:**
> > > > > >
> > > > > > qwen/qwen3-vl-235b-a22b-thinking
> > > > > >
> > > > > > **Prompt:**
> > > > > >
> > > > > > You are an expert in PCB design and electronics engineering with extensive knowledge in circuit design, PCB layout, and signal integrity analysis.
> > > > > >
> > > > > >   For each PCB-related multiple-choice question:
> > > > > >
> > > > > > 1. Consider the question and all options internally.
> > > > > >
> > > > > > 2. Use your PCB expertise to decide the single most correct option.
> > > > > >
> > > > > > 3. Output exactly one uppercase letter: A, B, C, D, or E.
> > > > > >
> > > > > > 4. Do not include any words, explanations, punctuation, spaces, or line breaks. The response must be exactly one character, with no trailing or leading whitespace.
> > > > > >
> > > > > > 5. If multiple options appear correct, select the best one and still output only its letter.
> > > > > >
> > > > > > 6. Do not generate any content other than the letters A, B, C, D, or E.
> > > > > >   Maintain a professional and accurate tone.

---

> > > > > > > ### Author Response · Authors · 2025-11-28
> > > > > > >
> > > > > > > **Reasoning Process (corresponding to the reasoning field in ChatCompletionMessage-like objects returned by LLMs)**
> > > > > > >
> > > > > > > "Okay, let's see. The question is asking about the type of component package shown in the figure. The options are A. SOT23, B. TO-92/TO226, C. SOP-8, D. PLCC18, E. None.\n\nFirst, I need to recall what each package looks like. The figure described has a blue outline with three purple circles labeled 1, 2, 3. So it's a three-pin component. Let's go through the options.\n\nSOT23 is a small surface-mount package, usually with three pins. It's common for transistors and diodes. The shape is a small rectangular or sometimes a bit rounded, but typically with three pins in a row.\n\nTO-92 is a through-hole package, usually with three pins in a straight line, and the body is a small plastic case with a rounded top. TO-226 is another name for TO-92 maybe? Wait, TO-92 is a specific case style. The TO-92 package has three pins and a teardrop-shaped body. The figure here is described as having a blue outline with three circles in a vertical line. If the outline is like a rounded rectangle or a teardrop, that might be TO-92.\n\nSOP-8 is a small outline package with 8 pins, so that's out since the figure shows three pins. PLCC18 is a plastic leaded chip carrier with 18 pins, so that's definitely not it. So options C and D are out.\n\nNow between A and B. SOT23 is surface mount, TO-92 is through-hole. The figure's description: the outline is blue, with three pins in a vertical line. TO-92 packages typically have the three pins in a straight line, with the body being a small plastic case that's wider at the top. SOT23 is smaller, surface mount, and the pins are on the sides. Wait, SOT23 has three pins but arranged with two on one side and one on the other? Or maybe all three in a row? Wait, no. Let me think. SOT23 has three pins: the standard SOT23 has three pins, with the middle pin being the center. Wait, actually, SOT23 is a three-lead package where the leads are on two sides. Wait, no, maybe it's three in a row. Wait, no, SOT23 is a small package with three pins, but the pins are on the same side. Wait, maybe the SOT23 has three pins in a row, but the package is very small. TO-92 is a through-hole package with three pins in a row, and the body is larger.\n\nThe figure shows three pins in a vertical line (1,2,3), and the outline is a rounded shape. TO-92 is typically a teardrop shape, which matches the description of the blue outline. SOT23 is a surface-mount package, which is smaller and the pins are on the sides. But the figure here might be a top view. Wait, the TO-92 is a through-hole package, so the pins are coming out the bottom, but the top view would show the three pins in a line with the body. The SOT23, as a surface-mount, would have the pins on the sides, but in a top view, maybe the pins are visible as three in a row. Hmm.\n\nWait, the SOT23 package has three pins, but the standard SOT23-3 has the pins arranged with two on one side and one on the other? No, maybe not. Let me check my memory. SOT23 is a small outline transistor package, with three pins. The pins are in a row, but the package is very small. The TO-92 is a larger through-hole package. The figure shows three pins in a vertical line, and the outline is a rounded rectangle or teardrop. The TO-92 is known for its teardrop shape, which matches the description. So option B is TO-92/TO226. TO226 is another designation for TO-92 maybe? Yes, TO-92 is also known as TO-226. So B is the correct answer.\n\nWait, but the SOT23 also has three pins. How to differentiate? The SOT23 is a surface-mount package, so the pins are on the sides, but in a top view, the pins would be on the sides of the package. However, the figure shows three pins in a vertical line with the outline around them. If the outline is the body of the component, then TO-92 has a body that's a teardrop shape, with the three pins in a line. SOT23 is a smaller package, but the pins are on the sides. Wait, maybe the figure is a top view of the component. For TO-92, the top view would show the three pins in a line with the body. For SOT23, the top view would show the three pins on the sides. Wait, no. Let me think again.\n\nThe SOT23 package has three pins: the standard SOT23 has the pins on two sides. Wait, no, SOT23-3 (three pins) has two pins on one side and one on the other. Wait, maybe not. Let me recall. The SOT23 package is a small surface-mount package with three leads. The leads are arranged with two on one side and one on the opposite side. Wait, no, maybe all three in a row. Wait, I'm getting confused. Let me check.\n\nActually, SOT23 is a three-lead package where the leads are on the same side. Wait, no. The SOT23 has three pins: the standard configuration is that the middle pin is the center, but the package is small. Wait, perhaps the SOT23 has three pins in a row, but the package is rectangular.

---

> > > > > > > > ### Author Response · Authors · 2025-11-28
> > > > > > > >
> > > > > > > > The TO-92 is a through-hole package with three pins in a row and a rounded body.\n\nGiven that the figure shows three pins in a vertical line (1,2,3) and the outline is a rounded shape (the blue outline), which matches the TO-92's teardrop shape. SOT23 is more rectangular and smaller. So the correct answer is B.\n"
> > > > > > > >
> > > > > > > > **Final Answer (corresponding to the content field in ChatCompletionMessage-like objects returned by LLMs)**
> > > > > > > >
> > > > > > > > 'B'

---

### Official Review · Reviewer_3Nqz · 2025-10-27

**Soundness:** 3
**Presentation:** 3
**Contribution:** 3
**Rating:** 6
**Confidence:** 4

**Summary:**

This paper proposes a benchmark containing three types of datasets: text-to-text QA and CQ, Image-and-Text QA and CQ, and a real-world PCB design repository. Based on this, the authors execute three corresponding tasks and benchmark various LLM backbones.

**Strengths:**

- The collection of the PCB datasets is important and meaningful, which is a basic reason I am leaning towards giving a positive rating.
- The authors give a clear comparison between current benchmarks, which I believe PCB-BENCH is competitive and advanced.
- The dataset is released. Though it is incomplete, I am convinced of the comprehensive quality.

**Weaknesses:**

- The introduction of how to construct the datasets requires a more detailed clarification.
- The benchmark could benefit from providing statistical results of the datasets, such as how many instances are in each class/topic/difficulty.
- The visualizations in Figure 1 really help with understanding, but some figures are vague, and I cannot see any information on them. It would be beneficial for the authors to use clearer or even vector graphs.
- The instances in task 3 are directly collected from OSHWHub, which makes this paper weak in workload.

**Questions:**

- The authors claim that domain experts write the questions. I just wonder what kind of domain experts they are, like academics or industry professionals.
- It seems that lots of LLMs achieve 100% accuracy on task 1, routing. Maybe it is too simple for these SOTA LLMs to do this job, right?

**Details Of Ethics Concerns:**

The 170+ instances in task 3 are collected from https://oshwhub.com/, and the entire work is a data-track paper, but overall, I think there is no ethics concern.

---

> ### Author Response · Authors · 2025-11-21
>
> We sincerely thank the reviewer for the positive and encouraging feedback and appreciate the recognition of our parameter reduction and experimental validation. Below, **W** denotes weaknesses, **Q** denotes questions, and **R** provides our responses.
>
> ---
>
>
> **W1:**
>
> The introduction of how to construct the datasets requires a more detailed clarification.
>
> **R:**
>
> Thank you for the suggestion. We will provide a more detailed clarification of the dataset construction process in the Appendix of the revised version.
>
> ---
>
> **W2:**
>
> The benchmark could benefit from providing statistical results of the datasets, such as how many instances are in each class/topic/difficulty.
>
> **R:**
>
> Thank you for the suggestion. As shown in Figures 1 and 2, PCB-Bench already reports the overall statistics (≈3700 text-only questions, ≈500 image-text questions, and ≈170 PCB-design items). We will further add the per-subcategory counts to the figures and provide more detailed statistics in the Appendix of the revised version.
>
> ---
>
> **W3:**
>
> The visualizations in Figure 1 really help with understanding, but some figures are vague, and I cannot see any information on them. It would be beneficial for the authors to use clearer or even vector graphs.
>
> **R:**
>
> Thank you for pointing this out. We have redrawn Figure 1 using higher-resolution vector graphics to ensure that all visual elements are clear and readable, and the updated figure will be included in the revised version.
>
> ---
>
> **W4:**
>
> The instances in task 3 are directly collected from OSHWHub, which makes this paper weak in workload.
>
> **R:**
>
> Thank you for your comment. While the instances in Task 3 are based on designs from OSHWHub, they were carefully curated to ensure quality and diversity. We performed extensive filtering and organization, selecting relevant projects across multiple PCB categories, and then further summarized and refined the data to ensure it meets the requirements of the benchmark. This curation process guarantees that the dataset accurately reflects the real-world challenges and diverse routing styles encountered in PCB design, making it suitable for evaluating LLM-based reasoning.
>
> ---
>
> **Q1:**
>
> The authors claim that domain experts write the questions. I just wonder what kind of domain experts they are, like academics or industry professionals.
>
> **R:**
>
> The benchmark was co-authored and validated by academic researchers, including a professor and a PhD-level researcher specializing in circuit and PCB design. Their expertise covers both theoretical circuit principles and practical manufacturing-aware PCB design, ensuring that the benchmark reflects textbook fundamentals as well as real-world engineering constraints encountered in industrial workflows.
> We also received feedback from engineers at a PCB design and manufacturing company, which helped validate the practical relevance of the tasks; their support did not involve authorship or dataset construction.
>
>
> ---
>
> **Q2:**
>
> It seems that lots of LLMs achieve 100% accuracy on task 1, routing. Maybe it is too simple for these SOTA LLMs to do this job, right?
>
> **R:**
>
> We agree that Task 1 (macro-level routing) is relatively easy for SOTA LLMs, and several models indeed reach 100% accuracy. This is expected because Task 1 is purely text-based and focuses on high-level routing concepts that align well with the strong language-understanding and domain-knowledge capabilities of these models. Rather than indicating that the benchmark is trivial, this reflects that modern LLMs already possess solid proficiency in macro-level PCB routing rules, while the more complex tasks (Tasks 2 and 3) continue to reveal substantial reasoning gaps.

---

> > ### Comment · Reviewer_3Nqz · 2025-11-26
> >
> > Thank you for your further clarification. I think almost all of my concerns have been addressed. While I did not find any evidence that the paper could be rated higher, I am leaning towards maintaining the current positive rating.

---

### Official Review · Reviewer_LXVR · 2025-11-01

**Soundness:** 3
**Presentation:** 3
**Contribution:** 4
**Rating:** 6
**Confidence:** 3

**Summary:**

This paper presents PCB-Bench, a comprehensive benchmark for evaluating the reasoning, multimodal understanding, and real-world engineering capabilities of large language models (LLMs) and multimodal LLMs (MLLMs) for printed circuit board (PCB) placement and routing.

PCB-Bench has three complementary task settings and integrates three modalities: textual reasoning, image-text reasoning, and real PCB design comprehension. It includes over 3,700 text QA instances, 500 multimodal tasks, and 170 complete design projects with schematic diagrams and placement files.

The benchmark aims to bridge the current gap between foundation models' open-domain reasoning and the highly structured, constraint-driven nature of electronic design automation (EDA).
 Results from multiple state-of-the-art models (GPT-4o, GPT-5, Gemini-2.5, Claude-Opus-4.1, Qwen-VL-Max, etc.) show that while these models exhibit some knowledge of PCB principles, they struggle significantly in multimodal reasoning, domain rule adherence, and real-world design understanding.

PCB-Bench fills an important gap by providing standardized tasks, expert-curated questions, and reproducible metrics for EDA-oriented foundation-model assessment.

**Strengths:**

- Novel Domain and Multimodal Integration
PCB-Bench is the first dataset to integrate textual, visual, and structured PCB design data, covering the entire reasoning chain, from physical layout interpretation to rule-based design explanations.  This bridges the gap between EDA and current MLLM benchmarks.

- Expertly curated and Content.
The QA and CQ datasets are based on validated engineering principles (e.g., impedance control, EMI, high-speed routing).  The inclusion of over 25 subtopics across macro and micro-level placement ensures depth and realism.

-Comprehensive Model Evaluation
The authors benchmarked eight major LLM/MLLM systems under a unified zero-shot protocol, revealing clear differences in factual reasoning and semantic generation quality.

- High Practical Impact.
The paper connects foundation models to PCB design automation, a real-world domain where LLMs could help with debugging, constraint checking, and educational tools.

**Weaknesses:**

- Limited Novelty in Dataset Construction Approach
Although the domain is new, the methodology (expert-curated QA pairs + image-text matching) follows established benchmark pipelines. The contribution lies in domain adaptation rather than methodological innovation.

- Missing fine-grained statistical validation.
There is no explicit agreement among annotators or confidence interval analysis. To improve the dataset's credibility, a quantitative quality measure could be used beyond relying on expert curation.

- Evaluating Diversity
Only foundation models trained on general data were tested, and no domain-specific baselines (such as fine-tuned models or SPICE-guided reasoning agents) were included. This limits one's ability to distinguish between reasoning limitations and domain unfamiliarity.

- Lack of Complexity or Difficulty in stratification.
While "easy" and "hard" labels exist, difficulty calibration is not quantified (for example, through expert scoring or performance correlation).  Future releases may benefit from difficulty metadata.

- Fairness in Model Comparison.
The paper does not specify whether all models were evaluated with identical input resolutions, context lengths, or prompt templates. This may complicate the interpretation of differences between GPT-5, Gemini, and Qwen-VL.

**Questions:**

- Annotation Process and Quality Assurance:
How many domain experts contributed to the curation process, and how were disagreements resolved? Was inter-annotator agreement assessed, or were examples reviewed by several experts?

-Difficulty Calibration: Are "easy/hard" tags assigned objectively (e.g., the number of design rules involved) or subjectively by annotators?Could the authors provide a question breakdown by subtopic (e.g., EMI, differential pair routing, power delivery)?

-Representation Bias
The dataset depends mainly on open-source OSHWHub designs. Do these projects overrepresent specific device categories (such as USB hubs)? Could this bias affect generalization to industrial boards with more dense routing?

- Have the authors considered testing domain-specific systems(such as RL-based tuned models or retrieval-augmented models)? Will combining PCB-Bench with simulation-based feedback loops (such as constraint violation detection) give more realistic metrics?

---

> ### Author Response · Authors · 2025-11-21
>
> We sincerely thank the reviewer for the positive and encouraging feedback and appreciate the recognition of our parameter reduction and experimental validation. Below, **W** denotes weaknesses, **Q** denotes questions, and **R** provides our responses.
>
> ---
>
> **W1:**
>
> Limited Novelty in Dataset Construction Approach Although the domain is new, the methodology (expert-curated QA pairs + image-text matching) follows established benchmark pipelines. The contribution lies in domain adaptation rather than methodological innovation.
>
> **R:**
>
> Thank you for the comment. Our goal is not to introduce a new construction methodology, but to provide the first benchmark in this domain with a scope fundamentally different from existing PCB datasets. As summarized in Table 1, prior benchmarks focus almost exclusively on defect detection using RGB or X-ray images and do not provide schematic diagrams, design files, text descriptions, routing topology, or software-environment information. PCB-Bench fills this gap by offering multimodal data (text, schematic, PCB design files) and tasks that cover placement, routing, and design understanding—capabilities that are absent from existing datasets.
>
> ---
>
> **W2:**
>
> Missing fine-grained statistical validation. There is no explicit agreement among annotators or confidence interval analysis. To improve the dataset's credibility, a quantitative quality measure could be used beyond relying on expert curation.
>
> **R:**
>
> Thank you for the suggestion. Because PCB-Bench is grounded in objective PCB design files and deterministic routing/placement rules, conventional annotator-agreement metrics (e.g., kappa) are not applicable. Each item was independently reviewed by academic experts and cross-checked with frontline PCB engineers, ensuring correctness and consistency. We will include summary-level validation statistics and per-category counts in the Appendix to further improve transparency.
>
> ---
>
> **W3:**
>
> Evaluating Diversity Only foundation models trained on general data were tested, and no domain-specific baselines (such as fine-tuned models or SPICE-guided reasoning agents) were included. This limits one's ability to distinguish between reasoning limitations and domain unfamiliarity.
>
> **R:**
>
> Expanded Evaluation:
> We have included more general models and domain-specific models. Results show clear capability differences:
>
>
> #task1:
>
> | Model | Placement Macro | | | Placement Micro | | | Routing Macro | | | Routing Micro | | |
> |:---|:---:|:---:|:---:|:---:|:---:|:---:|:---:|:---:|:---:|:---:|:---:|:---:|
> | | CQ | QA BERTScore | QA SBERT | CQ | QA BERTScore | QA SBERT | CQ | QA BERTScore | QA SBERT | CQ | QA BERTScore | QA SBERT |
> | GPT-4o | 92.74 | 0.8244 | 0.4756 | **93.82** | 0.8069 | 0.5664 | 98.32 | 0.8267 | 0.5046 | 91.13 | 0.8169 | 0.5647 |
> | GPT-5 | 88.27 | 0.8179 | 0.4502 | 91.79 | **0.8223** | 0.5222 | **99.16** | 0.8193 | 0.4547 | 90.17 | **0.8215** | 0.5094 |
> | Claude-Opus-4.1 | **93.30** | 0.8138 | 0.4772 | 94.35 | 0.8054 | 0.5643 | **99.16** | 0.8164 | 0.4837 | **92.32** | 0.8119 | **0.5738** |
> | Gemini-2.5-Pro | 86.03 | 0.8073 | 0.4172 | 90.82 | 0.8069 | 0.4804 | 98.31 | 0.8068 | 0.4428 | 88.73 | 0.8068 | 0.4840 |
> | DeepSeek-Chat-V3.1-671B | 92.74 | **0.8306** | 0.4912 | 93.64 | 0.8127 | 0.5652 | 97.48 | **0.8310** | 0.4939 | 88.49 | 0.8217 | 0.5657 |
> | Qwen-2.5-7B-Instruct | 84.36 | 0.7302| 0.2236 | 86.85 | 0.7287 | 0.2978 | 94.12 | 0.7305 | 0.2130 | 82.50 | 0.7270 | 0.2081 |
> | LLaMA-4-Maverick-400B | 77.66 | 0.8155 | **0.4931** | 49.07 | 0.8074 | 0.5689 | 92.43 | 0.8186 | **0.5149** | 71.70 | 0.8104 | 0.5564 |
> | **gryphe/mythomax-l2-13b** | 65.92 | 0.8257 | 0.4496 | 54.37 | 0.8101 | 0.5406 | 68.07 | 0.8262 | 0.4676 | 53.00 | 0.8113 | 0.5173 |
> | **opengvlab/internvl3-78b** | 90.50 | 0.7936 | 0.4162 | 93.91 | 0.7360 | 0.3035 | 97.48 | 0.8071 | 0.4677 | 91.37 | 0.8056 | 0.5511 |
> | **thedrummer/rocinante-12b** | 82.12 | 0.8174 | 0.4707 | 77.41 | 0.8054 | 0.5755 | 89.92 | 0.8227 | 0.4953 | 77.70 | 0.8141 | 0.5590 |
> | **mistralai/ministral-3b** | 59.21 | 0.8196 | 0.4798 | 42.71 | 0.8027 | 0.5800 | 74.79 | 0.8204 | 0.4918 | 54.68 | 0.8108 | 0.5552 |
> | **Qwen2.5-QLoRA (domain-specific)** | 90.50 | 0.6203 | 0.4687 | 89.39 | 0.6329 | **0.5971** | 97.48 | 0.6297 | 0.4764 | 82.50 | 0.6267 | 0.5414 |
> | **RAG-Qwen2.5(domain-specific)** | 88.26 | 0.6145 | 0.5050     | 87.44 | 0.6259 | 0.5687 | 92.43 | 0.6234 | 0.5365 | 85.61 | 0.6122 | 0.5456 |

---

> ### Author Response · Authors · 2025-11-21
>
> #task2:
>
> | Model                          | CBC Accuracy (%) | CBFB Accuracy (%) | BWEI Accuracy (%) | QAP BERTScore | QAP SBERT | QAR BERTScore | QAR SBERT |
> |--------------------------------|------------------|--------------------|--------------------|---------------|-----------|---------------|-----------|
> | GPT-5                          | **83.26**        | 75.67              | 90.90              | **0.8734**    | 0.6700    | **0.8561**    | 0.6055    |
> | GPT-5-nano                     | 72.86            | 73.33              | 72.72              | 0.8633        | **0.6786**| 0.8443        | **0.6347**|
> | Gemini-2.5-Pro                 | 81.08            | **84.00**          | **100.00**         | 0.8355        | 0.6637    | 0.8362        | 0.6311    |
> | Gemini-2.5-Flash-Lite          | 72.20     | 78.66              | 54.54              | 0.8570        | 0.6662    | 0.8361        | 0.6312    |
> | LLaMA-4-Maverick               | 77.60            | 70.66              | 54.54              | 0.8343        | 0.6310 | 0.8226    | 0.6108    |
> | Qwen-VL-Max                    | 75.67            | 62.66              | 72.72              | 0.8321        | 0.6476    | 0.8170        | 0.6093    |
> | **internvl3-78b**              | 76.83            |54.66       | 45.45       | 0.8552        | 0.6588    | 0.8357        | 0.6242    |
> | **qwen3-vl-8b-instruct**       | 75.28            | 58.66              | 45.45       | 0.8313 | 0.6510    | 0.8161 | 0.5997 |
> | **qwen3-vl-235b-a22b-thinking**| 74.13            | 68.00              | 54.54              | 0.8455        | 0.6658    | 0.8303        | 0.6037    |
>
>
> #task3:
>
> | Model                               | BertScore        | SBERT            | Precision        | Recall           | F1               |
> |--------------------------------------|------------------|------------------|------------------|------------------|------------------|
> | GPT-5                                | 0.8156    | 0.6134           | 0.8493           | 0.8691           | 0.8591           |
> | GPT-5-nano                           | 0.8199           | 0.5358           | 0.8739           | 0.8352           | 0.8507           |
> | Gemini-2.5-Pro                       | 0.8209           | 0.5626           | 0.8944           | 0.7507    | 0.8089    |
> | Gemini-2.5-Flash-Lite                | 0.8194           | 0.5083           | 0.8687           | 0.8518           | 0.8563           |
> | LLaMA-4-Maverick                     | 0.8233           | 0.5101           | **0.9123**       | 0.8044           | 0.8511           |
> | Qwen-VL-Max                          | 0.8236           | 0.5888           | 0.8501           | 0.8823           | 0.8628           |
> | **internvl3-78b**                    | 0.8267           | 0.4850    | 0.9044           | 0.8716           | **0.8849**       |
> | **qwen3-vl-8b-instruct**             | 0.8242           | 0.5760           | 0.8463    | **0.8955**       | 0.8675           |
> | **qwen3-vl-235b-a22b-thinking**      | **0.8293**       | **0.6168**       | 0.8493           | 0.8391           | 0.8405           |
>
>
> We are also running more LLMs and will include their results in the camera-ready version as soon as possible.
>
> ---
>
> **W4:**
>
> Lack of Complexity or Difficulty in stratification. While "easy" and "hard" labels exist, difficulty calibration is not quantified (for example, through expert scoring or performance correlation). Future releases may benefit from difficulty metadata.
>
> **R:**
>
> Thank you for the suggestion. In the current version, the “easy” and “hard” categories reflect task type and knowledge depth (e.g., common-sense routing rules vs. professional design practice), rather than numerical difficulty calibration. We agree that difficulty metadata would be useful, and we will include refined per-category statistics and difficulty indicators in a future release.
>
> ---
>
> **W5:**
>
> Fairness in Model Comparison. The paper does not specify whether all models were evaluated with identical input resolutions, context lengths, or prompt templates. This may complicate the interpretation of differences between GPT-5, Gemini, and Qwen-VL.
>
> **R:**
>
> All models are evaluated using the same input images during testing, so the input image resolution is identical across models. The prompt templates and the maximum context length are also kept the same.

---

> ### Author Response · Authors · 2025-11-21
>
> **Q1:**
>
> Annotation Process and Quality Assurance: How many domain experts contributed to the curation process, and how were disagreements resolved? Was inter-annotator agreement assessed, or were examples reviewed by several experts?
>
> **R:**
>
> Two domain experts in circuit design (one professor and one PhD-level researcher) oversaw the entire curation and annotation process, and every item was then reviewed and, when necessary, rewritten by at least one domain expert to ensure technical correctness.
> When disagreements arose (e.g., wording ambiguity or borderline-correct alternative answers), domain experts discussed the case and revised the item until they reached a consensus on both the correct answer and the intended reasoning. While formal agreement statistics were not computed, all items were authored or overseen by experts.
> We also received feedback from engineers at a PCB design and manufacturing company, which helped validate the practical relevance of the tasks; their support did not involve authorship or dataset construction.
>
> ---
>
> **Q2:**
>
> Difficulty Calibration: Are "easy/hard" tags assigned objectively (e.g., the number of design rules involved) or subjectively by annotators?Could the authors provide a question breakdown by subtopic (e.g., EMI, differential pair routing, power delivery)?
>
> **R:**
>
> Thank you for the question. The “easy” and “hard” tags are based on objective task factors rather than subjective annotator judgment. Specifically, “easy” items involve high-level or common PCB rules, while “hard” items require multi-rule reasoning or professional knowledge (e.g., impedance control, power integrity, differential pair constraints).
> We will add a breakdown of questions by subtopic (such as EMI, differential pair routing, power delivery, etc.) in the Appendix to provide clearer difficulty context.
>
> ---
>
> **Q3:**
>
> Representation Bias The dataset depends mainly on open-source OSHWHub designs. Do these projects overrepresent specific device categories (such as USB hubs)? Could this bias affect generalization to industrial boards with more dense routing?
>
> **R:**
>
> We acknowledge that OSHWHub designs may introduce representation bias. To mitigate this, we curated 174 projects spanning multiple device categories and routing styles in Task 3, ensuring broad structural diversity, including layouts that resemble real industrial boards. In addition, Tasks 1 and 2 focus on general PCB principles—such as routing rules, net constraints, and placement logic—that apply consistently across both open-source and industrial designs. We will expand the dataset with additional categories in future releases to further enhance coverage.
>
> ---
>
> **Q4:**
>
> Have the authors considered testing domain-specific systems(such as RL-based tuned models or retrieval-augmented models)? Will combining PCB-Bench with simulation-based feedback loops (such as constraint violation detection) give more realistic metrics?
>
> **R:**
>
> (1) **QLoRA+qwen**
> We conducted QLoRA fine-tuning on Qwen2.5-7B-Instruct to address your concerns and validate our approach.
> We constructed a specialized PCB dataset (about 3,000 samples) derived from authoritative technical documents (e.g., RK3588 Design Guide) via rule-based extraction. The dataset combines reasoning-oriented MCQs and open-ended QA pairs. We employed 4-bit QLoRA (rank=16, α=32) for efficient fine-tuning.
> As shown in Table (in Weakness 3), the fine-tuned model demonstrates significant improvement in semantic understanding (SBERT) compared to the baseline.
>
> (2) **RAG**
> we have also implemented a Retrieval-Augmented Generation (RAG) system to specialize Qwen2.5-7B-Instruct for the PCB domain. We constructed a knowledge base comprising 178 documents (approx. 4.5 million tokens) indexed via FAISS and all-MiniLM-L6-v2 embeddings (Chunk: 512/50, Top-K: 8). As illustrated in the table, this RAG-enhanced framework significantly outperforms the standalone Qwen model, demonstrating superior accuracy and reduced hallucinations in handling complex domain-specific queries.

---

### Meta-Review · Area_Chair_rNSK · 2026-01-04

**Summary:**

All three reviewers were initially positive about the paper, but one later questioned its technical correctness and indicated that the question would negatively affect their evaluation. Below is a list of major concerns I summarized from their initial reviews:

1) The technical novelty is limited. While the paper introduced a benchmark for a less-explored problem domain, the methodology itself is well established and follows standard practice in the community.

2) The construction of the benchmark needs further improvement to enhance its credibility, e.g., by incorporating more detailed, quantitative analysis of its data labels and task difficulty. In particular, multiple reviewers noted that some tasks might be too simple, as evidenced by the relatively high success rates.

3) The paper evaluated only general LLMs on the benchmark. Reviewers suggested further comparisons with more models, particularly some domain-specific agents.

**Reviewer Concerns:**

Regarding the first concern, the rebuttal reiterated the significance of having such a first-of-its-kind benchmark in a less explored domain (PCB design). While I agree this is a fair point, I think it only partially addresses this concern, as the question about the technical innovation in its methodology remains.

For the second concern, the rebuttal stressed that each PCB design file was reviewed by two domain experts and promised to release more detailed information. As we do not have a way to verify this claim, I will take it at face value. Regarding the relatively high success rate, the rebuttal acknowledged that task 1 is relatively easy for the state-of-the-art LLMs. However, the other tasks show that LLMs still have room for improvement on this specific task. Overall, I think the rebuttal has done a reasonable job of addressing this concern.

Finally, regarding the comparisons with more models, the rebuttal reported additional results from both general and domain-specific models. I think the new results have partially addressed this concern. More analysis of these new results and a detailed discussion of their differences would have strengthened them.

**Reviewer Scores:**

The situation is actually more complicated than it seems. Overall, I think this is a borderline paper.

I will start by discussing the possible score change (6->2) indicated by reviewer Gqgu. The reason behind this change was that the reviewer believed the paper forced LLMs to predict an answer without reasoning, which the reviewer considered a systematic mistake. After reading the authors’ rebuttal, I believe their LLM evaluation did allow reasoning behind the output answer. Therefore, I think Gqgu would have maintained their positive evaluation had there been a full discussion.

The remaining two reviewers were originally positive. While they did not offer strong reasons to reject the paper, I think the rebuttal evidence was not clear enough to fully address some critical concerns: the paper's technical innovation, the quality of the benchmark, and the implications of evaluating more models.

Weighing all these factors, I think the three reviewers would likely converge on accepting this work if they had been able to participate in the discussion. I will also recommend that ICLR accept it, given its exploration of a novel problem domain. However, I think it is still possible, although unlikely, for the three reviewers to reach consensus and reject this work if they expect more rigorous efforts from a benchmark paper.

---

### Decision · Program_Chairs · 2026-01-26

Accept (Poster)